ecology/evolution/palaeontology

Dinosauria, *Diplodocus*, Late Jurassic, Morrison Formation, ontogeny, Sauropoda

**Author for correspondence:**
Philip D. Mannion
e-mail: philipdmannion@gmail.com

# Anatomy and systematics of the diplodocoid *Amphicoelias altus* supports high sauropod dinosaur diversity in the Upper Jurassic Morrison Formation of the USA

Philip D. Mannion[1], Emanuel Tschopp[2,3] and John A. Whitlock[4,5]

[1]Department of Earth Sciences, University College London, London WC1E 6BT, UK
[2]Centrum für Naturkunde, Universität Hamburg, 20146 Hamburg, Germany
[3]Division of Paleontology, American Museum of Natural History, Central Park West at 79th Street, New York, NY 10024-5192, USA
[4]Department of Science and Mathematics, Mount Aloysius College, Cresson, PA 16630-1999, USA
[5]Section of Vertebrate Paleontology, Carnegie Museum of Natural History, Pittsburgh, PA 15213-4007, USA

PDM, 0000-0002-9361-6941; ET, 0000-0002-5245-6910; JAW, 0000-0002-1118-2925

Sauropod dinosaurs were an abundant and diverse component of the Upper Jurassic Morrison Formation of the USA, with 24 currently recognized species. However, some authors consider this high diversity to have been ecologically unviable and the validity of some species has been questioned, with suggestions that they represent growth series (ontogimorphs) of other species. Under this scenario, high sauropod diversity in the Late Jurassic of North America is greatly overestimated. One putative ontogimorph is the enigmatic diplodocoid *Amphicoelias altus*, which has been suggested to be synonymous with *Diplodocus*. Given that *Amphicoelias* was named first, it has priority and thus *Diplodocus* would become its junior synonym. Here, we provide a detailed re-description of *A. altus* in which we restrict it to the holotype individual and support its validity, based on three autapomorphies. Constraint analyses demonstrate that its phylogenetic position within Diplodocoidea is labile, but it seems unlikely that *Amphicoelias* is synonymous with *Diplodocus*. As such, our re-evaluation also leads us to retain *Diplodocus* as a distinct genus. There is no evidence to support the view that any of the currently

recognized Morrison sauropod species are ontogimorphs. Available data indicate that sauropod anatomy did not dramatically alter once individuals approached maturity. Furthermore, subadult sauropod individuals are not prone to stemward slippage in phylogenetic analyses, casting doubt on the possibility that their taxonomic affinities are substantially misinterpreted. An anatomical feature can have both an ontogenetic and phylogenetic signature, but the former does not outweigh the latter when other characters overwhelmingly support the affinities of a taxon. Many Morrison Formation sauropods were spatio-temporally and/or ecologically separated from one another. Combined with the biases that cloud our reading of the fossil record, we contend that the number of sauropod dinosaur species in the Morrison Formation is currently likely to be underestimated, not overestimated.

## 1. Introduction

The Upper Jurassic Morrison Formation of the western USA has yielded a high diversity of sauropods, including some of the most iconic dinosaurs, such as *Brachiosaurus*, *Brontosaurus* and *Diplodocus* [1–3]. *Apatosaurus* and *Camarasaurus* were also abundant components of this fauna, whereas a number of other sauropod taxa are known from far fewer remains [4]. Some of these have only been recognized this century, namely *Galeamopus* [5], *Kaatedocus* [6], *Smitanosaurus* [7] and *Suuwassea* [8], whereas others have been known for much longer, consisting of *Amphicoelias* [9], *Barosaurus* [10], *Haplocanthosaurus* [11] and *Supersaurus* [12]. Currently, 24 sauropod species assigned to 14 genera are recognized as valid in the Morrison Formation (e.g. [3–5,7,13–17]; table 1), although some authors have suggested that many of these species are synonyms, and that they represent growth series of a smaller number of valid taxa [36–40].

One of the earliest named and most enigmatic of Morrison sauropods is *Amphicoelias*. The type species, *Amphicoelias altus*, was erected by Cope in 1877 for two dorsal vertebrae, a partial pubis and a femur (figures 1 and 2). These elements were collected by A. Ripley in the same year, from Cope Quarry XII in Garden Park, north of Cañon City, in Fremont County, CO, USA, and were shipped to Cope by O.W. Lucas [41]. Following Cope's death, his collection was acquired by the American Museum of Natural History (AMNH) in 1902. Two further species of *Amphicoelias* were named by Cope from nearby Morrison Formation localities—*Amphicoelias latus* [9] and *Amphicoelias fragillimus* [35]—but neither species is currently considered to belong to the genus. *Amphicoelias latus* is universally regarded as a synonym of *Camarasaurus supremus* (e.g. [2,5,42]) and *A. fragillimus* was recently referred to the newly erected rebbachisaurid diplodocoid genus, *Maraapunisaurus*, by Carpenter [17]. No further remains can currently be unambiguously referred to *Amphicoelias*, making it one of the rarest taxa in the Morrison Formation, despite being known for over 140 years.

Cope [9] considered *Amphicoelias* to be a close relative of *Camarasaurus*, although he classified them in the separate, monogeneric families, Amphicoeliidae and Camarasauridae, respectively. *Amphicoelias* was generally regarded as either a close relative or synonym of *Camarasaurus* for the following four decades (e.g. [43,44]), before Osborn & Mook [42] undertook a review of Cope's sauropod taxa, in which they argued that *Amphicoelias* was most closely related to the diplodocids *Barosaurus* and *Diplodocus*. Nopcsa [45] also allied *Amphicoelias* with diplodocids and, from Romer [46] onwards, a diplodocid or diplodocoid placement has been universally accepted (e.g. [5,13,47–50]). Rauhut *et al.* [48] were the first to incorporate *Amphicoelias* into a phylogenetic analysis, recovering it as a 'basal' diplodocoid, outside Diplodocimorpha. The diplodocoid-focused analysis of Whitlock [49] also placed *Amphicoelias* as a non-diplodocimorph diplodocoid, a position recovered in subsequent analyses of revised versions of this data matrix (e.g. [51,52]). This placement was also supported in analyses of a recent independent phylogenetic data matrix of eusauropods [50]. By contrast, analyses of a second independent data matrix, focused on diplodocids, recovered *Amphicoelias* within Diplodocidae, either as a 'basal' member of this clade or within Apatosaurinae [5]. The latter position was also supported by analyses of a revised version of this data matrix [16].

Although generally considered a valid genus (e.g. [13,49]), there is not universal agreement on this point. Osborn & Mook [42] and other authors (e.g. [2]) have commented upon similarities between *Amphicoelias* and Morrison diplodocids, especially *Diplodocus*, and a small number of studies have argued that *Amphicoelias* is actually synonymous with *Diplodocus* [4,37,40]. Given that *Amphicoelias* [9] was named before *Diplodocus* [31], and, therefore, has priority, any such synonymization would have notable taxonomic ramifications.

Given the uncertainty of its taxonomic status and phylogenetic affinity, here we provide a detailed re-description of the holotypic remains of *A. altus*, reassess its validity and phylogenetic placement,

**Table 1.** Currently recognized sauropod genera and species in the Upper Jurassic Morrison Formation of North America. *Dyslocosaurus polyonychius* potentially represents an additional valid sauropod species from this formation, but its stratigraphic provenance remains uncertain [5,18]. '*Apatosaurus*' *minimus* might also represent a distinct sauropod species [19], although its affinities require further evaluation [5,13]. Note that the validity of *Diplodocus longus* (highlighted with an asterisk) is disputed and it is unlikely that this species can be diagnosed ([20] [and references therein]).

| taxon | authors |
| --- | --- |
| *Amphicoelias altus* | [9] |
| *Apatosaurus ajax* (type) | [21] |
| *Apatosaurus louisae* | [22] |
| *Barosaurus lentus* | [10] |
| *Brachiosaurus altithorax* | [23] |
| *Brontosaurus excelsus* (type) | [24] |
| *Brontosaurus parvus* | [25] |
| *Brontosaurus yahnahpin* | [26] |
| *Camarasaurus grandis* | [21] |
| *Camarasaurus lentus* | [27] |
| *Camarasaurus lewisi* | [28] |
| *Camarasaurus supremus* (type) | [29] |
| *Diplodocus carnegii* | [30] |
| *Diplodocus longus* (type)* | [31] |
| *Diplodocus hallorum* | [32] |
| *Galeamopus hayi* (type) | [5,33] |
| *Galeamopus pabsti* | [16] |
| *Haplocanthosaurus delfsi* | [34] |
| *Haplocanthosaurus priscus* (type) | [11] |
| *Kaatedocus siberi* | [6] |
| *Maraapunisaurus fragillimus* | [17,35] |
| *Smitanosaurus agilis* | [7,27] |
| *Supersaurus vivianae* | [12] |
| *Suuwassea emilieae* | [8] |

and discuss the systematics of remains previously attributed to this genus. Finally, we present a revised view of sauropod dinosaur diversity in the Upper Jurassic Morrison Formation.

## 1.1. Institutional abbreviations

AC, Beneski Museum of Natural History, Amherst College, Amherst, MA, USA; AMNH, American Museum of Natural History, New York, NY, USA; CM, Carnegie Museum of Natural History, Pittsburgh, PA, USA; CMNH, Cleveland Museum of Natural History, Cleveland, OH, USA; MOR, Museum of the Rockies, Bozeman, MT, USA; USNM, United States National Museum, Smithsonian Institution, Washington, DC, USA; YPM, Yale Peabody Museum, New Haven, CT, USA.

## 1.2. Anatomical abbreviations

ACDL, anterior centrodiapophyseal lamina; ACPL, anterior centroprezygapophyseal lamina; CPOF, centropostzygapophyseal fossa; CPRL, centroprezygapophyseal lamina; lSPOL, lateral spinopostzygapophyseal lamina; mSPOL, medial spinopostzygapophyseal lamina; PACDF, parapophyseal centrodiapophyseal fossa; PCDL, posterior centrodiapophyseal lamina; PCPL, posterior centroparapophyseal lamina; POCDF, postzygapophyseal centrodiapophyseal fossa; PODL,

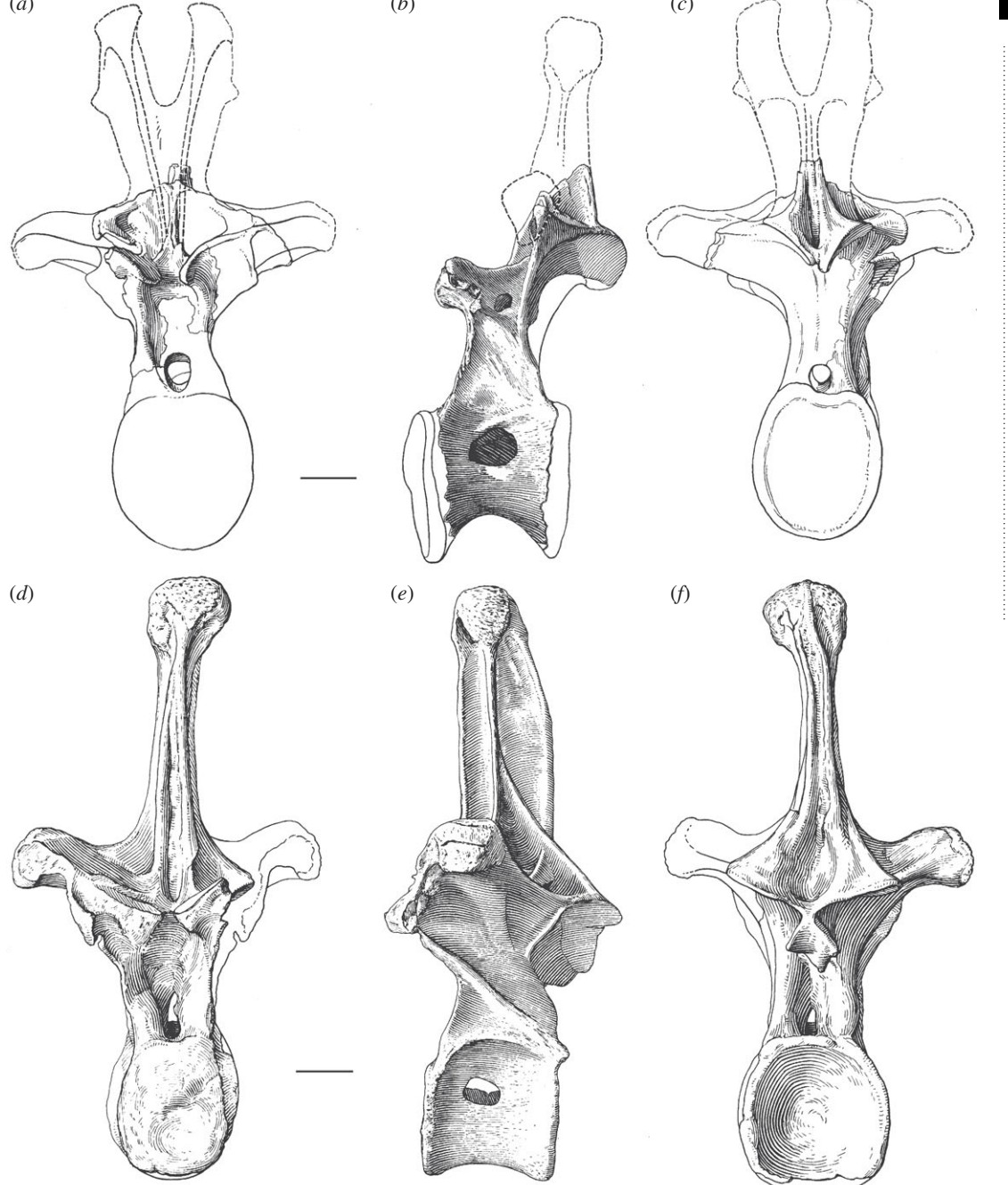

**Figure 1.** Line drawings of *A. altus* holotype dorsal vertebrae (AMNH FARB 5764) as originally presented by Osborn & Mook [42]: middle–posterior dorsal vertebra in (*a*) anterior, (*b*) left lateral and (*c*) posterior views; and posterior dorsal vertebra in (*d*) anterior, (*e*) left lateral and (*f*) posterior views. Dashed lines represent entirely reconstructed regions and unshaded areas reflect sections of the bone covered in consolidant. Scale bar, 100 mm.

postzygodiapophyseal lamina; POSDF, postzygapophyseal spinodiapophyseal fossa; POSL, postspinal lamina; PRDL, prezygodiapophyseal lamina; PRSDF, prezygapophyseal spinodiapophyseal fossa; PRSL, prespinal lamina; SPDL, spinodiapophyseal lamina; SPOL, spinopostzygapophyseal lamina; SPOL-F, spinopostzygapophyseal lamina fossa; SPRL, spinoprezygapophyseal lamina; TPOL, interpostzygapophyseal lamina; TPRL, interprezygapophyseal lamina.

# 2. Systematic palaeontology

SAUROPODA [31]

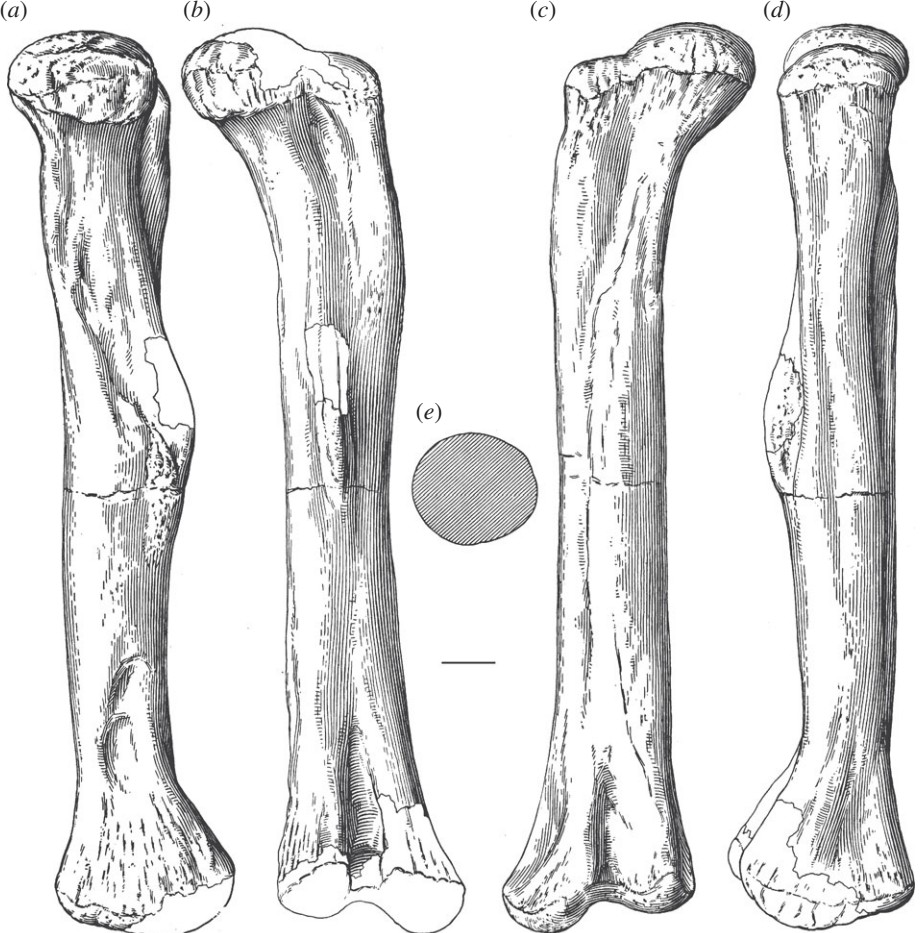

**Figure 2.** Line drawings of *A. altus* holotype right femur (AMNH FARB 5764), as originally presented by Osborn & Mook [42], in (*a*) medial, (*b*) posterior, (*c*) anterior and (*d*) lateral views, with the cross-sectional shape at the midshaft shown in (*e*). Unshaded areas represent entirely reconstructed regions of the femur. Scale bar, 100 mm.

NEOSAUROPODA [53]
DIPLODOCOIDEA [54]
*AMPHICOELIAS* [9]

**Type species**: *Amphicoelias altus* [9]

**Holotype:** AMNH FARB 5764—two middle–posterior dorsal vertebrae, a fragmentary pubis and a right femur (figures 1–8). Based on their recovery from a single locality, the lack of duplication of elements, the absence of size or preservational discrepancies, and that all elements are from the same region of the skeleton, we follow [9] and subsequent authors (e.g. [2,41,42]) in considering these remains as pertaining to a single individual.

**Locality and horizon:** Cope Quarry XII, Garden Park, 8 miles N/NE of Cañon City, Fremont County, CO, USA [9,41,42]; Morrison Formation, C5 systems tract, 150.44–149.21 Ma, lower Tithonian, Upper Jurassic [57–59].

**Revised diagnosis:** *Amphicoelias* can be diagnosed by two autapomorphies (denoted by an asterisk), as well as one local autapomorphy: (i) apex of the posterior dorsal neural spine with rounded, non-tapered lateral projections resulting from the expansion of spinodiapophyseal laminae*; (ii) femoral shaft with subcircular cross-section*; and (iii) femur distally bevelled, with the fibula condyle extending further distally than the tibial condyle.

**Additional information:** Numerous additional remains have been referred to *Amphicoelias* (e.g. [9,35,42,60]). These are discussed and re-evaluated below, but we follow recent authors (e.g. [5,50,51]) in restricting *Amphicoelias* to the type material.

**Curatorial history:** Curatorial history of the type and previously referred material is complicated, mostly because of inadequate field notes from the excavations, but also due to issues emanating from renumbering after the acquisition of Cope's Garden Park collection by the AMNH. After

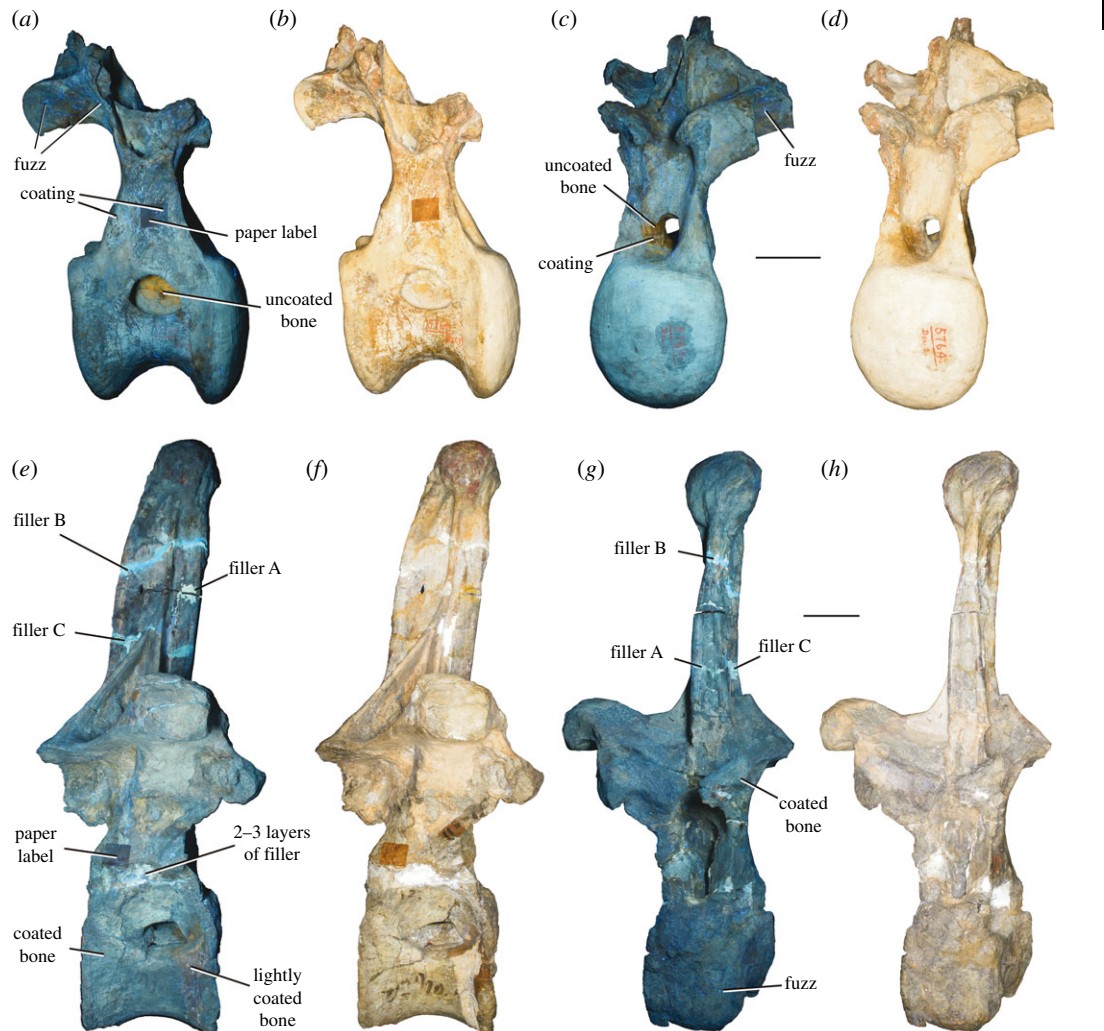

**Figure 3.** Conservation history of *A. altus* holotype dorsal vertebrae (AMNH FARB 5764). The middle–posterior (*a–d*) and posterior dorsal (*e–h*) vertebrae were photographed under UV light (UV-A in *a,d*; combined UV-A, -B and -C wavelengths in *e,h*), and under polarized light to increase contrast (*b,c,f,g*). The vertebrae are shown in (*a,b,e,f*) right lateral and (*c,d,g,h*) anterior views. Note the different types of filler material, indicating several generations of interventions, and the uniform bluish colour, which indicates consolidation with shellac. Shellac coating and attached fuzz remains nearly invisible under normal and polarized light. The individual photographs were not modified with any kind of editing software. A complete set of Progressive Photonics photographs is available in the electronic supplementary material.

acquisition, the material was catalogued with numbers ranging from AMNH FARB 5760 to 5777. The type specimens of the three *Amphicoelias* species erected by Cope [9,35] were catalogued as AMNH FARB 5764 (*A. altus*), AMNH FARB 5765 (*A. latus*) and AMNH FARB 5777 (*A. fragillimus*; this specimen is missing, and it is unclear if it was lost before, during or after the acquisition by the AMNH). Additional material was later referred to *Amphicoelias*, both in publications and internally in the museum's collections.

More recently, a second renumbering attempt seems to have been undertaken at AMNH, presumably based on the detailed historical work of McIntosh [41], who successfully identified several partial skeletons among the AMNH material from Garden Park. It is not known who conducted this renumbering, as this happened before current collection and curatorial staff took office, and no notes exist that could be used to identify those involved. To make matters even more complicated, these new numbers were solely noted on sheets of pink paper associated with the bones in the collection space and were not transferred onto the collection catalogue. Numbers used for the Garden Park material range from 30 001 to at least 30 014, and possibly higher (E.T., personal Observation, 2019). The renumbered specimens are mostly from the original catalogue numbers AMNH FARB 5760 and 5761, which were shown to include material from multiple individuals [41,42]. These original

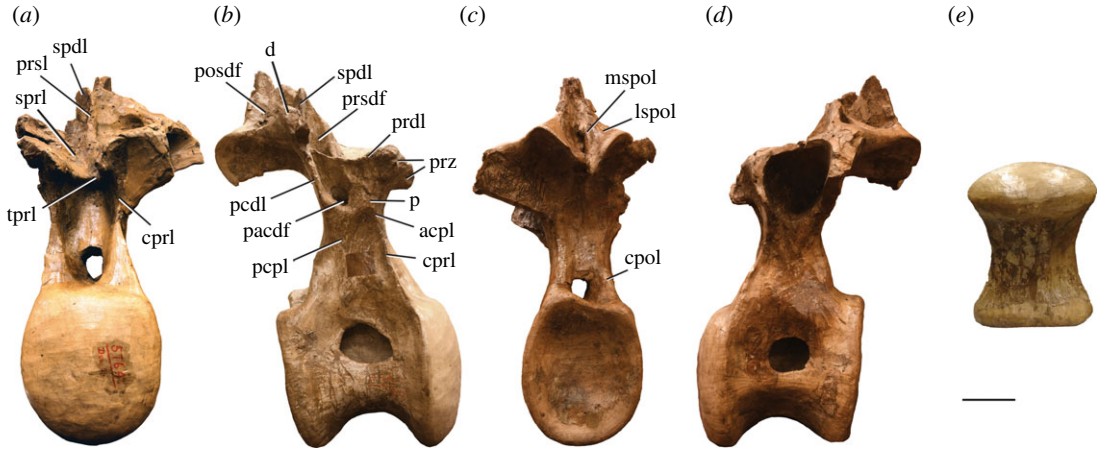

**Figure 4.** *Amphicoelias altus* holotype middle–posterior dorsal vertebra (AMNH FARB 5764) in (*a*) anterior, (*b*) right lateral, (*c*) posterior, (*d*) left lateral and (*e*) ventral (anterior surface towards top) views. acpl, anterior centroparapophyseal lamina; d, diapophysis; cpol, centropostzygapophseal lamina; cprl, centroprezygapophyseal lamina; lspol, lateral spinopostzygapophyseal lamina; mspol, medial spinopostzygapophyseal lamina; p, parapophysis; pacdf, parapophyseal centrodiapophyseal fossa; pcdl, posterior centrodiapophyseal lamina; pcpl, posterior centroparapophyseal lamina; posdf, postzygapophyseal spinodiapophyseal fossa; prdl, prezygodiapophyseal lamina; prsdf, prezygapophyseal spinodiapophyseal fossa; prsl, prespinal lamina; prz, prezygapophysis; spdl, spinodiapophyseal lamina; sprl, spinoprezygapophyseal lamina; tprl, interprezygapophyseal lamina. Scale bar, 100 mm.

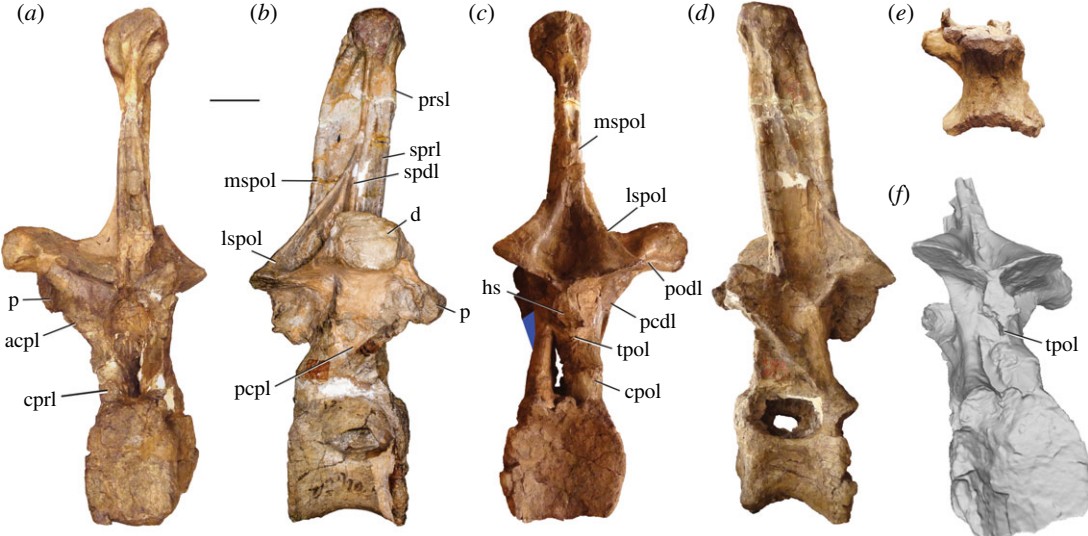

**Figure 5.** *Amphicoelias altus* holotype posterior dorsal vertebra (AMNH FARB 5764) in (*a*) anterior, (*b*) right lateral, (*c*) posterior, (*d*) left lateral, (*e*) ventral (anterior surface towards top) and (*f*) posterodorsal views. Blue polygon in *c* obscures hand used to brace the fragile vertebra for photography. acpl, anterior centroparapophyseal lamina; cpol, centropostzygapophyseal lamina; cprl, centroprezygapophyseal lamina; d, diapophysis; hs, hyposphene; lspol, lateral spinopostzygapophyseal lamina; mspol, medial spinopostzygapophyseal lamina; p, parapophysis; pcdl, posterior centrodiapophyseal lamina; pcpl, posterior centroparapophyseal lamina; podl, postzygodiapophyseal lamina; prsl, prespinal lamina; spdl, spinodiapophyseal lamina; sprl, spinoprezygapophyseal lamina; tpol, interpostzygapophyseal lamina. Scale bar, 100 mm.

numbers are the ones used in all scientific publications concerning the Cope specimens from Garden Park (e.g. [41,42]). The new numbers, instead, do not correspond to their entries in the AMNH collection catalogue, in which the numbers AMNH FARB 30 001–30 014 are listed as specimens of the turtles *Brachyopsemys tingitana* (holotype, AMNH FARB 30 001) and *Phosphatochelys* sp. (AMNH FARB 30 008), Testudines indet. (AMNH FARB 30 002 and 30 004–30 007), an indeterminate therapsid (AMNH FARB 30 003), and the holotype (AMNH FARB 30 009) and paratypes of the frog *Xenopus arabiensis* (AMNH FARB 30 010–30 014; C. Mehling, personal Communication, 2019, 2020). Among

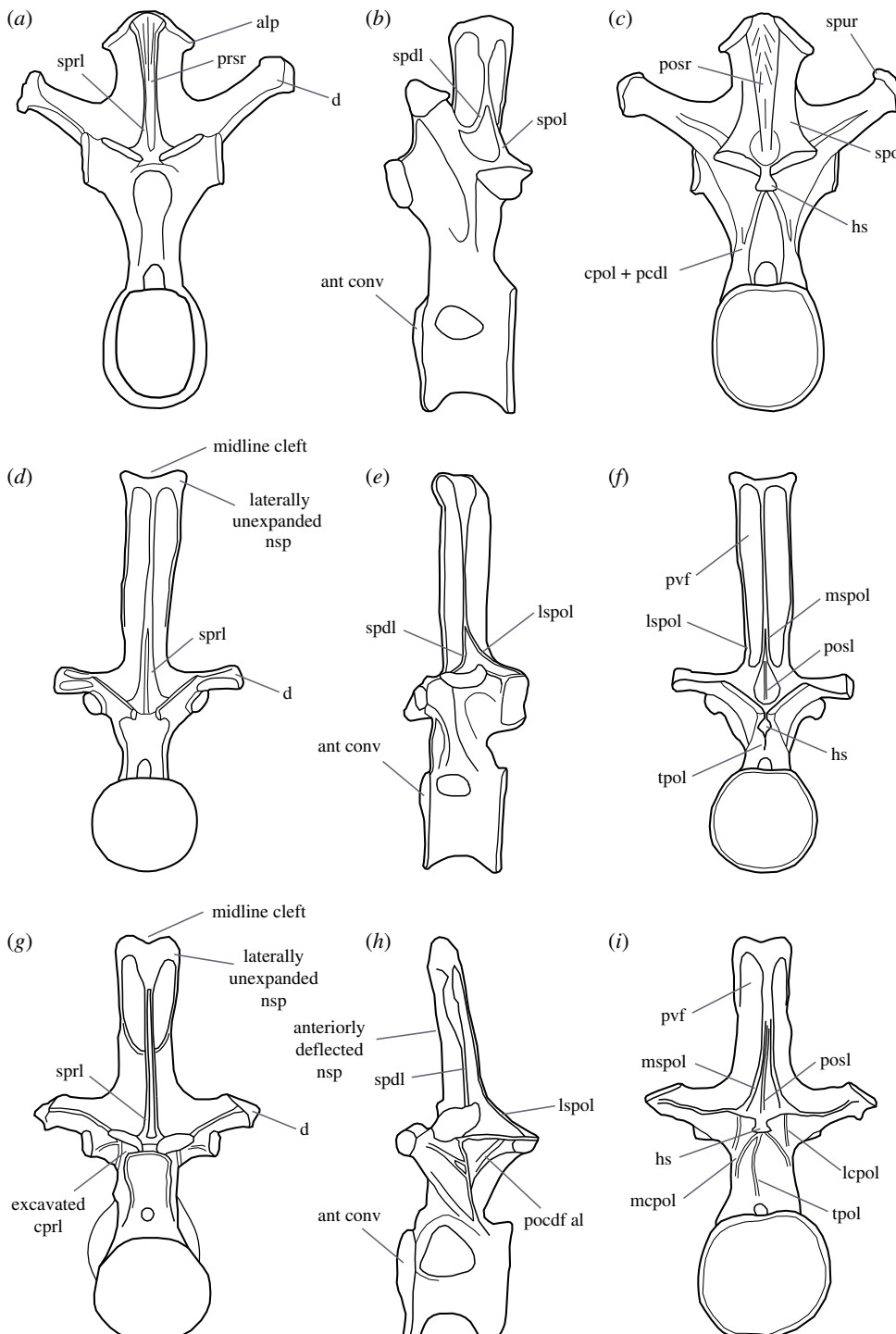

**Figure 6.** Comparative line drawings of posterior dorsal vertebrae of Morrison Formation diplodocoid sauropods occupying regions of the tree in which *A. altus* has been previously recovered and/or that have been proposed to be closely related: *Haplocanthosaurus priscus* (CM 572) in (*a*) anterior, (*b*) left lateral and (*c*) posterior views; *Apatosaurus louisae* (CM 3018) in (*d*) anterior, (*e*) left lateral and (*f*) posterior views; and *Diplodocus carnegii* (CM 84) in (*g*) anterior, (*h*) left lateral and (*i*) posterior views. Line drawings based on second or third most posterior dorsal vertebra as illustrated in Hatcher [30,55] and Gilmore [56]. al, accessory lamina; alp, aliform process; ant conv, anterior convexity; cpol, centropostzygapophyseal lamina; cprl, centroprezygapophyseal lamina; d, diapophysis; hs, hyposphene; lcpol, lateral centropostzygapophyseal lamina; lspol, lateral spinopostzygapophyseal lamina; mcpol, medial centropostzygapophyseal lamina; mspol, medial spinopostzygapophyseal lamina; nsp, neural spine; pcdl, posterior centrodiapophyseal lamina; pocdf, postzygocentrodiapophyseal fossa; posl, postspinal lamina; posr, postspinal rugosity; prsr, prespinal rugosity; pvf, posteroventrally opening fossa; spdl, spinodiapophyseal lamina; spol, spinopostzygapophyseal lamina; sprl, spinoprezygapophyseal lamina; tpol, interpostzygapophyseal lamina. Vertebrae not drawn to scale relative to one another.

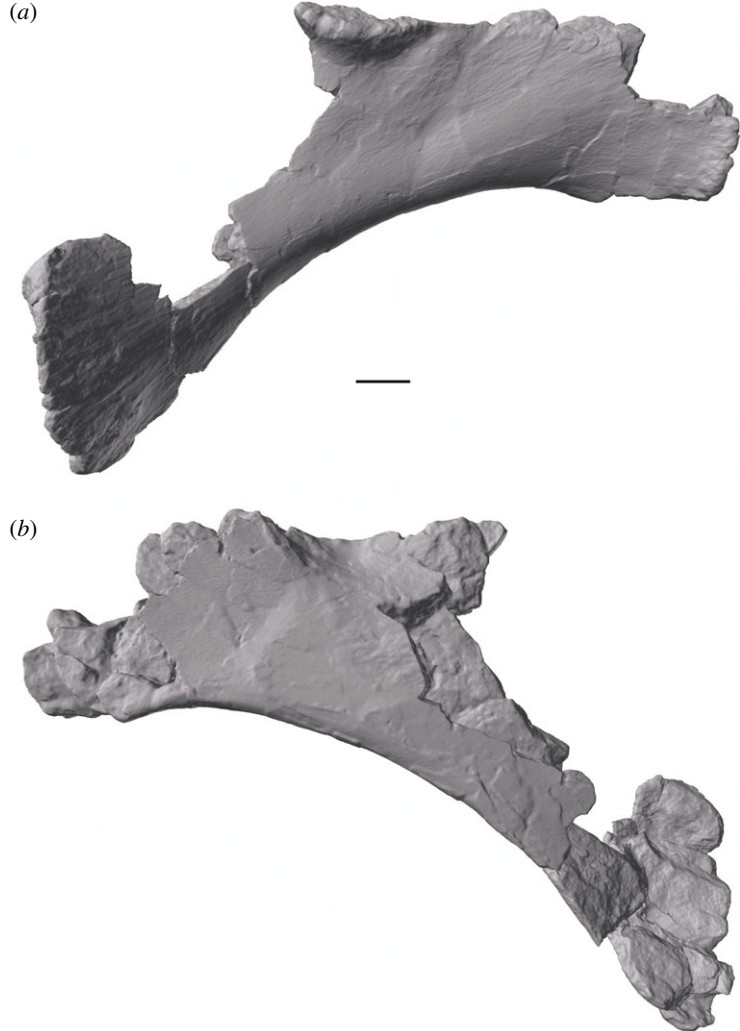

**Figure 7.** *Amphicoelias altus* holotype right pubis (AMNH FARB 5764) in (*a*) lateral and (*b*) medial views. Scale bar, 100 mm.

these erroneously recatalogued sauropod specimens are a humerus and pubis originally catalogued as AMNH FARB 5761, which are also associated with anonymous collection notes referring them to *A. altus*, as is a second pubis numbered AMNH FARB 5760, which was not renumbered. These elements correspond to the right humerus H.1 ([42]: fig. 89), and the pubes P.3 ([42]: fig. 105) and P.6 ([42]: fig. 106). Given that the renumbering has never been transferred to the official AMNH collection catalogue, that these numbers are registered as belonging to different species and specimens (including some type specimens) and that we are not aware of any scientific publication referring to these numbers, we ignore the new numbers between 30 001 and 30 014 associated with these bones in the collection space, and use only the original catalogue numbers, plus the specific bone identifiers proposed by Osborn & Mook [42] when necessary to refer to single bones or discuss affiliation to individual skeletons (e.g. H.1, P.3, P.6).

**Conservation history:** The two holotypic dorsal vertebrae of *A. altus* are incomplete, and parts of them are heavily restored (figure 1). Much of the restoration and repair likely stems from the original excavation in Colorado and preparation in Philadelphia, but the subsequent move from Philadelphia to New York, as well as further study (including ours), has resulted in additional damage. Mineralization of these fossil vertebrae also seems to be extensive, meaning that the bones are heavy, and manipulation easily results in damage to the delicate vertebral laminae and processes.

In an attempt to better identify reconstructed parts and repaired breaks, we used Progressive Photonics [61] to document the vertebrae in anterior and right lateral views (figure 3; electronic supplementary material). Progressive Photonics is a workflow to document an object under various

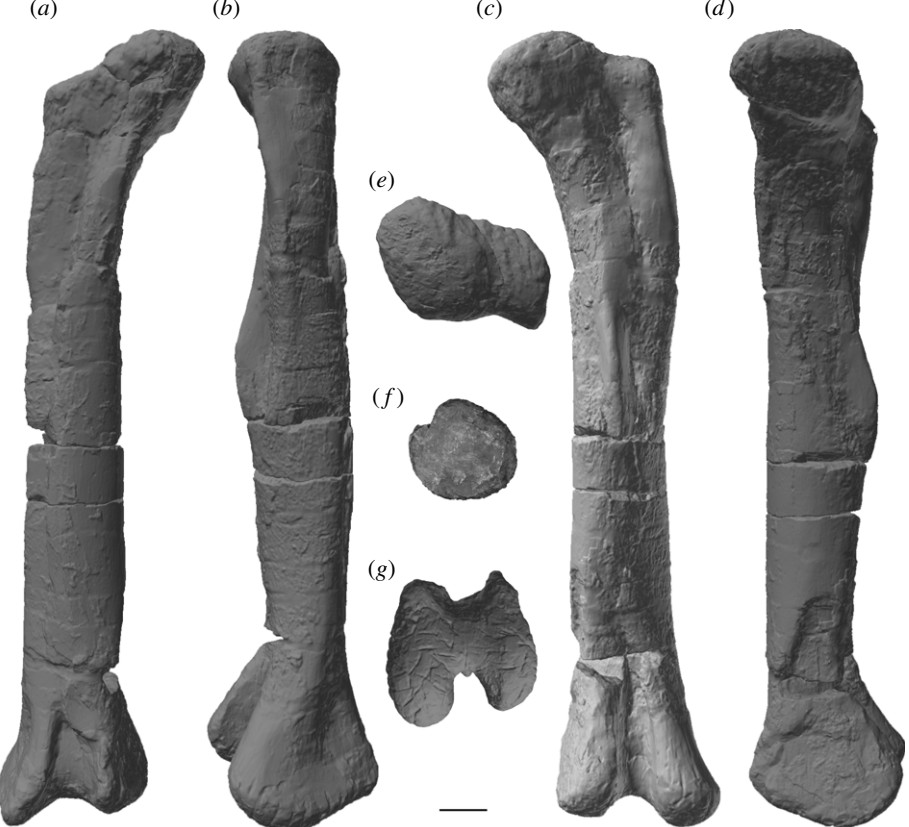

**Figure 8.** *Amphicoelias altus* holotype right femur (AMNH FARB 5764) in (*a*) anterior, (*b*) lateral, (*c*) posterior, (*d*) medial, (*e*) proximal, (*f*) midshaft, showing cross-sectional shape and (*g*) distal views. Anterior surface of femur towards the top of image in *e*–*g*. Scale bar, 100 mm.

lighting conditions and wavelengths, including frontal and oblique lighting, polarized light and UV stimulation [61]. UV stimulation, in particular, produces different reactions that can help to distinguish real fossil bone from various materials used in repair and restoration. Whereas distinct materials for repairs could be easily distinguished from each other, distinction between restored and real fossil bone was more difficult. At least three different adhesives were used to repair breaks, in some cases at the same fracture, indicating that the vertebrae broke several times at the same location (e.g. at the junction of the neural arch with the centrum in the posterior dorsal vertebra (figure 3)). Restoration of the vertebrae was likely performed early in their conservation history, given that restored parts and original bone react in a very similar way to UV stimulation, producing a bluish hue (figure 3). A similar bluish hue was identified as shellac coating in other historic fossil material from Wyoming [62], which was commonly used as a consolidant in the field and in fossil preparation until at least the 1930s [63]. This coating must have been applied over the entire restored vertebrae, thereby obscuring the distinction between original bone and the restored portions under UV stimulation. At the anterior-most portion of the right surface of the centrum of the posterior dorsal vertebra, the original shellac seems to have flaked off, revealing the actual, lighter colour of the bone, which does react slightly differently to UV stimulation (figure 3).

## 3. Description and comparisons of *Amphicoelias altus*

The four elements comprising the holotype of *A. altus* (AMNH FARB 5764) were scanned using an Artec Eva Structured Light scanner. The software Artec Studio (https://www.artec3d.com/) was used to align and clean the scans, as well as decimate them to a manageable size, while retaining high-resolution meshes. The individual fragments of the posterior dorsal vertebra, pubis and femur were scanned separately and later combined into a single file using the software Autodesk Meshmixer (https://www.meshmixer.com/). To more closely match the resolution of the other fragments of their

**Table 2.** Measurements of holotypic dorsal vertebrae of *A. altus* (AMNH FARB 5764). DvA, middle–posterior dorsal vertebra; DvB, posterior dorsal vertebra. Measurements in millimetres.

| dimension | DvA | DvB |
|---|---|---|
| anteroposterior length of centrum | ~220 | 242 |
| dorsoventral height of anterior end of centrum | — | 251 |
| dorsoventral height of posterior end of centrum | ~274 | 266 |
| mediolateral width of posterior end of centrum | ~231 | 258 |
| dorsoventral height of neural arch | ~239 | 212 |
| dorsoventral height of neural spine | — | 620 |
| anteroposterior length of neural spine (near base) | — | 158 |
| mediolateral width of neural spine (near base) | — | 72 |
| maximum mediolateral width of neural spine (near apex) | — | 148 |
| distance from midline to lateral tip of diapophysis | — | 303 |
| dorsoventral height at lateral tip of diapophysis | — | 111 |

respective bones, the distal end of the pubis and the middle section of the femur were further decimated. This decimation was performed in MeshLab, using the Quadric Edge Collapse Decimation filter [64]. Upon loading all the fragments of each bone into Meshmixer, each mesh was selected separately, and transformed manually in the virtual space to match up with the other meshes. Once the fragments were virtually reattached, all meshes were selected and united using the 'combine' command, with the settings set to keep the highest accuracy. The combined scans were then exported as an OBJ file. These scans are incorporated into our anatomical figures and the full models are available at Morphosource (https://www.morphosource.org/concern/biological_specimens/000346344?locale=en).

We make anatomical comparisons with eusauropods in general, but with particular focus on Morrison Formation diplodocoid taxa occupying regions of the tree in which *Amphicoelias* has been previously recovered and/or that have been proposed to be closely related. This includes the non-diplodocimorph diplodocoid *Haplocanthosaurus*, Apatosaurinae and *Diplodocus* (figure 6). Nomenclature for vertebral laminae and fossae follows Wilson [65] and Wilson *et al*. [66], respectively. The approximate serial positions of the two vertebrae are estimated based on comparisons with sauropods that preserve complete dorsal vertebral sequences, such as *Apatosaurus* (e.g. [56]), *Camarasaurus* (e.g. [42]) and *Diplodocus* (e.g. [30]). Although substantial portions of the vertebrae have undergone reconstruction, it is still possible to discern genuine morphology, even in places where the bone is entirely coated with consolidant and other materials.

## 3.1. Middle–posterior dorsal vertebra

A middle–posterior dorsal vertebra is preserved, lacking parts of the anterior and posterior surfaces of the centrum, as well as most of the neural spine (figures 1, 3 and 4; see table 2 for measurements). Despite being reconstructed with a bifid neural spine in Osborn & Mook [42], not enough of the neural spine is preserved to be able to determine its morphology.

The ventral surface of the centrum has been slightly crushed. There is a weak, rounded ridge close to the midline, although this does not form a distinct keel; otherwise, the ventral surface is fairly flat centrally, becoming gently transversely convex towards the lateral margins. There are no ventrolateral ridges or fossae. As reconstructed, each lateral pneumatic foramen has an elliptical outline, with its long axis oriented anteroposteriorly, although it is also fairly tall dorsoventrally. Although the margins of both are mostly reconstructed, this overall shape appears fairly accurate. Each foramen is situated on the dorsal half of the vertebra, occupying approximately half of the centrum length, with a slight anterior bias. The openings are deep, leaving a thin midline septum. Each foramen ramifies strongly dorsally and, especially, ventrally, as well as a small distance anteriorly and posteriorly. In this regard, *Amphicoelias* is similar to most neosauropods, with the notable exceptions of dicraeosaurids and some titanosaurs, in which the lateral excavations are shallow [67]. There are no

vertical ridges inside the foramen, contrasting with the condition in several diplodocids [51], including specimens of *Apatosaurus*, *Barosaurus* and *Diplodocus* [5]. However, there is a subhorizontal ridge at the anteroventral corner of the foramen, which is not visible in lateral view: this is present on both sides, although is better developed on the right side.

The sharp lateral margin of the centroprezygapophyseal lamina (CPRL) merges with the anterior centroparapophyseal lamina (ACPL). A posterior centroparapophyseal lamina (PCPL) is also present, although this only becomes visible towards the parapophysis, situated on the lateral surface of the prezygapophysis. Posteroventrally, it appears to merge with the posterior centrodiapophyseal lamina (PCDL). The latter lamina is near-vertical and forms the posterior margin to a deep parapophyseal centrodiapophyseal fossa (PACDF) on the lateral surface of the upper part of the neural arch, bounded anteroventrally by the PCPL and dorsally by the gently posterodorsally directed prezygodiapophyseal lamina (PRDL).

Although this region is heavily reconstructed, the anterior neural canal opening is clearly set within a fossa, as in most eusauropods [68]. The prezygapophyseal articular surfaces are gently convex mediolaterally, and there are well-developed hypantral surfaces. A weakly developed, horizontal interprezygapophyseal lamina (TPRL) is strongly U-shaped in dorsal view. A TPRL separates the prezygapophyses in most eusauropods, whereas their articular surfaces are confluent in rebbachisaurids [69,70]. The postzygapophyseal articular surfaces are gently concave. Although there is no hyposphene, its absence is almost certainly a preservational artefact, based on the ventral surface of the postzygapophyseal midline being clearly damaged. The poor preservation in this region means that we also cannot determine whether a vertical midline lamina (interpostzygapophyseal lamina (TPOL)) extended between the posterior neural canal opening and the postzygapophyseal complex.

The partially preserved left diapophysis has been broken and slightly deformed. As reconstructed, it projects primarily laterally, with a subtle dorsal deflection, which is consistent with the other, better preserved, dorsal vertebra. The spinoprezygapophyseal laminae (SPRLs) extend from close to the medial margins of the posterior ends of the prezygapophyses and merge dorsally, forming a narrow, anteriorly projecting prespinal lamina (PRSL). Ventral to their junction, there is no evidence for a PRSL. The base of the subvertical spinodiapophyseal lamina (SPDL) is preserved, and fully extends down to the diapophysis. A prezygapophyseal spinodiapophyseal fossa (PRSDF) is formed anterior to the SPDL, bounded ventrally by the PRDL, and anterodorsally by the SPRL. The spinopostzygapophyseal lamina (SPOL) is bifurcated a short distance above the postzygapophyses, with a large spinopostzygapophyseal lamina fossa (SPOL-F) in between the medial (mSPOL) and lateral (lSPOL) branches. A bifid SPOL is common across an array of eusauropod lineages [47,71], although *Amphicoelias* differs from some rebbachisaurids in which the SPOL is divided throughout its entire length [49,50]. The mSPOL is directed dorsomedially along its preserved basal portion, with no midline postspinal lamina (POSL) within the postspinal fossa. As such, the dorsal vertebrae of *Amphicoelias* differ from those of most diplodocimorphs (including *Apatosaurus* and *Diplodocus*), in which pre- and/or postspinal laminae are often present throughout nearly the full length of the neural spine (figure 6), and are not always formed solely by convergence of the SPRLs or SPOLs [49,50,65]. In this regard, the dorsal neural spines of *Amphicoelias* have a similar laminae configuration to that of *Haplocanthosaurus priscus* [55], although the SPOL is non-bifid in that species (figure 6). The lSPOL and SPDL merge a short distance from the dorsalmost tip of the preserved neural spine in *Amphicoelias*. A postzygapophyseal spinodiapophyseal fossa (POSDF) is formed by the SPDL, the subhorizontal postzygodiapophyseal lamina (PODL), and the anterodorsally directed lSPOL.

## 3.2. Posterior dorsal vertebra

A relatively complete dorsal vertebra, missing the left diapophysis, is preserved (figures 1, 3 and 5; see table 2 for measurements). We interpret its position as slightly posterior in the vertebral series relative to the aforementioned vertebra. Breakages do not indicate any clear signs of internal camellae, contrasting with the tissue structure of the presacral vertebrae of titanosauriforms [47].

The anterior surface of the centrum is incomplete along its left side, and the posterior surface along its right side. Despite this, we can extrapolate that the complete centrum would be slightly dorsoventrally taller than wide. The anterior articular surface of the centrum is irregular, but it is predominantly flat. Although it does not form a distinct condyle, a dorsally restricted convexity is present, which results in a slightly opisthocoelous centrum, as is the case in many diplodocoids, including *Apatosaurus*, *Diplodocus* and *Haplocanthosaurus* [5,68] (figure 6). By contrast, the posterior articular surface of the centrum is deeply concave. The ventral surface of the centrum is anteroposteriorly concave and gently

convex transversely, but it flattens out along its midline. There are no ventrolateral or ventral midline ridges, nor are there any ventral fossae or excavations.

The lateral pneumatic foramen is a deep structure that leaves a thin midline septum. It is restricted to the dorsal third of the centrum and ramifies dorsally and ventrally. It is not possible to determine whether there were internal ridges. The morphology of the lateral pneumatic foramen appears to be very different on either side. On the left side, it is a large opening that extends for most of the centrum length; however, all but the ventral margin of this opening is damaged. On the right side, the foramen is much shorter anteroposteriorly (with a strong anterior bias) and is set within a shallow, circular fossa. A fossa is clearly absent on the left side. The lateral surface of the centrum, ventral to the pneumatic foramen, is gently concave anteroposteriorly and fairly flat dorsoventrally (although this surface has been slightly broken, displaced and deformed on the left side).

The neural arch pedicles extend for the full anteroposterior length of the centrum. The anterior neural canal opening is set within a dorsoventrally tall fossa. At the base of the neural arch, each CPRL merges with the ACPL and forms a sharp ridge that is directed primarily vertically. At approximately one-third of its length, this combined CPRL–ACPL divides into a widely rounded, but distinct, CPRL that extends dorsomedially to contact the ventral margin of the hypantrum, and a narrow, sharp ACPL that connects to the ventral margin of the parapophysis. The CPRL is neither bifurcated, nor is its anterior surface excavated, contrasting with the morphology of several diplodocids (including *Apatosaurus*, *Diplodocus* and *Supersaurus*) and rebbachisaurids [50,68,72]. As is the case in the middle–posterior dorsal vertebrae of most non-titanosauriform sauropods [51,73], there is no evidence for an anterior centrodiapophyseal lamina (ACDL) on either side, and this appears to be a genuine absence. A poorly preserved remnant of the subvertical PCDL is present on the right side, with no evidence for ventral expansion or bifurcation.

As reconstructed, the PCPL is directed anterodorsally at an angle of approximately 45° to the horizontal. However, this is illustrated at a slightly shallower angle in Osborn & Mook [42], which is presumably more accurate, likely reflecting the result of several generations of repair at the junction of the centrum and neural arch (see 'Conservation history'). The PCPL meets the ACPL at the base of the parapophysis, which is situated on the lateral surface of the prezygapophysis, but it does not extend further dorsally than the latter process. The mediolaterally elongate prezygapophyseal articular surfaces are flat and face dorsomedially. They are oriented at a steep angle (approx. 30–35° to the horizontal), but this might have been accentuated by crushing. No TPRL is preserved, but the area in between the prezygapophyses has clearly been damaged.

CPOLs appear to be largely reconstructed along their ventral portions, but extend mainly vertically, forming the lateral margins of the posterior neural canal opening. The CPOL is conjoined with the PCDL along this section, as is the case in nearly all diplodocoids, including both species of *Haplocanthosaurus* [5,7,74] (figure 6). The dorsal portion of the CPOL projects posterodorsally in lateral view, although this might have been distorted, and connects to the anterior end of the ventral surface of the postzygapophysis. Unlike the condition in flagellicaudatans [47,49] (figure 6), as well as a small number of other eusauropod taxa [50], the dorsal portions of the CPOLs are not divided into medial and lateral branches. The postzygapophyseal articular surfaces are flat and face ventrolaterally. At their ventral midline, the prominent hyposphene has a triangular shape in the posterior view, with the apex of this triangle pointing dorsally. A vertical ridge extends from the midline of the ventral surface of the hyposphene down towards the posterior neural canal opening. A distinct midline ridge, generally defined as a TPOL, is present in this region of the middle–posterior dorsal vertebrae of several diplodocids (figure 6), including *Apatosaurus* and *Diplodocus* [13,67], some rebbachisaurids, as well as a small number of taxa outside of Diplodocoidea (e.g. *Tehuelchesaurus*) [5,68]. A shallow centropostzygapophyseal fossa (CPOF) is present on the posterolateral surface of the neural arch, bounded anterodorsally by the CPOL, and medially by the postzygapophysis and hyposphene. There is no evidence for an accessory lamina within the shallow postzygapophyseal centrodiapophyseal fossa (POCDF). The presence of this lamina appears to characterize the middle–posterior dorsal vertebrae of most diplodocines [5,51] (figure 6).

Only the right diapophysis is preserved, and it projects mainly laterally, with a slight dorsal deflection. As a result, it differs from the dorsally deflected diapophyses of the middle–posterior dorsal vertebrae of dicraeosaurids, rebbachisaurids and *Haplocanthosaurus* (figure 6) [49,67]. It is mediolaterally short and, in general, robust, contrasting with the elongate diapophyses of brachiosaurids [75] and rebbachisaurids [68]. No spur projects from the diapophysis, such as that present in *Haplocanthosaurus* (figure 6) and *Brontosaurus parvus* [5], and its distal end lacks the distinct dorsal surface that characterizes

most somphospondylans [13]. Laterally, the diapophysis expands ventrally, and it has an approximately semicircular shape in lateral view, with a flat dorsal margin. The PRDL is not well preserved, and the PODL is partly reconstructed on the right side, but is a prominent, subhorizontal structure.

The neural spine projects almost entirely dorsally, with perhaps a very slight anterior deflection. However, the latter is mainly because the neural spine subtly decreases in anteroposterior length dorsally, with the posterior margin sloping slightly to face a little dorsally, as well as posteriorly. Regardless, it clearly differs from the strongly anteriorly inclined posterior dorsal neural spines of *Diplodocus* (e.g. [30]) (figure 6). In general, the anterior and posterior margins of the neural spine are subparallel, lacking the subtriangular outline that characterizes many titanosauriforms in lateral view [71].

SPRLs form the anterolateral margins of the neural spine, converging dorsally at approximately two-thirds of the spine height. Ventral to this convergence, the anterior surface of the neural spine in between the SPRLs is infilled with a gently rugose surface, but there is no distinct PRSL. Although heavily reconstructed, there is an SPDL on the right side. This lamina meets and merges with the steep, anterodorsally directed lSPOL at approximately the spine midheight, from which point the combined lateral lamina continues subvertically to the apex of the neural spine. The base of the lSPOL and the incomplete upper part of the combined lateral lamina are also preserved on the left side. On the right side, the POSDF hosts a prominent bony shelf, which arises from a point approximately equidistant between the diapophysis and the postzygapophysis. This shelf is subtriangular in lateral view, with its external surface facing anterolaterally and slightly dorsally. It contacts the lSPOL a short distance dorsal to the postzygapophysis. In lateral view, this structure gives the appearance of creating two deep subfossae within the POSDF, either side of the shelf, but these are actually connected with one other ventral to the shelf. At about midheight of the shelf, a subhorizontal ridge is present anteromedial to the shelf; this ridge also roofs the anterior 'subfossa'. Each mSPOL is subvertical for most of its length, and only converges with its counterpart close to the neural spine apex. No POSL is present ventral to this convergence. Unlike diplodocids [7,51,74] (figure 6), there are no posteroventrally opening fossae (pockets) on the posterior surface of the neural spine, close to its apex.

The dorsal surface of the neural spine is flat, lacking the subtle bifurcation (midline cleft) that often persists into all but the posteriormost dorsal vertebrae of many diplodocids [5,51]. For example, this cleft is present up to, and including, dorsal vertebra 8 (of 9) in *Barosaurus lentus* [76], and dorsal vertebra 9 (of 10) in *Apatosaurus louisae* [56] and *Diplodocus carnegii* [30] (figure 6). By contrast, no cleft is present in the posterior dorsal neural spines of *Supersaurus vivianae* [15].

The apex of the posterior dorsal neural spine of *Amphicoelias* is unusual. Although it becomes transversely expanded dorsally, as in most other eusauropods, the nature of this expansion does not appear to be homologous to that of other taxa. In non-diplodocimorph taxa, including *Haplocanthosaurus* (figure 6), lateral flaring of the neural spine results from the projection of triangular aliform processes [67,77] that are formed primarily from the expansion of the SPOLs [71]. The posterior dorsal neural spines of dicraeosaurids and rebbachisaurids are 'petal'-shaped in anterior/posterior view, expanding transversely through approximately 75% of their length, and then tapering dorsally [49,67,78]. By contrast, the neural spine of *Amphicoelias* expands only close to its apex, and does not taper, other than having a convex dorsal margin in anterior/posterior view. In lateral view, the expansion has a subtriangular outline, with the apex of this triangle pointing ventrally. As such, the posterior neural spine morphology of *Amphicoelias* seems to differ from all other sauropods, including diplodocids (figure 6), which tend to lack a lateral expansion altogether [13,49]. We regard this morphology as an autapomorphy of *A. altus*.

## 3.3. Pubis

The pubis is incomplete and preserved in two pieces (figure 7). The missing portions include the acetabular margin, the ambiens process, the obturator foramen and the complete articular surface for the ischium, which makes it difficult to determine whether this is a left or right pubis. However, we interpret the pubis as being from the right side, consistent with the holotypic femur, and in contrast with the interpretation of Osborn & Mook [42]. Assuming that our interpretation is correct, the internal (dorsomedial in articulation) surface of the pubis is proximodistally concave and mostly flat transversely. The proximal facet for the articulation with the ilium is relatively flat, with rugose margins. The articular facet for the left pubis is at about midlength, indicating that the ischial articular surface did not extend this far distally.

**Table 3.** Measurements of holotypic femur of *A. altus* (AMNH FARB 5764). Measurements in millimetres.

| dimension | measurement |
| --- | --- |
| proximodistal length | ~1700 |
| distance from proximal end to distal tip of fourth trochanter | ~900 |
| maximum diameter of shaft at distal end of proximal half | 233 |
| diameter of shaft at distal end of proximal half, perpendicular to maximum diameter | 222 |
| midshaft mediolateral width | 219 |
| midshaft anteroposterior length | 238 |
| mediolateral width of shaft at two-thirds of femur length (from proximal end) | 228 |
| anteroposterior length of shaft at two-thirds of femur length (from proximal end) | 211 |
| minimum shaft circumference | 701 |
| distal end anteroposterior length | ~354 |
| mediolateral width of tibial condyle | 166 |
| mediolateral width of fibular condyle | ~187 |

## 3.4. Femur

The right femur is currently in three pieces (and was clearly broken further prior to preparation) but appears to be largely complete in terms of length (figures 2 and 8; see table 3 for measurements). However, there is material missing along the three pieces, with lots of reconstruction in places. With the exception of the anterior portion of the distal condyles, the distal tip of the femur is entirely reconstructed (figure 2; see [42]). The femur has also undergone transverse compression, particularly along the distal half, and there also appears to have been some anti-clockwise twisting of the shaft relative to the proximal and distal ends.

The femoral head projects mainly medially, and slightly dorsally. It is notably raised relative to the proximal articular surface of the greater trochanter. Although there is a weakly developed lateral bulge, the lateral margin of the proximal third of the femur is not medially deflected relative to the lateral margin of the femoral midshaft, contrasting with the femora of many macronarians [71,79]. The posterior surface of the proximal end, towards the lateral margin, forms a concavity, giving the initial impression of a trochanteric shelf, such as that seen in several rebbachisaurids [49] and many somphospondylans [71]. However, this concavity seems to be merely the product of crushing: slightly more distally, the anteroposterior thickness of the lateral margin is not notably thinner than the rest of the femur.

There is no proximodistally elongate midline ridge (linea intermuscularis cranialis) on the anterior surface of the shaft, such as that seen in some derived titanosaurs [80]. A large nutrient foramen, such as that observed on the anterior surface of the femora of *Dicraeosaurus* [47] and some diplodocid individuals [5], is also absent. Although heavily restored, the fourth trochanter is clearly a prominent structure, contrasting with the femora of some rebbachisaurids, in which this process is barely discernible [51]. It has been crushed and slightly displaced laterally, but this likely relates to the taphonomic twisting of the shaft noted earlier. However, even accounting for deformation, it seems highly unlikely that the fourth trochanter could have been visible in anterior view. This differs from the condition in many 'basal' macronarians (including *Brachiosaurus* and *Camarasaurus*), as well as both species of *Haplocanthosaurus*, in which the fourth trochanter is medially deflected [49,50,71,81]. The distal tip of the fourth trochanter extends distal to the femoral midlength.

As noted by Cope [9], the femoral midshaft is characterized by a subcircular cross-section. Although this might have been accentuated by crushing, the ratio of the mediolateral to anteroposterior diameter of the femoral shaft is fairly consistent, ranging from 0.9 (at midshaft) to 1.1. Consequently, we regard a subcircular cross-section to be a genuine feature of the femur of *Amphicoelias*. Most sauropod femora have transversely expanded, elliptical midshaft cross-sections, with ratios of the mediolateral to anteroposterior diameters exceeding 1.4, including specimens of the Morrison Formation sauropods *Apatosaurus*, *Barosaurus*, *Brachiosaurus*, *Brontosaurus*, *Camarasaurus*, *Diplodocus*, *Galeamopus*, and *Haplocanthosaurus* [5,16,47,71,82]. Although several diplodocoids have lower ratios than macronarians

(e.g. *Tornieria* = 1.3; [5]), these do not approach the subcircular morphology that we, therefore, regard as autapomorphic for the femur of *A. altus*, as previously proposed by Wilson & Smith [60].

The femur has an anteroposteriorly thick shaft relative to that of the distal end (ratio > 0.6) [49]. This is comparable to some 'basal' macronarians (e.g. *Camarasaurus*), several rebbachisaurids and the dicraeosaurid *Amargasaurus* [5,49,50]. Most other sauropods, including the majority of diplodocids for which this can be assessed (e.g. *Apatosaurus*, *Diplodocus*), have much lower ratios, although some femora of *Brontosaurus* (e.g. YPM VP.001980) are comparable to those of *Amphicoelias*. Based on the preserved anterior portion, the femur is bevelled at its distal end, with the fibular condyle extending further distally than the tibial condyle. This type of bevelling is generally restricted to derived titanosaurs [47], with most other sauropods characterized by either a more distally extensive tibial condyle (e.g. *Diplodocus*) or a distal end that is perpendicular to the long axis of the femoral shaft (e.g. *Apatosaurus*) [71]. Within Diplodocoidea, *Haplocanthosaurus delfsi* is the only other species in which the fibular condyle extends further distally than its tibial counterpart ([34]: fig. 16; CMNH 10 380: P.D.M., personal observation, 2008) [50]. Given that the morphology of the femur of *Haplocanthosaurus delfsi* otherwise differs notably from that of *Amphicoelias*, we regard the distal bevelling observed in the femur of *A. altus* as a local autapomorphy within Diplodocoidea. The distal articular surface is anteroposteriorly convex, but it does not curve notably onto the anterior surface of the femur. Poor preservation means that it is not possible to determine whether the fibular condyle is posteriorly divided.

## 3.5. Ontogenetic stage

It was not possible to get permission from the AMNH to sample the *A. altus* type material for histological analysis. As such, our interpretation of the ontogenetic stage of this individual is restricted to external anatomical indicators and, therefore, necessarily coarse. Although the identification of ontogenetic stage based on size alone is problematic (e.g. [83–86]), the length of the femur of *A. altus* (approx. 1700 mm) is comparable to that of many of the largest sauropods (including other Morrison Formation species), with only some titanosauriforms notably exceeding this length [87,88]. This suggests that this individual is likely to represent an adult. Only one size-independent feature is available to us, which is that the neurocentral synostoses of the dorsal vertebrae are fully fused. In skeletally immature archosaur individuals, these are often only partially fused or fully open [84,89]. Although full fusion of neurocentral sutures can occur in some immature archosaur individuals [90], and the timing of fusion can vary along the vertebral column [85,86], the condition in the dorsal vertebrae of *Amphicoelias* is consistent with our interpretation that it represents an adult individual, based on the size of its femur. There are no available indicators to determine whether this was a young adult, potentially still growing, or if it was an old individual, although there are no features associated with advanced age (e.g. heterotopic ossification; [91]).

# 4. Description and affinities of material previously referred to *Amphicoelias*

## 4.1. AMNH FARB 5764 and 5764a

Cope [9] included only the two dorsal vertebra, pubis and femur described above when erecting *A. altus*. Later, a tooth was accessioned under the type number (AMNH FARB 5764), and a scapula, coracoid, ulna and distal half of a femur were accessioned under AMNH FARB 5764a, and all identified as *A. altus* in the AMNH collections. Although these specimens are thought to have also been collected in the Cañon City area, they were not mentioned in Cope's original description, and there is no evidence that any of them came from the type locality [2,41,42], with information on provenance lost for most of the specimens [41]. Nevertheless, the postcranial remains were provisionally referred to *A. altus* by Osborn & Mook [42]. The referral of the tooth was rejected because of its lack of resemblance to the teeth of *Diplodocus*, then thought to be a close relative of *Amphicoelias* [42]. The arbitrary assignment of the pectoral and forelimb elements was based on their apparent dissimilarity to those of *Camarasaurus*, and general resemblance to *Diplodocus*. By contrast, the distal femur, identified as a left, was considered to agree so closely with the type specimen that Osborn & Mook [42] regarded it as 'possible…that it is the mate to the type femur of *A. altus*' (p. 385). However, McIntosh [41] determined that this femur came from the nearby Cope Quarry IV instead.

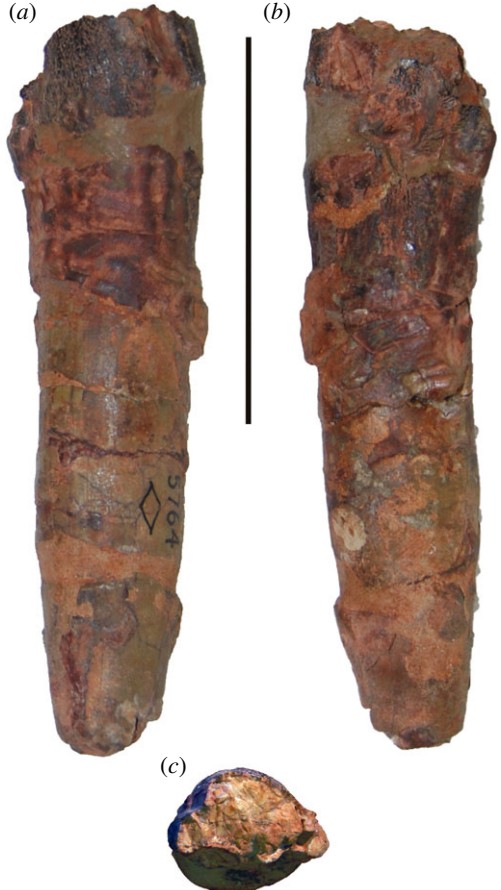

**Figure 9.** AMNH FARB 5764, tooth, in (*a*) labial, (*b*) lingual and (*c*) occlusal views. Scale bar, 50 mm.

Whitlock [49] only excluded the tooth in his *A. altus* operational taxonomic unit (OTU), whereas Mannion *et al.* [50,51] excluded all four of these referred elements. Tschopp *et al.* [5] noted that the tooth, scapula and coracoid all more closely resemble *Camarasaurus* than a diplodocoid (see also [2]). They followed Mannion *et al.* [51] in excluding these three elements but ran two sets of preliminary phylogenetic analysis with and without the ulna. Its inclusion had no effect on the topological placement of *A. altus*, although Tschopp *et al.* [5] excluded it from their final analyses. To avoid potentially creating a chimera, we recommend the exclusion of the tooth, scapula, coracoid and ulna from *A. altus*. However, we re-describe and figure all of these previously referred elements here (figures 9–12), re-evaluating their taxonomic affinities (see table 4 for measurements). Both the scapula and coracoid are described with the long axis of the scapular blade orientated horizontally.

The tooth (AMNH FARB 5764) consists of most of the root, as well as a small portion of the poorly preserved base of the crown (figure 9). The root is straight, with a cylindrical cross-section that mesiodistally narrows towards its base. Its external surface is smooth. At its broken base, the crown has a D-shaped cross-section and is only slightly mesiodistally expanded relative to the root. The basal portion of the crown is characterized by a mesiodistally convex labial surface with some evidence for weakly developed labial grooves. By contrast, the lingual surface is generally flat, although there appears to be a midline lingual ridge. Both surfaces are characterized by fine, anastomosing wrinkled enamel. The basal margin of this enamel is not perpendicular to the long axis of the root. Taken as a whole, this character complex is closest to that of the teeth of non-titanosauriform macronarians [47,67,71], especially *Camarasaurus* (e.g. [93,94]). However, we note that we do not currently know the morphology of the teeth of the earliest diverging diplodocoids: it is possible that the teeth of *A. altus* and taxa such as *Haplocanthosaurus* most closely resemble those of 'basal' macronarians, with the development of cylindrical teeth a synapomorphy of Diplodocimorpha, rather than Diplodocoidea. Consequently, we identify the AMNH FARB 5764 tooth as aff. *Camarasaurus*, but the possibility remains that it is attributable to *A. altus*.

The scapula (AMNH FARB 5764a [Sc. 7]) is preserved in four pieces (figure 10). It lacks part of the acromion, the dorsal edge of most of the distal blade and part of the end of the blade. The short acromial

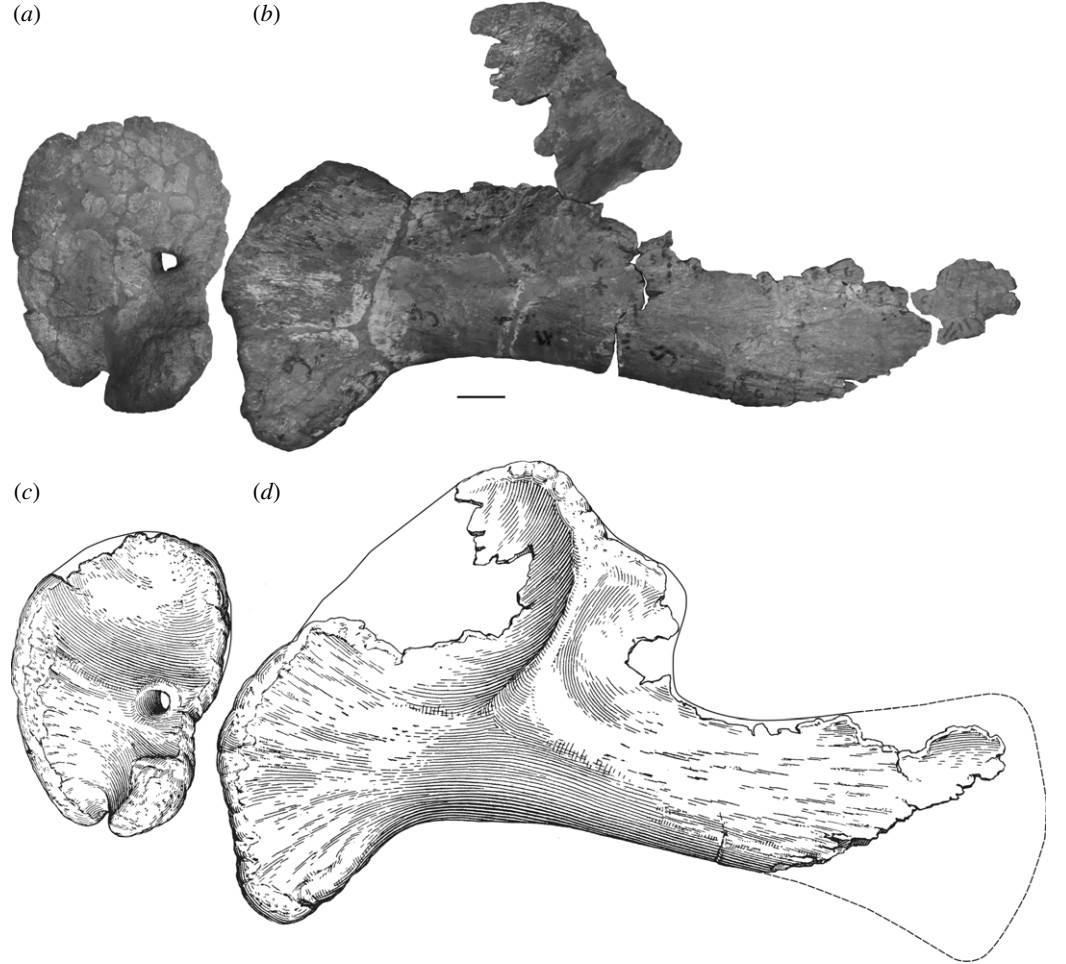

**Figure 10.** AMNH FARB 5764a [Cor. 3], left coracoid (*a,c*), and AMNH FARB 5764a [Sc. 7], left scapula (*b,d*), in lateral view. Scale bar, 100 mm. (*c,d*) Modified from Osborn & Mook [42].

ridge is at an obtuse angle to the long axis of the distal blade. Anterior to this ridge, the acromion expands dorsoventrally. The preserved dorsal margin of the acromion, therefore, has a somewhat sinuous shape in lateral view. This is an unusual morphology in sauropods, although a scapula assigned to *Camarasaurus lentus* has a broadly similar profile ([95]: fig. 6.10A). The glenoid is strongly expanded mediolaterally, and slightly medially bevelled. The latter feature is primarily restricted to somphospondylans, but it has also been noted in the scapula of some apatosaurines [5,14,47]; however, the bevelling is more clearly developed in most of those taxa. There is no subtriangular process on the ventral margin of the posteroventral corner of the acromion. The presence of this process is primarily a titanosauriform feature [68,71,75], but it is variably present in specimens of some diplodocids (e.g. *Diplodocus hallorum*; [5]). There is a subtle rugosity on the medial surface of the blade, close to the acromion. The proximal part of the scapular blade is D-shaped in cross-section, with a flat medial surface and dorsoventrally convex lateral surface, as is the case in most eusauropods [47]. This lateral convexity results in a distinct longitudinal crest, such as that seen in several neosauropods, including *Apatosaurus* and *Camarasaurus* [96]. Although the distal-most portion of the blade is not preserved, a small piece indicates that the blade expanded dorsally at its distal end. No expansion is indicated along the ventral margin, as preserved; however, this usually occurs only towards the very distal end of the scapula (e.g. [5]). Although the subtle bevelling of the glenoid could indicate referral to an apatosaurine, the unusual morphology of the acromion is more in keeping with attribution to *Camarasaurus*. As such, we conservatively regard the AMNH FARB 5764a scapula as belonging to an indeterminate neosauropod.

The coracoid (AMNH FARB 5764a [Cor. 3]) is nearly complete, lacking only a small portion of its rounded anterodorsal margin (figure 10). Based on its size and seemingly good fit (figure 10), it is possible that it is from the same individual as the scapula [42]. The coracoid is dorsoventrally taller

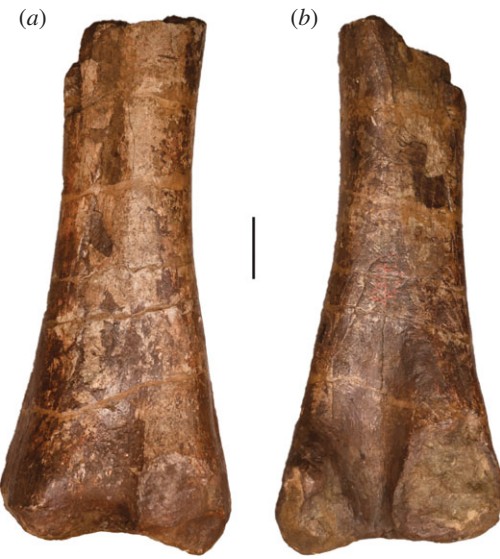

**Figure 11.** AMNH FARB 5764a [Ul. 1], left ulna, in (*a*) lateral, (*b*) anterior, (*c*) medial, (*d*) posterior, (*e*) proximal and (*f*) distal views. Scale bar, 100 mm. (*c, e* and *f*) Modified from Osborn & Mook [42].

**Figure 12.** AMNH FARB 5764a [Fem. 2], distal femur, in (*a*) anterior and (*b*) posterior views. Scale bar, 100 mm.

than anteroposteriorly long, lacking the 'squared' morphology that characterizes apatosaurines [5,67]. It has a straight articular surface for the scapula. The glenoid is strongly expanded mediolaterally, which differs from most diplodocids [97]. As is the case in most neosauropods [73], the articular surface of the glenoid extends onto the lateral surface of the coracoid. Where preserved, this characterizes all Morrison Formation sauropods, with the exception of *Haplocanthosaurus priscus* [50]. The glenoid is anteroventrally bound by a distinct, anteroposteriorly short, U-shaped notch. A similar condition characterizes many eusauropods, including *Camarasaurus* and *Haplocanthosaurus priscus*, whereas this groove is shallow in *Brachiosaurus* and most diplodocimorphs, including Morrison Formation members of this clade [5,50,68,71]. Anterior to the notch in AMNH FARB 5764a, at the ventral end of the anterior margin, there is some broken bone surface indicating the possible presence of a glenoid lip. The coracoid foramen is situated slightly dorsal to the glenoid, extending

**Table 4.** Measurements of specimens previously referred to *A. altus*: AMNH FARB 5764 (tooth), 5764a (scapula, coracoid, ulna, femur) and 5761 (humerus). Measurements in millimetres.

| element | dimension | measurement |
|---------|-----------|-------------|
| tooth | base of crown maximum mesiodistal width | 25 |
| | base of crown maximum labiolingual width | 18 |
| scapula | anteroposterior length as preserved | 1450 |
| | dorsoventral height of acromion | 940 |
| | minimum dorsoventral height of scapular blade | 270 |
| coracoid | maximum dorsoventral height | 710 |
| | maximum anteroposterior length | 450 |
| humerus | proximodistal length | 1140 |
| | maximum mediolateral width of proximal end | ~510 |
| | mediolateral width at midshaft | 215 |
| | anteroposterior diameter at midshaft | 125 |
| | mediolateral width at distal end | 370 |
| ulna | proximodistal length | 1035 |
| | mediolateral width of proximal end | 265 |
| femur | proximodistal length as preserved | 840 |
| | mediolateral width at midshaft | 190 |
| | anteroposterior diameter at midshaft | 160 |
| | mediolateral width of distal end | 360 |

dorsomedially and a little posteriorly through the coracoid. Based on the features outlined above, we identify the AMNH FARB 5764a coracoid as aff. *Camarasaurus*. This would also be broadly consistent with our interpretation of the scapula if the two elements are from the same individual.

The ulna (AMNH FARB 5764a [Ul.1]) is preserved in one piece (figure 11). Its proximal and especially distal articular surfaces are partly eroded, but still indicate their general, original shape. The proximal articular surface of the ulna has the typical triradiate outline of sauropods. Its anterior and anterolateral proximal rami are approximately equidimensional, forming a slightly obtuse angle. The posterior ramus does not form a distinct projection dorsally (there is no well-defined olecranon process) or posteriorly (the ulna is V-shaped, rather than Y-shaped, in proximal view). Relative to the length of the ulna, the proximal end is slender (table 4), even accounting for some missing material. This is in marked contrast to the robust proximal ulnae of apatosaurines [2,47]. A transversely broad ridge extends distally from the posterior margin. Around midheight, the medial margin of this ridge becomes less distinct, such that the transition between the medial and posterior surfaces is rounded. However, the lateral margin of this ridge expands at approximately midheight, continuing distally for approximately another quarter of the length of the bone, before it also disappears. On the anterior surface, the scar for the articulation with the radius is only weakly developed, or possibly also slightly eroded. The distal articular surface is transversely compressed and expanded posteriorly. Given the lack of anatomical overlap, coupled with uncertainty in its provenance, it is not possible to refer the ulna to *A. altus*. Given that none of the features noted above indicate referral to a particular taxon, but are broadly consistent with most eusauropod clades, we regard the AMNH FARB 5764a ulna as belonging to an indeterminate eusauropod.

The distal half of the femur (AMNH FARB 5764a [Fem. 2]) preserves the distal-most extension of the fourth trochanter (figure 12), which was not recognized by Osborn & Mook [42]. The location of this feature demonstrates that the femur is a right element, rather than a left as originally identified, and it is likely that the femur is proximodistally much shorter than indicated in the reconstruction in Osborn & Mook [42]. The preserved distal end of the fourth trochanter has a convex medial surface and is restricted to the medial margin of the posterior surface of the shaft. Although slender (table 4), the shaft below the fourth trochanter lacks the subcircular outline (mediolateral to anteroposterior diameter ratio = 1.2) that characterizes the holotypic femur of *A. altus*. The anterior parts of the distal

(a)    (b)

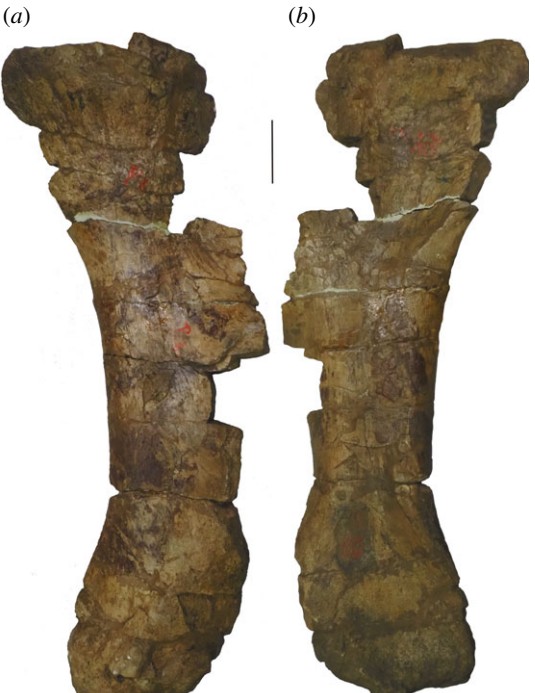

**Figure 13.** AMNH FARB 5761 [Pb. 6], left (?) pubis, in presumed (a) lateral and (b) medial views. Scale bar, 100 mm.

condyles are damaged, likely during excavation. Whereas parts of the distal articular surface are preserved, the posterior extension of the condyles is damaged as well. The distal articular condyles are moderately expanded transversely, and there is no indication of a well-developed epicondyle (although this might be affected by damage). The tibial condyle projects slightly further distally than the fibular condyle, which is the reverse of the condition in the holotypic femur of *A. altus*. As such, this distal femur clearly differs from that of *A. altus*. The combination of a posteriorly restricted fourth trochanter, a relatively slender midshaft and a dorsolaterally bevelled distal surface leads us to identify the AMNH FARB 5764a femur as an indeterminate diplodocine, possibly with close affinities to *Diplodocus*.

## 4.2. Additional AMNH FARB material catalogued as *Amphicoelias altus*

Two pubes and a humerus (figures 13 and 14) from the Morrison Formation of the Garden Park area in the AMNH collections have been tentatively identified as *A. altus* in the accession records. No basis for these identifications is documented but they were presumably based on locality and/or gross morphology.

A left (?) pubis (AMNH FARB 5761 [Pb. 6]; figure 13) has no associated information concerning locality, other than being from Cope's Garden Park collection. It is associated with a note in the collections, which seems to identify this bone as AMNH FARB 30012, but this number was erroneously attributed to this pubis and should be ignored (see 'Curatorial history'). The pubis is in two pieces, and the medial side is not preserved, including the articular surface for its counterpart. An anonymous hand-written note in the collections indicates that its referral to *A. altus* was based on its similarity in slenderness to the holotypic pubis. However, as in the holotypic pubis of *A. altus*, there is little that can be observed concerning informative anatomy. It possesses no features currently considered to be autapomorphic of any sauropod taxon and its preserved anatomy is consistent with that of numerous eusauropod clades. As a result, we regard this specimen as belonging to an indeterminate eusauropod.

The second pubis (AMNH FARB 5760 [Pb. 3]) preserves only the proximal half, and there is also no further locality information associated with this specimen. No significant morphological information can be gleaned that could enable the identification of this bone to any sauropod taxon. Combined with the lack of provenance information, there is no basis for its referral to *Amphicoelias* and we consider it to represent an indeterminate eusauropod.

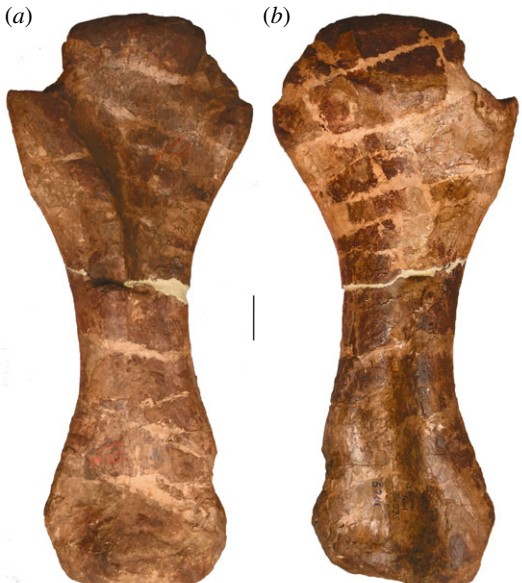

**Figure 14.** AMNH FARB 5761 [H.1], right humerus, in (*a*) anterior and (*b*) posterior views. Scale bar, 100 mm.

A right humerus (AMNH FARB 5761 [H. 1]; figure 14; see table 4 for measurements) was collected at the same site as the *A. altus* holotypic material in 1882, i.e. several years later than the original excavation of 1877 [41]. As with the Pb. 6 pubis, this humerus has also been associated with a new specimen number (AMNH FARB 30011), which should be ignored (see 'Curatorial history'). The humerus is fairly complete, lacking only the proximomedial and proximolateral corners. It is relatively stout (robustness index of 0.3), and is symmetrically expanded proximally, indicating that it is probably referable to Diplodocoidea [5]. The deltopectoral crest has a distinct distal end that extends to approximately 42% of the total proximodistal length of the humerus. Its lateral surface is anteroposteriorly concave and posteriorly accompanied by a distinct, striated ridge that extends for a little more than the proximal half of the deltopectoral crest. The tubercle for the attachment of the M. coracobrachialis is situated in the centre of the anterior concavity of the proximal end, as in most neosauropods [50], and differs from the medially displaced tubercle that characterizes the diplodocine *Galeamopus pabsti* [16]. At the midshaft, the humerus has an elliptical cross-section, with the mediolateral diameter 1.7 times greater than its anteroposterior dimension. The distal articular surface is associated with a relatively distinct intercondylar groove on the posterior surface of the shaft. Although damaged, the medial and lateral ridges on the anterior surface of the distal end would have been located close to the midline of the articular surface. The ratio of the length of this humerus to that of the type femur of *Amphicoelias* is 0.64 [41], which is consistent with the ratios in most diplodocoids, but is much lower than other eusauropods [2,5,13,47,50]. However, McIntosh [41] considered it unlikely that this robust humerus could have belonged to *Amphicoelias*, with its slender femur. Presumably based on its symmetrically expanded proximal portion and the relative robusticity, McIntosh [41] referred this humerus to *Apatosaurus* instead. Although it remains possible that the humerus ultimately belongs to *Amphicoelias*, this would mean that the animal had very unusual limb proportions in terms of relative robusticity. As such, we follow McIntosh in excluding this humerus from *Amphicoelias* and, based on its overall morphology, we refer it to Diplodocoidea.

## 4.3. 'Amphicoelias latus': AMNH FARB 5765

In the same publication in which he named *A. altus*, Cope [9] erected a second species—'*Amphicoelias latus*'—on the basis of four caudal vertebrae and a femur (AMNH FARB 5765) from a nearby locality (Cope Quarry XV). The femur of this specimen is much more robust than that of *A. altus*, and also has an anteroposteriorly compressed, elliptical midshaft cross-section. Osborn & Mook [42] noted this difference and regarded '*Amphicoelias latus*' as a junior synonym of *Camarasaurus supremus*. This referral has been followed by subsequent authors (e.g. [2,13,41]), and was supported through phylogenetic analysis [5]. Woodruff & Foster [37] incorrectly stated that previous authors (i.e. [41,42]) had synonymized '*Amphicoelias latus*' with *A. altus*, which they remarked that their 'analysis agrees

with' (p. 216). However, there is no basis for such a referral and we agree with other workers that 'Amphicoelias latus' is a junior synonym of *Camarasaurus supremus*.

### 4.4. *Maraapunisaurus* (*Amphicoelias*) *fragillimus*

Cope [35] named a third species of *Amphicoelias* from a nearby locality (Cope Quarry III) the following year. Based only on a middle–posterior dorsal neural arch (AMNH FARB 5777), Cope [35] erected *A. fragillimus*. This specimen was unfortunately lost (or destroyed) but, despite this, has been the focus of several studies because of its potentially gigantic size [17,37,98]. Most authors have synonymized it with *A. altus* (e.g. [2,13,37,42]), or listed it as a nomen dubium (e.g. [5] (note that these authors did not provide a diagnosis for *A. fragillimus*, as incorrectly stated by Woodruff [40]). However, Carpenter [17] provided a novel reinterpretation in which he considered *A. fragillimus* as a rebbachisaurid diplodocoid, erecting the new genus *Maraapunisaurus*. All that remains of *Maraapunisaurus* (*Amphicoelias*) *fragillimus* is the drawing of the neural arch in posterior view, as presented by Cope [35]. Based on this, we agree with Carpenter [17] that it differs in a number of anatomical features from *A. altus* (e.g. the morphology of the SPOLs, and the presence of a distinct postspinal lamina). We tentatively concur that *Maraapunisaurus* might represent a rebbachisaurid, following the arguments presented by Carpenter [17].

### 4.5. MOR 592

MOR 592 is a skeleton from the Morrison Formation of Montana that consists of a braincase, partial dentary, 12 presacral and 7 caudal vertebrae, a pelvis and femur. In a conference abstract, Wilson & Smith [60] suggested that MOR 592 might be referable to *Amphicoelias*, based on the slenderness of the femur, a subcircular femoral cross-section at midshaft and reduced pleurocentral openings in the posterior dorsal centra. Those authors noted that the phylogenetic position of the material was unstable, and the assignment to *Amphicoelias* was regarded as tentative (J.A. Wilson Mantilla, personal communication, 2011, in [49]). Whitlock [49] revisited this material and considered it to potentially represent a dicraeosaurid, based on the sharp supraoccipital crest and a symphyseal tuberosity on the dentary, a diagnosis followed by some later work (e.g. [99]). However, based on the postcrania, Woodruff & Fowler [36] suggested that MOR 592 was instead a juvenile morphotype of a diplodocine. Woodruff & Foster [37] and Woodruff *et al.* [38,100] later went further and considered MOR 592 to be a juvenile specimen of *Diplodocus*. Most recently, MOR 592 has been recovered as a dicraeosaurid in the phylogenetic analysis of Whitlock & Wilson Mantilla [7].

Regardless of whether MOR 592 is considered a dicraeosaurid or an immature specimen of *Diplodocus*, it is generally agreed that it does not belong to *Amphicoelias*. However, the femur of MOR 592 is apparently characterized by a subcircular cross-section at the midshaft [60], and its distal end appears to be slightly bevelled, with the fibular condyle extending further distally than its tibial counterpart ([38]: fig. 15), both of which are herein regarded as potential autapomorphies of *Amphicoelias*. Photographs of the femur of MOR 592, provided by C. Woodruff, reveal that the ratio of the mediolateral to anteroposterior diameters of its midshaft is approximately 1.3. As such, the femur of MOR 592 lacks the subcircular cross-section that characterizes that of *Amphicoelias*. The distal bevelling in MOR 592 is also less pronounced than in the type of *Amphicoelias*. Furthermore, it is possible that both the shape of the midshaft and the distal end morphology have been affected by crushing in MOR 592, with mediolateral compression apparent from the heavily cracked anterior surface ([38]: fig. 15). For now, we exclude MOR 592 from *Amphicoelias*, but this specimen is clearly in need of detailed study to determine its taxonomic affinities.

## 5. Phylogenetic position of *Amphicoelias altus*

### 5.1. Approach

To assess the phylogenetic position of *A. altus*, we used three independent data matrices, all of which already include this species as an OTU. The first of these is the data matrix of Mannion *et al.* [50]. This focused on the evolutionary relationships of eusauropods in general, with *Amphicoelias* consistently resolved as a non-diplodocimorph diplodocoid. Scores for *Amphicoelias* in this data matrix are unchanged following our revision herein. The second data matrix is that of Whitlock & Wilson

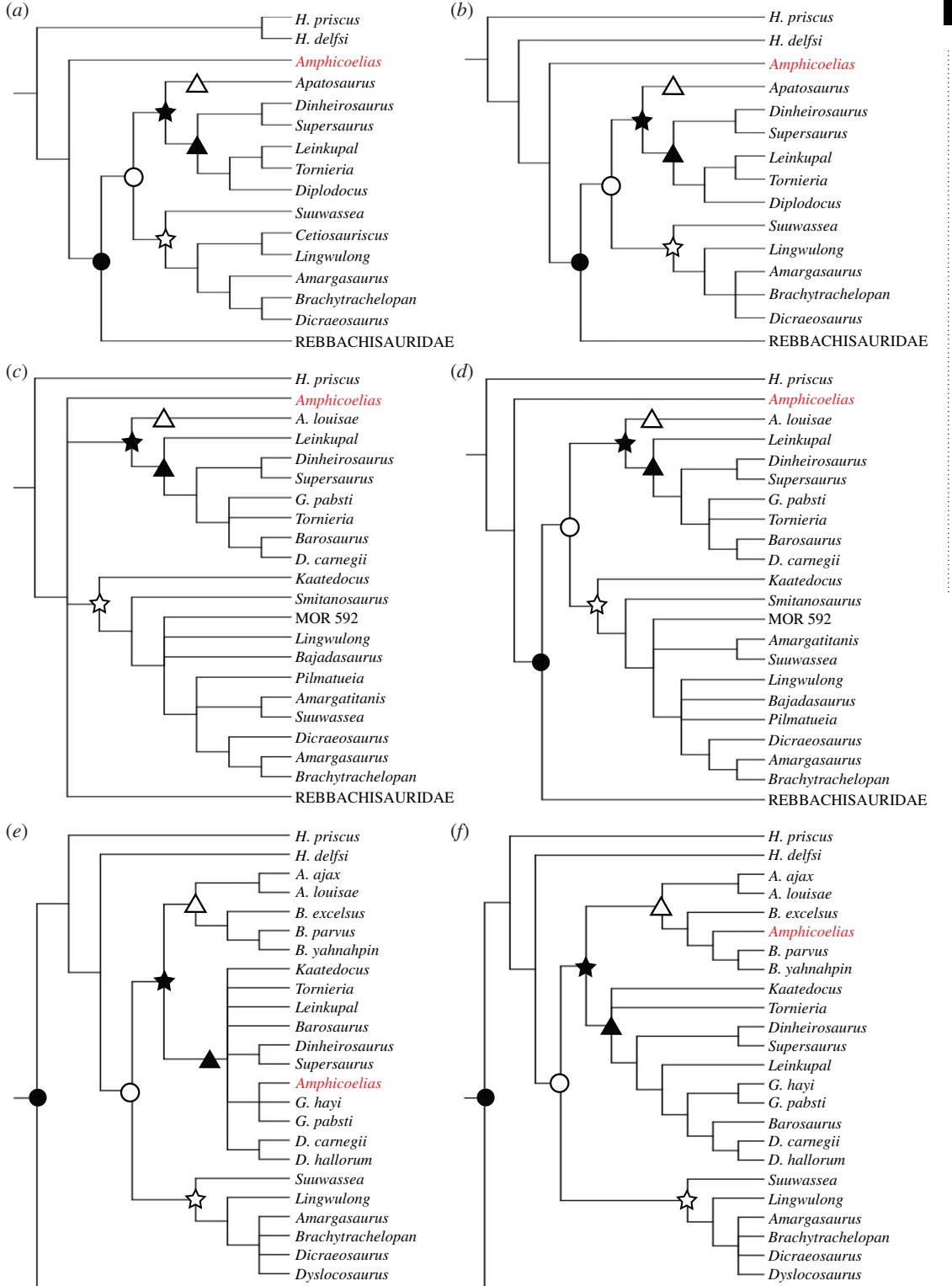

**Figure 15.** Phylogenetic position of A. altus showing strict consensus trees of diplodocoid interrelationships based on Mannion et al. [50] with (a) EQW and (b) EIW, Whitlock & Wilson Mantilla [7] with (c) EQW and (d) EIW, and Xu et al. [92] with (e) EQW and (f) EIW. Black circle, Diplodocimorpha; white circle, Flagellicaudata; black star, Diplodocidae; white star, Dicraeosauridae; black triangle, Diplodocinae; white triangle, Apatosaurinae.

Mantilla [7], which focused on diplodocoid relationships. *Amphicoelias* was excluded from that version of the data matrix, but was included in previous iterations (e.g. [49,51]), in which it was consistently recovered as a non-diplodocimorph diplodocoid. It is rescored for the dataset of Whitlock & Wilson

**Table 5.** Results of phylogenetic analyses using the three data matrices, including constraints applied to *A. altus*. Number of steps values for EIW analyses rounded to one decimal place.

| data matrix | constraint | EQW | | EIW | |
|---|---|---|---|---|---|
| | | MPTs | steps | MPTs | steps |
| Mannion *et al.* [50] | none | 22 704 | 2551 | 735 | 133.5 |
| Mannion *et al.* [50] | Diplodocidae | 163 056 | 2552 | 735 | 133.7 |
| Mannion *et al.* [50] | Apatosaurinae | 163 056 | 2555 | 735 | 133.8 |
| Mannion *et al.* [50] | Diplodocinae | 163 056 | 2555 | 735 | 133.8 |
| Mannion *et al.* [50] | *Diplodocus* as sister taxon | 163 056 | 2557 | 735 | 133.9 |
| Whitlock & Wilson Mantilla [7] | none | 24 | 361 | 12 | 12.7 |
| Whitlock & Wilson Mantilla [7] | Diplodocidae | 4 | 361 | 4 | 12.8 |
| Whitlock & Wilson Mantilla [7] | Apatosaurinae | 4 | 362 | 4 | 12.9 |
| Whitlock & Wilson Mantilla [7] | Diplodocinae | 12 | 362 | 12 | 12.9 |
| Whitlock & Wilson Mantilla [7] | *Diplodocus* as sister taxon | 4 | 363 | 4 | 13.0 |
| Xu *et al.* [92] | none | 60 | 1632 | 6 | 88.2 |
| Xu *et al.* [92] | Apatosaurinae | 120 | 1633 | N/A | N/A |
| Xu *et al.* [92] | Diplodocinae | N/A | N/A | 6 | 88.2 |
| Xu *et al.* [92] | *Diplodocus* as sister taxon | 60 | 1635 | 6 | 88.3 |
| Xu *et al.* [92] | sister taxon of other flagellicaudatans | 442 | 1634 | 6 | 88.3 |
| Xu *et al.* [92] | non-diplodocimorph diplodocoid | 175 | 1639 | 3 | 88.5 |

Mantilla [7]. The third data matrix is that of Xu *et al.* [92], which is a slightly expanded version of the dataset presented by Tschopp & Mateus [16]. This matrix focused on flagellicaudatan relationships, with *Amphicoelias* consistently recovered as an apatosaurine. We have made 12 character-score changes to the *Amphicoelias* OTU in this matrix. Furthermore, we have also made 37 changes to character scores of the existing OTU of *Brontosaurus excelsus* (YPM VP.001980 and VP.001981), as well as adding in three additional OTUs of Morrison Formation sauropods to increase the sampling of approximately contemporaneous, non-diplodocid taxa: *Dyslocosaurus polyonychius* (AC 663), *Dystrophaeus viaemalae* (USNM 2364) and *Haplocanthosaurus delfsi* (CMNH 10380). These changes and additions to the Xu *et al.* [92] data matrix are all based on personal observations by E. Tschopp, as well as information presented in McIntosh & Williams [34]. The revised character scores for *A. altus* and *Brontosaurus excelsus* are documented in the appendix A.

We ran Parsimony analyses for the three data matrices, applying both equal weighting (EQW) and extended implied weighting (EIW, using a *k*-value of 9) of characters [101,102]. We also tested the effect of alternative phylogenetic positions through the application of several constraints (table 5). All phylogenetic analyses were carried out using the 'Stabilize Consensus' option in the 'New Technology Search' in TNT v. 1.5 [103,104], with characters ordered following previous treatments [7,16,50,92]. Searches used sectorial searches, drift and tree fusing, with the consensus stabilized five times before the resultant trees were used as the starting topologies for a 'Traditional Search', using tree bisection–reconnection. All three data matrices used herein are provided as TNT files (electronic supplementary material), with stored settings for assigning characters as ordered.

## 5.2. Results

The numbers of most parsimonious trees (MPTs) and steps for all analyses are summarized in table 5. Strict consensus trees showing the interrelationships of Diplodocoidea, without constraints, are presented in figure 15. Mannion *et al.* [50] recovered *Amphicoelias* as the sister taxon to Diplodocimorpha (figure 15*a,b*). Constraining *Amphicoelias* within Diplodocidae in this data matrix requires only one extra step with EQW and 0.2 steps with EIQ, with it recovered as the most 'basal' member of this clade. When forced into Diplodocinae, the number of steps increases by four with EQW and 0.3 with EIW, with *Amphicoelias*

recovered as the most 'basal' member of the clade. The same increase in step number is required to enforce a position in Apatosaurinae. Six extra steps with EQW and 0.4 steps with EIW are required to recover *Amphicoelias* as the sister taxon to *Diplodocus*.

Analysis of the Whitlock & Wilson Mantilla [7] recovers *Amphicoelias* in a polytomy with the three diplodocimorph clades with EQW and as the sister taxon to Diplodocimorpha with EIW (figure 15*c*,*d*). Other relationships are largely congruent with previous iterations of this matrix, including the recovery of MOR 592 as a dicraeosaurid. Enforcing *Amphicoelias* into Diplodocidae requires no additional steps with EQW and 0.1 steps with EIW, with it recovered as the most 'basal' member of this clade. A position within Diplodocinae requires one extra step with EQW and 0.2 steps with EIW: *Amphicoelias* is placed in a polytomy with other members of this clade, but it is recovered outside of both the *Diplodocus* + *Galeamopus* and *Dinheirosaurus* + *Supersaurus* clades. The same increase in step number is required to enforce a position in Apatosaurinae. Two extra steps with EQW and 0.3 steps with EIW are required to recover *Amphicoelias* as the sister taxon to *Diplodocus*.

Analysis of the Xu *et al.* [92] matrix using EQW recovers *Amphicoelias* within Diplodocinae, in a polytomy with the two species of *Galeamopus* (figure 15*e*). By contrast, the application of EIW places *Amphicoelias* within Apatosaurinae, clustering with species of *Brontosaurus* (figure 15*f*). Other relationships are largely congruent with previous iterations of this matrix, although the two *Haplocanthosaurus* species are consistently recovered as more closely related to Flagellicaudata than Rebbachisauridae. Constraining *Amphicoelias* to be a member of Apatosaurinae in the EQW analysis requires only one additional step. The number of steps in the EIW analysis is almost unchanged (less than 0.001) when *Amphicoelias* is forced inside of Diplodocinae, where it is recovered in a polytomy with the two *Galeamopus* species. Three extra steps with EQW and 0.1 steps with EIW are required to recover *Amphicoelias* as the sister taxon to *Diplodocus*. A sister taxon relationship between *Amphicoelias* and other flagellicaudatans requires two extra steps with EQW and 0.1 steps with EIW. Constraining *Amphicoelias* to be a non-diplodocimorph diplodocoid requires seven additional steps with EQW and 0.3 steps with EIW.

# 6. Discussion

## 6.1. Is *Amphicoelias altus* synonymous with *Diplodocus*?

Although a non-diplodocimorph diplodocoid position is the most parsimonious placement for *A. altus* in analyses of two of our three data matrices, we cannot currently reject a placement within Diplodocidae, either as an apatosaurine or a diplodocine, with a sister taxon relationship with *Diplodocus* only requiring a relatively small number of additional steps. However, our detailed re-description, combined with phylogenetic analyses [5,16,48–51]; this study), supports *A. altus* as a distinct diplodocoid, diagnosed by three autapomorphies. Although only a small number of steps are required to recover *Amphicoelias* in alternative positions, this is perhaps not so surprising, given the incomplete nature of the only known specimen. In the three data matrices evaluated here, *Amphicoelias* can be scored for 10% of characters (56/542) in that of Mannion *et al.* [50], with this percentage rising to 12% (58/491) in that of Xu *et al.* [92], and 16% (32/203) in Whitlock & Wilson Mantilla [7]. Given that we are able to identify three autapomorphies regardless of its placement in Diplodocoidea, and that this is expanded by additional local autapomorphies when positioned within Diplodocidae, we suggest that a more completely preserved specimen of *Amphicoelias* is likely to be further distinguishable from other diplodocoids across a greater spectrum of its anatomy. Our analyses based on the data matrix of Xu *et al.* [92] also suggest that *Amphicoelias* is more likely to be closely related to either *Brontosaurus* or *Galeamopus* than it is to *Diplodocus* or any other Morrison Formation sauropod. As such, we argue against recent proposals that *Amphicoelias* is likely to be synonymous with *Diplodocus* [4,37,40], which also means that we retain the latter as a valid genus. Given the labile position of *Amphicoelias* and the current state of flux in Morrison Formation diplodocid systematics (e.g. [5]), we consider it premature and unhelpful to make taxonomic changes based on our study. However, in the case of future possible synonymizations, *Amphicoelias* [9] has priority over both *Brontosaurus* [24] and *Galeamopus* [5].

Although we restrict the known record of *Amphicoelias* to a single individual, it is possible that it was more common than we currently recognize. Given that much of the collection efforts in the Morrison Formation began at the dawn of dinosaur palaeontology, many elements were excavated and labelled as one of the handful of well-known sauropod taxa at the time, particularly *Apatosaurus*, *Camarasaurus* and *Diplodocus*. Recent years have witnessed the erection of multiple new Morrison Formation taxa from material previously referred to existing species, including *Galeamopus* [16], *Kaatedocus* [6] and

*Smitanosaurus* [7], and re-evaluation of historic material has shown that some specimens previously referred to *Apatosaurus* and *Diplodocus* might represent *Galeamopus* instead [97]. It is, therefore, possible that material pertaining to *Amphicoelias* awaits a collections visitor in a cabinet labelled *Apatosaurus* or *Diplodocus*.

## 6.2. Sauropod diversity in the Morrison Formation

Recent taxonomic studies support the validity of 24 sauropod species assigned to 14 genera in the Upper Jurassic Morrison Formation ([5,7,13,14,16,17]; table 1). In a series of papers, Woodruff and co-workers have argued that this diversity is overestimated (see also [105]), and that many species considered valid in recent studies are synonymous, primarily interpreted as semaphoronts or 'ontogimorphs' (i.e. growth series) of a smaller number of valid taxa [36–38,40,92]. These putative synonyms include *A. altus*, *Barosaurus lentus*, *Haplocanthosaurus priscus*, *Kaatedocus siberi*, *Smitanosaurus agilis* and *Suuwassea emilieae*. In addition, Woodruff and co-workers considered it unlikely that this high diversity of megaherbivores could have all coexisted in one geographical region. Below we evaluate these suggestions that sauropod diversity in the Morrison Formation is inflated (see also [106]).

### 6.2.1. Are some Morrison sauropod species ontogimorphs?

One key aspect of the proposal that sauropod diversity in the Morrison Formation is inflated is the hypothesis that some currently recognized species are ontogimorphs of other species. If this is correct, then we should expect to observe substantial anatomical changes along a growth series in a single sauropod species, such that immature individuals lack apomorphies present in adult individuals. One further consequence of this is that we might, therefore, expect sauropod species based on immature specimens to be affected by stemward slippage (i.e. recovery in an artefactually 'basal' position) when included in phylogenetic analyses, given that they should lack apomorphies that would otherwise draw them to their 'correct' placement.

Unfortunately, there is a global dearth of sauropod remains for which we have a growth series that can be unambiguously referred to a single species, making it difficult to test hypotheses pertaining to taxonomic inflation resulting from ontogeny. The best-known test case of ontogenetic variation in a sauropod is the dwarf macronarian *Europasaurus holgeri* from the Late Jurassic of Germany, which is represented by more than a dozen individuals from one locality, with remains representing juvenile through to adult stages of growth [107]. Here and below, we follow Hone *et al.* [108] in broad definitions of 'juveniles' versus 'subadults', with specimens in the former age class showing little or no skeletal fusion. Carballido & Sander [89] demonstrated that most anatomical features, including autapomorphies, were present by the late immature stage of growth, and that these provide the same phylogenetic signal as adult individuals. Although only based on one species, and a potentially unusual one because of its dwarf status, *Europasaurus* suggests that sauropod anatomy does not change dramatically once individuals reach subadult growth stages, and that autapomorphies will neither develop nor be lost beyond that stage. Similar inferences can be made from recent descriptions of juvenile material referred to the monospecific genus *Barosaurus* [74,109], where clearly defined autapomorphies of the taxon appear early in osteological development.

The most robust test of this hypothesis in the Morrison Formation comes from Ikejiri *et al.* [95]. Two adult individuals of *C. lentus* at different ontogenetic stages were found in one quarry and compared with two subadult specimens from elsewhere in the formation. Although Ikejiri *et al.* [95] noted several anatomical differences between the two adult individuals, these were typical of the features associated with advanced age or repetitive stress [91]. All four semaphoronts in the study were clearly referable to the same species (and distinguishable from other species) based on autapomorphies, as well as the overwhelming majority of anatomical features. The limited evidence from *Camarasaurus* and *Europasaurus* suggests that sauropods did not radically change their anatomy once they reached mature stages of growth, undermining claims that species such as *A. altus*, *Barosaurus lentus* and *Haplocanthosaurus priscus*, known from adult remains, represent growth stages in *Diplodocus* ontogeny.

The second expectation of the ontogimorph hypothesis is that immature sauropod individuals might be affected by stemward slippage when included in phylogenetic analyses. This seems unlikely to be the case for subadult specimens, given that they should possess a character suite that is extremely close to that of adult members of their species (e.g. [89]). As such, we dispute the claim that species such as *Haplocanthosaurus priscus* are pulled into a 'basal' diplodocoid position as a result of their ontogenetic stage, when they are clearly known from adult, or near-adult, specimens [99]. None of the type

specimens of Morrison Formation sauropod species regarded as potential ontogimorphs are interpreted as juveniles [99,108,110], and it is ultimately unlikely that many of them are even truly subadult in a modern-day biological sense [108]. Nevertheless, three small-bodied taxa might be based on subadult specimens, i.e. *Kaatedocus*, *Smitanosaurus* and *Suuwassea*. These taxa are all recovered close to the 'base' of Flagellicaudata, either as early diverging dicraeosaurids or diplodocids [7,16]. However, it would necessitate major increases in the number of evolutionary steps for phylogenetic analyses to recover them clustering with their proposed senior synonym, *Diplodocus* [99]. In addition, the placement of at least two of these taxa in Dicraeosauridae cannot be accredited to stemward slippage along the diplodocid clade.

Furthermore, even the inclusion of juvenile specimens does not necessarily result in their stemward slippage. In their diplodocid-focused phylogenetic analysis, Tschopp *et al.* [5] recovered the juvenile type specimen of *Brontosaurus* (*Elosaurus*) *parvus* as most closely related to the adult apatosaurine specimen found in association [25,56], despite the presence of several plesiomorphic features that resulted in character-score variation between the two individuals (see also [14]). As such, it seems unlikely that the phylogenetic positions of taxa such as *Amphicoelias*, *Barosaurus*, *Haplocanthosaurus priscus*, *Kaatedocus*, *Smitanosaurus* or *Suuwassea*, known from subadult or adult remains, are incorrect to the point that they have been misidentified as distinct from *Diplodocus* or other Morrison taxa. It is important to note that whether apatosaurine species belong to *Apatosaurus* or *Brontosaurus*, or if *Brontosaurus* or *Galeamopus* species belong to *Amphicoelias*, is ultimately irrelevant to discussions of diversity (cf. [105]), given the arbitrary nature of genera.

The above does not preclude the possibility that some phylogenetically informative features can have an ontogenetic signal too. For example, diplodocine individuals might possess a postparietal foramen that is lost during ontogeny [38] *and* its presence is also a synapomorphy of some dicraeosaurids [49]. Similarly, ontogenetically old individuals of *Camarasaurus* might incorporate a sixth sacral vertebra (e.g. [111]) *and* a 6-vertebra sacrum also characterizes Somphospondyli [77]. However, although these ontogenetic features might introduce some degree of 'directed noise' [112], they do not outweigh the overwhelming phylogenetic signal when OTUs are not early juveniles, at least in sauropods. Returning to *Europasaurus*, this taxon is universally recovered as an early diverging macronarian (e.g. [50,75,96,107]). However, likely via paedomorphic retention, *Europasaurus* possesses several features (including a postparietal foramen) that are inconsistent with a placement in Macronaria or even Neosauropoda [89,113], but these do not result in stemward slippage [114]. We do not disagree that ontogeny is an important issue and one that needs to be considered, but there is currently no evidence to support the claim that Morrison sauropod diversity is overestimated as a result. As also concluded by Wedel & Taylor [99], we should be wary of placing too much emphasis on 'critical' phylogenetic characters (*sensu* [36]).

### 6.2.2. Was apparent Morrison Formation sauropod diversity too high to be ecologically viable?

The second key aspect of the proposal that sauropod diversity in the Morrison Formation is inflated is the hypothesis that the coexistence of a high diversity of megaherbivores in one geographical region is not ecologically viable. This is much more difficult to test, but the notion of 24 co-occurring megaherbivore species is, at face value, potentially difficult to accept, with Woodruff [40, pp. 93, 105] considering it 'unlikely' and 'ecologically taxing'. Although the Morrison Formation covers an area exceeding 1.2 million $km^2$ [115], it is possible that this would have been unable to support enough viable individuals of 24 species (though see below). There are two key assumptions in this hypothesis, pertaining to coexistence and required levels for ecological viability, that we discuss below.

The first part of this hypothesis makes the assumption that all of these 24 species co-occurred with one another. However, this is misleading. Firstly, none of these 24 species are found throughout the spatial extent of the Morrison Formation. Many of these species are limited to a small number (less than 5) of occurrences (e.g. *Brachiosaurus altithorax*, *Haplocanthosaurus priscus*) or are currently known from a single locality (e.g. *A. altus*, *Haplocanthosaurus delfsi*, *Suuwassea emilieae*). However, even species known from abundant remains show evidence for some degree of geographical restriction (e.g. [4,39,97,116]). Furthermore, numerous fossiliferous Morrison localities have been sampled extensively and yet none contain more than five sympatric sauropod species [4]. These distributional data alone suggest that the 24 Morrison sauropod species did not all co-occur geographically.

Secondly, the Morrison Formation was deposited over a period of at least 7 Myr [58,59,117]. None of the 24 sauropod species are recovered throughout the formation's full temporal extent and many species were clearly not contemporaneous with others (e.g. [4,5,117]), although a substantial proportion of

localities with radiometric dates are approximately coeval [39]. Although framed around the 'problem' of having 24 contemporaneous megaherbivore species, Woodruff [40] showed the Morrison Formation divided into six 'biozones', in which the most diverse of these contains 12 distinct taxa (and note that three of these are specifically indeterminate members of contemporaneous genera). As such, stratigraphical data suggest that the 24 Morrison sauropod species did not all co-occur temporally either.

The second assumption of the hypothesis is that a large number of co-occurring sauropod species was ecologically unviable. Farlow *et al.* [118] modelled megaherbivore abundance in the Morrison Formation. Their results suggested an upper limit of a few hundred individuals per km$^2$, probably reduced to a few tens of individuals if these comprised entirely large subadults and adults. At their most conservative estimate of only four large individuals per km$^2$, this would still indicate that the areal extent of the Morrison Formation could have supported nearly 5 million individuals of giant sauropods at any one time. The approach of Farlow *et al.* [118] necessitated numerous assumptions and, even if their lowest estimate was correct, it is unlikely that this would have translated consistently across the entirety of the region given environmental heterogeneity. However, we highlight that the sole attempt to quantitatively test whether the Morrison Formation could have supported such high numbers of megaherbivores provided no evidence to suggest that this was ecologically taxing, regardless of whether most sauropods were contemporaneous. Furthermore, whereas there are numerous occurrences referred to some of the species of *Apatosaurus*, *Brontosaurus*, *Camarasaurus* and *Diplodocus* (e.g. [4]), the other Morrison sauropod taxa are known from only a small number of documented individuals. Although some of these poorly known species might ultimately be better represented in fossil collections than we currently realize (e.g. [97]), and there might be a historical bias against the collection of smaller-bodied sauropods, this dichotomy in terms of abundance likely reflects some degree of ecological veracity, i.e. many Morrison sauropods probably were relatively rare components of the preserved environments.

A large body of work has evaluated hypotheses of ecological niche partitioning in sauropods, especially Morrison Formation species. These studies demonstrate differences in browsing height, feeding strategy and dietary preferences between taxa (e.g. [94,119–130]). Combined with known geographical distributions, as well as the growing realization that the Morrison Formation was not a single, homogeneous environment (e.g. [39,123,131]), these provide a wealth of evidence to suggest that multiple contemporaneous sauropod species could have been viable through a combination of ecological, environmental and geographical partitioning, as well as uneven species abundance distributions. Furthermore, it is likely that some environments within the Morrison Formation are less well represented in the fossil record than others. Much of the formation remains unexposed [4], with clear spatial biases ([39]: fig. 1), and some environments might be less conducive to preservation in the first place (e.g. warmer and drier environments with low sedimentation rates). As such, we might have a very spatially (and environmentally) skewed idea of the Morrison Formation, one that under-represents, rather than inflates, sauropod diversity.

### 6.2.3. Summary

The arguments presented above suggest that there is currently no evidence for sauropod diversity inflation in the Morrison Formation. Combined with the serendipity of preserving species in the fossil record in the first place (e.g. [132]), the likelihood that we have not comprehensively sampled the full range of environments that existed through time and space (e.g. [133]), and our inability to recognize many extant archosaur (cryptic) species from morphology alone (e.g. [134]), we contend that the number of sauropod dinosaur species in the Morrison Formation is currently likely to be underestimated, not overestimated.

## 7. Conclusion

Our revision of the anatomy of the Upper Jurassic North American Morrison Formation taxon, *Amphicoelias altus*, supports its validity as a distinct species of diplodocoid sauropod dinosaur. Although its position within Diplodocoidea is labile, it seems unlikely that *A. altus* is synonymous with *Diplodocus*, and thus we are also able to preserve the latter as a valid genus. We evaluate recent claims that many other 'contemporaneous' sauropods represent growth series of other species, and thus Morrison Formation sauropod diversity is overestimated. There is currently no evidence to support either view, with many species unequivocally non-contemporaneous. Given this and the biases that obfuscate our reading of the fossil record, we suggest that known sauropod diversity is more likely to be underestimated, than overestimated, in the Morrison Formation.

Data accessibility. The datasets supporting this article have been uploaded as part of the electronic supplementary material.

Authors' contributions. P.D.M. conceived of and coordinated the study, and performed the analyses. E.T. implemented the Progressive Photonics workflow and produced the 3D models. J.A.W. drafted most of the anatomical figures. All authors participated in the design of the study, contributed and interpreted data, drafted figures and wrote the manuscript. All authors gave final approval for publication and agree to be held accountable for the work performed therein.

Competing interests. We declare we have no competing interests.

Funding. P.D.M.'s research was supported by a Royal Society University Research Fellowship (UF160216). E.T.'s contribution was supported by an AMNH Division of Paleontology Postdoctoral Fellowship.

Acknowledgements. We thank Mark Norell for enabling us to study and 3D scan *Amphicoelias altus* at the AMNH, as well as Carl Mehling, Verne Lee, Bruce Javors and the late Jack Conrad for their help accessing the material. We are also grateful to Mike Eklund and Bruce Javors for contributing photographs of AMNH specimens, and to Carolyn Merrill for producing the 3D scans. Cary Woodruff also kindly provided us with photographs of the femur of MOR 592. We also acknowledge the Willi Hennig Society, which has sponsored the development and free distribution of TNT. Reviews from Paul Barrett and Jeff Wilson Mantilla greatly improved an earlier version of this manuscript.

# Appendix A. Revised scores

The following changes were made to the existing scores of the OTUs *Amphicoelias altus* and *Brontosaurus excelsus* in the data matrix of Xu *et al*. [92]. In each case, the first number denotes the character and the numbers/symbols in parentheses denote the original and new scores:

*Amphicoelias altus*: 129 (1 → ?), 234 (? → 1), 235 (? → 0), 261 (0 → 1), 267 (0 → ?), 270 (0 → 1), 273 (1 → 0), 278 (0 → 1), 280 (0 → 1), 283 (0 → 1), 445 (0 → 1), 446 (? → 0)

*Brontosaurus excelsus*: 148 (? → 1), 149 (? → 1), 166 (? → 0), 234 (? → 1), 235 (? → 0), 241 (0 → 0/1), 245 (0/1 → 0), 261 (0 → 0/1), 262 (0 → 0/1), 279 (1 → 0/1), 284 (1 → 0/1), 285 (1 → 0/1), 289 (? → 0), 290 (1 → 0/1), 297 (? → 0), 311 (? → 0), 324 (1 → 0/1), 326 (1 → 0/1), 361 (? → 0), 371 (0 → 0/1), 393 (0 → 0/1), 419 (? → 0), 422 (1 → 0), 423 (? → 0), 434 (0/1 → 1), 443 (0 → 0/1), 445 (0 → 0/1), 447 (0 → 0/1), 448 (0/1 → 0), 453 (? → 1), 456 (1 → 0/1), 457 (? → 0), 458 (? → 1), 459 (0 → 0/1), 464 (1 → 0/1), 488 (? → 0), 489 (? → 0).

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
