## [Peer Review File · Royal Society Open Science]

Review History

RSOS-200963.R0 (Original submission)

Review form: Reviewer 1 (Paul Barrett)

Is the manuscript scientifically sound in its present form?

Yes

Are the interpretations and conclusions justified by the results?

Yes

Is the language acceptable?

Yes

Do you have any ethical concerns with this paper?

No

Have you any concerns about statistical analyses in this paper?

No

Recommendation?

Accept with minor revision (please list in comments)

Comments to the Author(s)

Amphicoelias is one of the least-studied sauropods from the Late Jurassic Morrison Formation and the authors provide a detailed, thoughtful account that provides new anatomical information and establishes its taxonomic status more firmly. They use previous suggestions that Amphicoelias was an adult individual of another Morrison sauropod taxon to initiate a useful discussion on sauropod diversity and palaeoecology, and the influence of ontogeny on taxonomic decision making.

My comments are all relatively minor and most are provided on the annotated .pdf (Appendix A). They can be summarized in general as follows:

1. Although the authors do a thorough job in diagnosing Amphicoelias, a few more comparisons with other Morrison diplodocoids would reinforce their conclusions regarding the distinctiveness of Amphicoelias relative to the other taxa in the 'fauna', rather than relying on coarser clade-level comparisons.
2. The authors might consider some improvements to the figures (labeling of a few more key features mentioned in the text; comparative images to support the identification of autapomorphies).
3. Providing further character evidence (or noting the lack of character evidence) to support the identifications of specimens previously referred to Amphicoelias that they now regard as 'Eusauropoda indet.'
4. Addition of one or two further references.
5. Some minor typos/phrasing issues.

Paul M. Barrett

Review form: Reviewer 2

Is the manuscript scientifically sound in its present form?

Yes

Are the interpretations and conclusions justified by the results?

Yes

Is the language acceptable?

Yes

Do you have any ethical concerns with this paper?

No

Have you any concerns about statistical analyses in this paper?

No

Recommendation?

Major revision is needed (please make suggestions in comments)

Comments to the Author(s)

See attached (Appendices B & C).

Decision letter (RSOS-200963.R0)

Dear Dr Mannion

The Editors assigned to your paper RSOS-200963 "Anatomy and systematics of the diplodocoid *Amphicoelias altus* supports high sauropod dinosaur diversity in the Upper Jurassic Morrison Formation, USA" have made a decision based on their reading of the paper and any comments received from reviewers.

Regrettably, in view of the reports received, the manuscript has been rejected in its current form. However, a new manuscript may be submitted which takes into consideration these comments.

We invite you to respond to the comments supplied below and prepare a resubmission of your manuscript -- note: owing to attachment file limits in ScholarOne, you may find the reviewers' comments are cut from the message. If this is the case, please contact the editorial office who will resend the files via an alternative mechanism. Below the referees' and Editors' comments (where applicable) we provide additional requirements. We provide guidance below to help you prepare your revision.

Please note that resubmitting your manuscript does not guarantee eventual acceptance, and we do not generally allow multiple rounds of revision and resubmission, so we urge you to make every effort to fully address all of the comments at this stage. If deemed necessary by the Editors, your manuscript will be sent back to one or more of the original reviewers for assessment. If the original reviewers are not available, we may invite new reviewers.

Please resubmit your revised manuscript and required files (see below) no later than 16-Mar-2021. Note: the ScholarOne system will 'lock' if resubmission is attempted on or after this deadline. If you do not think you will be able to meet this deadline, please contact the editorial office immediately.

Please note article processing charges apply to papers accepted for publication in Royal Society Open Science (<https://royalsocietypublishing.org/rsos/charges>). Charges will also apply to papers transferred to the journal from other Royal Society Publishing journals, as well as papers submitted as part of our collaboration with the Royal Society of Chemistry (<https://royalsocietypublishing.org/rsos/chemistry>). Fee waivers are available but must be requested when you submit your manuscript (<https://royalsocietypublishing.org/rsos/waivers>).

Thank you for submitting your manuscript to Royal Society Open Science and we look forward to receiving your resubmission. If you have any questions at all, please do not hesitate to get in touch.

on behalf of Prof Kevin Padian (Subject Editor)

Associate Editor Comments to Author:

Thanks for your submission. Two reviewers have commented on the paper, with each providing useful commentary on your work. Please revise your work accordingly.

Editor comments to author:

Thanks for your submission. I am going to redirect the AE's recommendation from "major revision" to "reject/resub" mainly because the 3-week timeframe for a revision may be too short. Both reviewers have useful comments but Reviewer 2 brings up some perceived deficiencies that should require attention, notably improvements in the descriptions and illustrations, and the lack of a phylogenetic analysis (which should come naturally with a revised diagnosis). The perception that you are dismissive of some arguments of other authors should also be addressed. Please attend to these comments in your revision, and we look forward to a resubmission. Best wishes.

Reviewer comments to Author:

Reviewer: 1

Comments to the Author(s)

Amphicoelias is one of the least-studied sauropods from the Late Jurassic Morrison Formation and the authors provide a detailed, thoughtful account that provides new anatomical information and establishes its taxonomic status more firmly. They use previous suggestions that Amphicoelias was an adult individual of another Morrison sauropod taxon to initiate a useful discussion on sauropod diversity and palaeoecology, and the influence of ontogeny on taxonomic decision making.

My comments are all relatively minor and most are provided on the annotated .pdf (attached). They can be summarized in general as follows:

1. Although the authors do a thorough job in diagnosing Amphicoelias, a few more comparisons with other Morrison diplodocoids would reinforce their conclusions regarding the distinctiveness of Amphicoelias relative to the other taxa in the 'fauna', rather than relying on coarser clade-level comparisons.
2. The authors might consider some improvements to the figures (labeling of a few more key features mentioned in the text; comparative images to support the identification of autapomorphies).
3. Providing further character evidence (or noting the lack of character evidence) to support the identifications of specimens previously referred to Amphicoelias that they now regard as 'Eusauropoda indet.'
4. Addition of one or two further references.
5. Some minor typos/phrasing issues.

Paul M. Barrett

Reviewer: 2

Comments to the Author(s)

See attached.

===PREPARING YOUR MANUSCRIPT===

===PREPARING YOUR REVISION IN SCHOLARONE===

- An individual file of each figure (EPS or print-quality PDF preferred [either format should be produced directly from original creation package], or original software format).
 - An editable file of each table (.doc, .docx, .xls, .xlsx, or .csv).
 - An editable file of all figure and table captions.
- Note: you may upload the figure, table, and caption files in a single Zip folder.
- Any electronic supplementary material (ESM).
 - If you are requesting a discretionary waiver for the article processing charge, the waiver form must be included at this step.
 - If you are providing image files for potential cover images, please upload these at this step, and inform the editorial office you have done so. You must hold the copyright to any image provided.
 - A copy of your point-by-point response to referees and Editors. This will expedite the preparation of your proof.

- Ensure that your data access statement meets the requirements at <https://royalsociety.org/journals/authors/author-guidelines/#data>. You should ensure that you cite the dataset in your reference list. If you have deposited data etc in the Dryad repository, please include both the 'For publication' link and 'For review' link at this stage.
- If you are requesting an article processing charge waiver, you must select the relevant waiver option (if requesting a discretionary waiver, the form should have been uploaded at Step 3 'File upload' above).
- If you have uploaded ESM files, please ensure you follow the guidance at <https://royalsociety.org/journals/authors/author-guidelines/#supplementary-material> to include a suitable title and informative caption. An example of appropriate titling and captioning may be found at https://figshare.com/articles/Table_S2_from_Is_there_a_trade-off_between_peak_performance_and_performance_breadth_across_temperatures_for_aerobic_scope_in_teleost_fishes_/3843624.

Author's Response to Decision Letter for (RSOS-200963.R0)

See Appendix D.

RSOS-210377.R0

Review form: Reviewer 1 (Paul Barrett)

Is the manuscript scientifically sound in its present form?

Yes

Are the interpretations and conclusions justified by the results?

Yes

Is the language acceptable?

Yes

Do you have any ethical concerns with this paper?

No

Have you any concerns about statistical analyses in this paper?

No

Recommendation?

Accept as is

Comments to the Author(s)

The authors have made all of my suggested changes and I think this will be a valuable contribution on Morrison sauropods.

Review form: Reviewer 3

Is the manuscript scientifically sound in its present form?

Yes

Are the interpretations and conclusions justified by the results?

Yes

Is the language acceptable?

Yes

Do you have any ethical concerns with this paper?

No

Have you any concerns about statistical analyses in this paper?

No

Recommendation?

Accept as is

Comments to the Author(s)

This is a very well put together manuscript. It is thorough, well conceived and well written. I would recommend it for publication.

The conclusions presented in the manuscript are based upon sound reasoning following a thorough run-through of the available evidence; I am not perturbed by the the fact that the type material of *Amphicoelias altus* could not undergo histological analysis, as I firmly believe that the analysis of the anatomy and the phylogenetic analyses are of sufficient quality to still allow the authors to make the statements that they make in the conclusions.

There are no faults with the phylogenetic analyses as far as I am able to tell.

The figures are of good quality and show sufficient detail. They will also provide a useful resource for other researchers who may use this manuscript to gather anatomical data on the subject taxa for future studies.

Overall, I think that this is an important contribution and I would be happy to see it published in its current form.

I cannot see any citations or references that are missing from the lists.

I would, of course, be happy to review any further revisions that may be produced.

M. G. Baron

Decision letter (RSOS-210377.R0)

Dear Dr Mannion,

I am pleased to inform you that your manuscript entitled "Anatomy and systematics of the diplodocoid *Amphicoelias altus* supports high sauropod dinosaur diversity in the Upper Jurassic Morrison Formation, USA" is now accepted for publication in Royal Society Open Science.

on behalf of Prof Kevin Padian (Subject Editor)
openscience@royalsociety.org

Reviewer comments to Author:
Reviewer: 1

Comments to the Author(s)
The authors have made all of my suggested changes and I think this will be a valuable contribution on Morrison sauropods.

Reviewer: 3

Comments to the Author(s)
This is a very well put together manuscript. It is thorough, well conceived and well written. I would recommend it for publication.

The conclusions presented in the manuscript are based upon sound reasoning following a thorough run-through of the available evidence; I am not perturbed by the the fact that the type material of *Amphicoelias altus* could not undergo histological analysis, as I firmly believe that the analysis of the anatomy and the phylogenetic analyses are of sufficient quality to still allow the authors to make the statements that they make in the conclusions.

There are no faults with the phylogenetic analyses as far as I am able to tell.

The figures are of good quality and show sufficient detail. They will also provide a useful resource for other researchers who may use this manuscript to gather anatomical data on the subject taxa for future studies.

Overall, I think that this is an important contribution and I would be happy to see it published in its current form.

I cannot see any citations or references that are missing from the lists.

I would, of course, be happy to review any further revisions that may be produced.

M. G. Baron

Appendix A**ROYAL SOCIETY
OPEN SCIENCE****Anatomy and systematics of the diplodocoid *Amphicoelias altus* supports high sauropod dinosaur diversity in the Upper Jurassic Morrison Formation, USA**

Journal:	Royal Society Open Science
Manuscript ID	RSOS-200963
Article Type:	Research
Date Submitted by the Author:	03-Jun-2020
Complete List of Authors:	Mannion, Philip; University College London, Earth Sciences Tschopp, Emanuel; Universität Hamburg, Centrum für Naturkunde; American Museum of Natural History, Division of Paleontology Whitlock, John; Mount Aloysius College Department of Science and Mathematics; Carnegie Museum of Natural History, Section of Vertebrate Paleontology
Subject:	ecology < BIOLOGY, evolution < BIOLOGY, palaeontology < BIOLOGY
Keywords:	Dinosauria, Diplodocus , Late Jurassic, Morrison Formation, Ontogeny, Sauropoda
Subject Category:	Organismal and Evolutionary Biology

Author-supplied statements

Relevant information will appear here if provided.

Ethics

Does your article include research that required ethical approval or permits?:

This article does not present research with ethical considerations

Statement (if applicable):

CUST_IF_YES_ETHICS :No data available.

Data

It is a condition of publication that data, code and materials supporting your paper are made publicly available. Does your paper present new data?:

Yes

Statement (if applicable):

The datasets supporting this article have been uploaded as part of the supplementary material.

Conflict of interest

I/We declare we have no competing interests

Statement (if applicable):

CUST_STATE_CONFLICT :No data available.

Authors' contributions

This paper has multiple authors and our individual contributions were as below

Statement (if applicable):

PM conceived of and coordinated the study. All authors participated in the design of the study, contributed data, performed the analyses and interpretation of the data, and wrote the manuscript. ET and JW drafted the figures. All authors gave final approval for publication and agree to be held accountable for the work performed therein.

Anatomy and systematics of the diplodocoid *Amphicoelias altus* supports high sauropod dinosaur diversity in the Upper Jurassic Morrison Formation of the USA

Philip D. Mannion¹, Emanuel Tschopp² and John A. Whitlock^{4,5}

¹ Department of Earth Sciences, University College London, London, WC1E 6BT, UK

² Centrum für Naturkunde, Universität Hamburg, 20146 Hamburg, Germany

³ Division of Paleontology, American Museum of Natural History, Central Park West at 79th Street, New York 10024-5192, USA

⁴ Department of Science and Mathematics, Mount Aloysius College, Cresson, Pennsylvania 16630-1999, USA

⁵ Section of Vertebrate Paleontology, Carnegie Museum of Natural History, Pittsburgh, Pennsylvania 15213-4007, USA

Author for correspondence (Email: philipdmannion@gmail.com)

RRH: *AMPHICOELIAS ALTUS* & MORRISON SAUROPOD DIVERSITY

LRH: MANNION, TSCHOPP & WHITLOCK

ABSTRACT

Sauropod dinosaurs were an abundant and diverse component of the Upper Jurassic Morrison Formation of the USA, with 24 currently recognised species. However, some authors consider this high diversity to have been ecologically unviable and the validity of some species has been questioned, with suggestions that they represent growth series ('ontogimorphs') of other species. Under this scenario, high sauropod diversity in the Late Jurassic of North America is greatly overestimated. One putative ontogimorph is the enigmatic diplodocoid *Amphicoelias altus*, which has been suggested to be synonymous with *Diplodocus*. Given that *Amphicoelias* was named first, it has priority and thus *Diplodocus* would become a junior synonym. Here we provide a detailed re-description of *Amphicoelias altus* in which we restrict it to the holotype individual and support its validity, based on three autapomorphies. Our re-evaluation supports recent phylogenetic analyses that recover *Amphicoelias* as distantly related to *Diplodocus*, and thus the latter is also retained as a valid taxon. There is no evidence to support the view that any of the currently recognised Morrison sauropods are ontogimorphs. Available data indicate that sauropod anatomy did not dramatically alter once individuals approached maturity. Furthermore, subadult sauropod individuals are not prone to stemward slippage in phylogenetic analyses, casting doubt on the possibility that their taxonomic affinities are substantially misinterpreted. An anatomical feature can have both an ontogenetic and phylogenetic signature, but the former does not outweigh the latter when other characters overwhelmingly support the affinities of a taxon. Many sauropods were spatiotemporally and/or ecologically separated from one another. Combined with the biases that cloud our reading of the fossil record, we contend that the number of sauropod dinosaur species in the Morrison Formation is currently likely to be underestimated, not overestimated.

Keywords: Dinosauria; *Diplodocus*; Late Jurassic; Morrison Formation; Ontogeny; Sauropoda

1. Introduction

The Upper Jurassic Morrison Formation of the western USA has yielded a high diversity of sauropods, including some of the most iconic dinosaurs, such as *Brachiosaurus*, *Brontosaurus*, and *Diplodocus* (Bakker, 1971; McIntosh, 1990; Foster, 2007). *Apatosaurus* and *Camarasaurus* were also abundant components of this fauna, whereas a number of other sauropod taxa are known from far fewer remains (Foster, 2003). Some of these have only been recognised this century, comprising *Galeamopus* (Tschopp et al., 2015), *Kaatedocus* (Tschopp and Mateus, 2013), *Smitanosaurus* (Whitlock and Wilson Mantilla, in press), and *Suuwassea* (Harris and Dodson, 2004), whereas others have been known for much longer, consisting of *Amphicoelias* (Cope, 1877a), *Barosaurus* (Marsh, 1890), *Haplocanthosaurus* (Hatcher, 1903a), and *Supersaurus* (Jensen, 1985). Currently, 24 sauropod species assigned to 14 genera are recognised as valid in the Morrison Formation (e.g. Upchurch et al., 2004a,b; Lovelace et al., 2008; Tschopp et al., 2015, 2017; Carpenter, 2018; Whitlock and Wilson Mantilla, in press; Table 1), although some authors have suggested that many of these species are synonyms, and that they represent growth series of a smaller number of valid taxa (Woodruff and Fowler, 2012; Woodruff and Foster, 2014; Woodruff et al. 2017, 2018; Woodruff, 2019).

One of the earliest named and most enigmatic of Morrison sauropods is *Amphicoelias*. The type species, *Amphicoelias altus*, was erected by Cope in 1877 for two dorsal vertebrae, a partial pubis, and a femur. These elements were collected by A. Ripley from Cope Quarry XII in Garden Park, north of Cañon City, in Fremont County, Colorado, and were shipped to Cope by O. W. Lucas (McIntosh, 1998). Following Cope's death, his collection was acquired by the American Museum of Natural History (AMNH) in 1902. Two further species of *Amphicoelias* were named by Cope – *Amphicoelias latus* (Cope, 1877a) and *Amphicoelias fragillimus* (Cope, 1878) – but neither species is currently considered to belong to the genus (e.g. Osborn and Mook, 1921; Carpenter 2018). No further remains can currently be unambiguously referred to *Amphicoelias*, making it one of the rarest taxa in the Morrison Formation, despite being known for over 140 years.

Cope (1877a) considered *Amphicoelias* to be a close relative of *Camarasaurus*, although he classified them in the separate, monogeneric families, Amphicoeliidae and Camarasauridae, respectively. *Amphicoelias* was generally regarded as either a close relative or synonym of *Camarasaurus* for the following four decades (e.g. Marsh, 1896; Huene, 1908), before Osborn and Mook (1921) undertook a review of Cope's sauropod taxa, in which they argued that *Amphicoelias* was most closely related to the diplodocids *Barosaurus* and *Diplodocus*. Nopcsa (1928) also allied *Amphicoelias* with diplodocids and, from Romer (1956) onwards, a diplodocid or diplodocoid placement has been universally accepted (e.g. Wilson, 2002; Upchurch et al., 2004a; Rauhut et al., 2005; Whitlock, 2011a; Tschopp et al., 2015; Mannion et al., 2019). Rauhut et al. (2005) were the first to incorporate *Amphicoelias* into a phylogenetic analysis, recovering it as the most 'basal' diplodocoid, outside Diplodocimorpha. The diplodocoid-focused analysis of Whitlock (2011a) also placed *Amphicoelias* as a non-diplodocimorph diplodocoid, a position recovered in subsequent analyses of revised versions of this data matrix (e.g. Mannion et al., 2012; Gallina et al., 2014). This placement was also supported in analyses of a recent independent phylogenetic data matrix of eusauropods (Mannion et al., 2019). By contrast, analyses of a second independent data matrix, focussed on diplodocids, recovered *Amphicoelias* within Diplodocidae, either as a 'basal' member of this clade, or within

Apatosaurinae (Tschopp et al., 2015). The latter position was also supported by analyses of a
revised version of this data matrix (Tschopp and Mateus, 2017).

Although generally considered a valid genus (e.g. Upchurch et al., 2004a), there is not
universal agreement on this point. Osborn and Mook (1921) and other authors (e.g.
McIntosh, 1990) have commented upon similarities between *Amphicoelias* and other
Morrison diplodocids, especially *Diplodocus*, and a small number of studies have argued that
*Amphicoelias* is actually synonymous with *Diplodocus* (Foster, 2003; Woodruff and Foster,
2014; Woodruff, 2019). Given that *Amphicoelias* (Cope, 1877a) was named before
*Diplodocus* (Marsh, 1878), and therefore has priority, any such synonymisation would have
notable taxonomic ramifications.

Given the uncertainty of its taxonomic status and phylogenetic affinity, here we provide a
detailed re-description of the holotypic remains of *Amphicoelias altus*, reassess its validity,
and discuss the systematics of remains previously attributed to this genus. Finally, we
present a revised view of sauropod diversity in the Morrison Formation.

1.1. Institutional abbreviations

**AMNH**, American Museum of Natural History, New York, USA; **MOR**, Museum of the
Rockies, Bozeman, Montana, USA.

1.2. Anatomical abbreviations

**ACDL**, anterior centrodiapophyseal lamina; **ACPL**, anterior centroprezygapophyseal lamina;
**CPOF**, centropostzygapophyseal fossa; **CPRL**, centroprezygapophyseal lamina; **ISPOL**, lateral
spinopostzygapophyseal lamina; **mSPOL**, medial spinopostzygapophyseal lamina; **PACDF**,
parapophyseal centrodiapophyseal fossa; **PCDL**, posterior centrodiapophyseal lamina; **PCPL**,
posterior centroparapophyseal lamina; **PODL**, postzygodiapophyseal lamina; **POSDF**,
postzygapophyseal spinodiapophyseal fossa; **POSL**, postspinal lamina; **PRDL**,
prezygodiapophyseal lamina; **PRSDF**, prezygapophyseal spinodiapophyseal fossa; **PRSL**,
prespinal lamina; **SPDL**, spinodiapophyseal lamina; **SPOL**, spinopostzygapophyseal lamina;
**SPOL-F**, spinopostzygapophyseal lamina fossa; **SPRL**, spinoprezygapophyseal lamina; **TPRL**,
interprezygapophyseal lamina.

2. Systematic Paleontology

SAUROPODA Marsh, 1878
NEOSAUROPODA Bonaparte, 1986
DIPLODOCOIDEA Marsh, 1884
*AMPHICOELIAS* Cope, 1877a

**Type species:** *Amphicoelias altus* Cope, 1877a

**Holotype:** AMNH FARB 5764 – two middle–posterior dorsal vertebrae, a fragmentary pubis,
and a right femur (Figs 1–5).

**Locality and horizon:** Cope Quarry XII, Garden Park, 8 miles N/NE of Cañon City, Fremont
County, Colorado, USA (Cope, 1877a; Osborn and Mook, 1921; McIntosh, 1998); Morrison

Formation, C5 systems tract, 150.44–149.21 Ma, lower Tithonian, Upper Jurassic (Carpenter,
1998; Trujillo and Kowallis, 2015; Maidment and Muxworthy, 2019).

**Revised diagnosis:** *Amphicoelias* can be diagnosed by two autapomorphies (denoted by an
asterisk), as well as one local autapomorphy: (1) apex of posterior dorsal neural spine with
rounded, non-tapered lateral projections resulting from expansion of spinodiapophyseal
laminae*; (2) femoral shaft with subcircular cross section*; and (3) femur distally bevelled,
with the fibula condyle extending further distally than the tibial condyle.

**Additional information:** Numerous additional remains have been referred to *Amphicoelias*
(e.g. Cope, 1877a, 1878; Osborn and Mook, 1921; Wilson and Smith, 1996). These are
discussed and re-evaluated below, but we follow recent authors (e.g. Mannion et al., 2012,
2019; Tschopp et al., 2015) in restricting *Amphicoelias* to the type material.

**Curatorial history:** Curatorial history of the type and previously referred material is
complicated, mostly because of inadequate field notes from the excavations, but also due to
issues emanating from renumbering after the acquisition of Cope's Garden Park collection
by the AMNH. After acquisition, the material was catalogued with numbers ranging from
AMNH FARB 5760 to 5777. The type specimens of the three *Amphicoelias* species erected
by Cope were catalogued as AMNH FARB 5764 (*A. altus*), AMNH FARB 5765 (*A. latus*), and
AMNH FARB 5777 (*A. fragillimus*; this specimen is missing, and it is unclear if this specimen
was lost before, during, or after the acquisition by the AMNH). Additional material was later
referred to *Amphicoelias*, both in publications and internally in the museum's collections.

More recently, a second renumbering attempt seems to have been undertaken at AMNH,
presumably based on the detailed historical work of McIntosh (1998), who successfully
identified several partial skeletons among the AMNH material from Garden Park. It is not
known who conducted this renumbering, as this happened before current collection and
curatorial staff took office, and no notes exist that could be used to identify those involved.
To make matters even more complicated, these new numbers were solely noted on sheets of
pink paper associated with the bones in the collection space and were not transferred onto
the collection catalogue. Numbers used for the Garden Park material range from 30001 to
at least 30014, and possibly higher (ET, pers. obs. 2019). The renumbered specimens are
mostly from the original catalogue numbers AMNH FARB 5760 and 5761, which were shown
to include material from multiple individuals (Osborn and Mook, 1921; McIntosh, 1998).
These original numbers are the ones used in all scientific publications concerning the Cope
specimens from Garden Park (e.g. Osborn and Mook, 1921; McIntosh, 1998). The new
numbers, instead, do not correspond to their entries in the AMNH collection catalogue, in
which the numbers AMNH FARB 30001–30014 are listed as specimens of the turtles
*Brachyopsemys tingitana* (holotype, AMNH FARB 30001) and *Phosphatochelys* sp. (AMNH
FARB 30008), Testudines indet. (AMNH FARB 30002 and 30004–30007), an indeterminate
therapsid (AMNH FARB 30003), and the holotype (AMNH FARB 30009) and paratypes of the
frog *Xenopus arabiensis* (AMNH FARB 30010–30014; C. Mehling, pers. comm. 2019, 2020).
Among these erroneously recatalogued sauropod specimens are a humerus and pubis
originally cataloged as AMNH FARB 5761, which are also associated with anonymous
collection notes referring them to *Amphicoelias altus*, as is a second pubis numbered AMNH
FARB 5760, which was not renumbered. These elements correspond to the right humerus
H.1 (Osborn and Mook 1921: fig. 89), and the pubes P.3 (Osborn and Mook 1921: fig. 105)

and P.6 (Osborn and Mook 1921: fig. 106). Given that the renumbering has never been
transferred to the official AMNH collection catalogue, that these numbers are registered as
belonging to different species and specimens (including some type specimens), and that we
are not aware of any scientific publication referring to these numbers, we ignore the new
numbers between 30001 and 30014 associated with these bones in the collection space,
and use only the original catalogue numbers, plus the specific bone identifiers proposed by
Osborn and Mook (1921) when necessary to refer to single bones or discuss affiliation to
individual skeletons (e.g. H.1, P.3, P.6).

**Conservation history:** The two holotypic dorsal vertebrae of *Amphicoelias altus* are
incomplete, and parts of them are heavily restored. Much of the restoration and repair
likely stems from the original excavation in Colorado and preparation in Philadelphia, but
the subsequent move from Philadelphia to New York, and further study (including ours) has
resulted in additional damage. Mineralization of these fossil vertebrae seems to be very
extensive, meaning that the bones are heavy, and manipulation easily results in damage to
the delicate vertebral laminae and processes.

In an attempt to better identify reconstructed parts and repaired breaks, we used
Progressive Photonics (Eklund et al., 2018) to document the vertebrae in anterior and right
lateral views (electronic supplementary material). Progressive Photonics is a workflow to
document an object under various lighting conditions and wavelengths, including frontal
and oblique lighting, polarized light, and UV stimulation (Eklund et al., 2018). UV
stimulation, in particular, produces different reactions that can help to distinguish real fossil
bone from various materials used in repair and restoration. Whereas distinct materials for
repairs could be easily distinguished from each other, distinction between restored and real
fossil bone was more difficult. At least three different adhesives were used to repair breaks,
in some cases at the same fracture, indicating that the vertebra broke several times at the
same location (e.g. at the junction of the neural arch with the centrum in the posterior
dorsal vertebra [Fig. 5]). Restoration of the vertebrae was likely performed early in their
conservation history, because restored parts and original bone react in a very similar way to
UV stimulation, producing a blueish hue (Fig. 5). A similar blueish hue was identified as
shellac coating in other historic fossil material from Wyoming (Tschopp et al., in press),
which was commonly used as a consolidant in the field and in fossil preparation until at least
the 1930s (Linares Soriano and Carrascosa, Moliner 2016). This coating must have been
applied over the entire restored vertebrae, thereby obscuring the distinction between
original bone and the restored portions under UV stimulation. At the anterior-most portion
of the right surface of the centrum of the posterior dorsal vertebra, the original shellac
seems to have flaked off, revealing the actual, lighter colour of the bone, which does react
slightly differently to UV stimulation (Fig. 5).

**3. Description and comparisons of *Amphicoelias altus***

**3.1. Middle–posterior dorsal vertebra**

Nomenclature for vertebral laminae and fossae follows Wilson (1999) and Wilson et al.
(2011), respectively. An anteriorly and posteriorly incomplete centrum, most of the neural
arch, and the base of the neural spine of a middle–posterior dorsal vertebra is preserved
(Figs 1, 5; see Table 2 for measurements). Despite being reconstructed with a bifid neural

spine in Osborn and Mook (1921: fig. 119), not enough of the neural spine is preserved to be
able to determine its morphology.

The ventral surface of the centrum has been slightly crushed. There is a weak, rounded
ridge close to the midline, although this does not form a distinct keel; otherwise, the ventral
surface is fairly flat centrally, becoming gently transversely convex towards the lateral
margins. There are no ventrolateral ridges or fossae. The lateral pneumatic foramen is fairly
consistent on each side of the centrum, but the margins of both are mostly reconstructed.
Each foramen is situated on the dorsal half of the vertebra, occupying approximately half of
the centrum length, with a slight anterior bias. As reconstructed, each foramen has an
elliptical outline, with its long axis oriented anteroposteriorly, although it is also fairly tall
dorsoventrally; this reconstruction appears fairly accurate. Each foramen is deep, leaving a
thin midline septum and, internally, it ramifies strongly dorsally and, especially ventrally, as
well as a small distance anteriorly and posteriorly. In this regard, *Amphicoelias* is similar to
most neosauropods, with the notable exceptions of dicraeosaurids and some titanosaurs, in
which the lateral excavations are shallow (Upchurch, 1998). There are no vertical ridges
inside the foramen, contrasting with the condition in several diplodocids (Mannion et al.,
2012; Tschopp et al., 2015), but there is a sub-horizontal ridge at the anteroventral corner,
which is not visible in lateral view – this is present on both sides, although is more
developed on the right side.

The sharp lateral margin of the centroprezygapophyseal lamina (CPRL) also forms the
anterior centroparapophyseal lamina (ACPL). A posterior centroparapophyseal lamina
(PCPL) is also present, although this only becomes visible towards the parapophysis,
situated on the lateral surface of the prezygapophysis. The posterior centrodiapophyseal
lamina (PCDL) is near-vertical, and forms the posterior margin to a deep parapophyseal
centrodiapophyseal fossa (PACDF) on the lateral surface of the upper part of the neural
arch, bounded anteroventrally by the PCPL, and dorsally by the gently posteroventrally
directed prezygodiapophyseal lamina (PRDL).

Although this region is heavily reconstructed, the anterior neural canal opening is clearly
set within a fossa, as in most eusauropods (Carballido et al., 2012). The prezygapophyseal
articular surfaces are gently convex mediolaterally, and there are well developed hypantral
surfaces. A weakly developed, horizontal interprezygapophyseal lamina (TPRL) is strongly U-
shaped in dorsal view. A TPRL separates the prezygapophyses in most eusauropods,
whereas their articular surfaces are confluent in rebbachisaurids (Apesteguía et al., 2010;
Wilson and Allain, 2015). The postzygapophyseal articular surfaces are gently concave,
suggesting that the morphology of the prezygapophyseal surfaces might be genuine.
Although there is no hyposphene, its absence is almost certainly a preservational artefact,
based on the ventral surface of the postzygapophyseal midline being clearly damaged. The
poor preservation in this region means that we also cannot determine whether a vertical
midline lamina extended between the posterior neural canal opening and the
postzygapophyseal complex.

The diapophysis projects laterally and is dorsally deflected, but it has been broken and
slightly deformed, and so this dorsal deflection might merely be artefactual. The
spinoprezygapophyseal laminae (SPRLs) extend from close to the medial margins of the
posterior ends of the prezygapophyses and merge close to the apex of the incomplete
neural spine, forming a narrow, anteriorly projecting prespinal lamina (PRSL). Ventral to
their junction, there is no evidence for a PRSL. The base of the subvertical
spinodiapophyseal lamina (SPDL) is preserved, and fully extends down to the diapophysis. A

prezygapophyseal spinodiapophyseal fossa (PRSDF) is formed anterior to the SPDL, bounded ventrally by the PRDL, and anterodorsally by the SPRL. The spinopostzygapophyseal lamina (SPOL) is bifurcated a short distance above the postzygapophyses, with a large spinopostzygapophyseal lamina fossa (SPOL-F) in between the medial (mSPOL) and lateral (lSPOL) branches. A bifid SPOL is common across an array of eusauropod lineages (Wilson, 2002; Mannion et al., 2013), and *Amphicoelias* differs from some rebbachisaurids in which the SPOL is divided throughout its entire length (Whitlock, 2011a; Mannion et al., 2019). The mSPOL is directed dorsomedially along its preserved basal portion, with no midline postspinal lamina (POSL) within the postspinal fossa. As such, the dorsal vertebrae of *Amphicoelias* differ from those of most diplodocimorphs, in which pre- and postspinal laminae are present throughout nearly the full length of the neural spine, and are often not formed solely by convergence of the SPRLs or SPOLs (Wilson, 1999; Whitlock, 2011a; Mannion et al., 2019). The dorsal neural spines of *Haplocanthosaurus priscus* (Hatcher, 1903b) have a similar laminae configuration to *Amphicoelias*. The lSPOL and SPDL merge a short distance from the dorsalmost tip of the preserved neural spine. A postzygapophyseal spinodiapophyseal fossa (POSDF) is formed by the SPDL, the sub-horizontal postzygodiapophyseal lamina (PODL), and the anterodorsally directed lSPOL.

3.2. Posterior dorsal vertebra

A relatively complete dorsal vertebra, slightly posterior in the vertebral series relative to the aforementioned vertebra, is preserved (Figs 2, 5; see Table 2 for measurements). Breakages do not indicate any clear signs of internal camellae, contrasting with the tissue structure of the presacral vertebrae of titanosauriforms (Wilson, 2002).

The centrum is slightly dorsoventrally taller than wide (note that the anterior surface is incomplete along its left side, and the posterior surface along its right side). The anterior articular surface of the centrum is irregular, but is predominantly flat. Although it does not form a distinct condyle, a dorsally restricted convexity is present, which results in a 'slightly opisthocoeleous' centrum (following Carballido et al., 2012; Tschopp et al., 2015). By contrast, the posterior articular surface of the centrum is deeply concave. The ventral surface of the centrum is gently convex transversely, but flattens out along its midline. There are no ventrolateral or ventral midline ridges, nor are there any ventral fossae or excavations.

The lateral pneumatic foramen is a deep structure that leaves a thin midline septum. It is restricted to the dorsal third of the centrum and, internally, it ramifies dorsally and ventrally. It is not possible to determine whether there were internal ridges. The morphology of the lateral pneumatic foramen appears to be very different on either side. On the left side, it is a large opening that extends for most of the centrum length; however, all but the ventral margin of this opening is damaged. On the right side, the foramen is much shorter anteroposteriorly (with a strong anterior bias) and is set within a shallow, circular fossa. A fossa is clearly absent on the left side. The lateral surface of the centrum, ventral to the pneumatic foramen, is gently concave anteroposteriorly and fairly flat dorsoventrally (although this surface has been slightly broken, displaced and deformed on the left side).

The neural arch extends for the full anteroposterior length of the centrum. The anterior neural canal opening is set within a dorsoventrally tall fossa. At the base of the neural arch, each CPRL also forms the ACPL, and comprises a sharp ridge that is directed primarily vertically. At approximately one-third of its length, this combined CPRL-ACPL divides into a

widely rounded, but distinct, CPRL that extends dorsomedially to contact the ventral edge of
the hypantrum, and a narrow, sharp ACPL that connects to the ventral corner of the
parapophysis. The CPRL is neither bifurcated, nor is its anterior surface excavated,
contrasting with the morphology of several diplodocids and rebbachisaurids (Upchurch,
1995; Carballido et al., 2012; Tschopp et al., 2015). There is no evidence for an anterior

[revised manuscript text omitted]
 1.5 (Mannion et al., 2012, 2013). Although several diplodocoids have low ratios (e.g. *Tornieria* = 1.3; Mannion et al., 2012; Tschopp et al., 2015), these do not approach the subcircular morphology that we therefore regard as autapomorphic for the femur of *Amphicoelias*.

The femur has an anteroposteriorly thick shaft relative to that of the distal end (ratio > 0.6) (Whitlock, 2011a). This is comparable to some 'basal' macronarians (e.g. *Camarasaurus*), several rebbachisaurids, and the dicraeosaurid *Amargasaurus*, but most other sauropods (including diplodocids) have much lower ratios (Whitlock, 2011a; Tschopp et al., 2015; Mannion et al., 2019). Distally, the femur is bevelled, with the fibular condyle extending further distally than the tibial condyle. This type of bevelling is generally restricted to derived titanosaurs (Wilson, 2002), and is therefore regarded as a local

[revised manuscript text omitted]

The scapula (AMNH FARB 5764a [Sc. 7]) is preserved in four pieces (Fig. 7). It lacks part of
the acromion, the dorsal edge of most of the distal blade, and part of the end of the blade.
The short acromial ridge is at an obtuse angle to the long axis of the distal blade. Anterior to
this ridge, the acromion expands dorsoventrally. The preserved dorsal margin of the
acromion therefore has a somewhat sinuous shape in lateral view, which is an unusual
condition in sauropods. The glenoid is strongly expanded mediolaterally, and slightly
medially bevelled. The latter feature is primarily restricted to somphospondylans, but has
also been noted in the scapula of *Apatosaurus* (Wilson, 2002). There is a rugosity on the
medial surface of the blade, close to the acromion. The proximal part of the scapular blade
is D-shaped in cross-section, with a flat medial surface and dorsoventrally convex lateral
surface, as is the case in most eusauropods (Wilson, 2002). Although the distal-most portion
of the blade is not preserved, a small piece indicates that the blade expanded dorsally at its
distal end, whereas no ventral expansion is indicated along that edge, as preserved.
Although the bevelled glenoid could indicate referral to *Apatosaurus*, we conservatively
regard the AMNH FARB 5764a scapula as belonging to an indeterminate eusauropod.

The coracoid (AMNH FARB 5764a [Cor. 3]) is nearly complete, lacking only a small portion
of its rounded anterodorsal margin (Fig. 7). Based on its size, it is possible that is from the
same individual as the scapula (Osborn and Mook, 1921). The coracoid is slightly
dorsoventrally taller than anteroposteriorly long and has a straight articular surface for the
scapula. The glenoid is strongly expanded mediolaterally, which is unusual for diplodocids
(Tschopp et al., 2019). The articular surface of the glenoid extends onto the lateral surface
of the coracoid, and is anteroventrally bound by a distinct, relatively short, U-shaped notch.
Anterior to the notch, at the ventral end of the anterior margin, there is some broken bone
surface indicating the possible presence of a glenoid lip. The coracoid foramen is situated
slightly dorsal to the glenoid, extending dorsomedially and a little posteriorly through the
coracoid. We consider the AMNH FARB 5764a coracoid as representing an indeterminate
eusauropod.

The ulna (AMNH FARB 5764a [Ul.1]) is preserved in one piece (Fig. 8). Its proximal and
especially distal articular surfaces are partly eroded, but still indicate their general, original
shape. The proximal articular surface of the ulna has the typical triradiate outline of
sauropods. Its anterior and anterolateral proximal rami are approximately equidimensional,
forming a slightly obtuse angle. The posterior ramus does not form a distinct projection
dorsally (there is no well-defined olecranon process) or posteriorly (the ulna is V-shaped,
rather than Y-shaped, in proximal view). A transversely broad ridge extends distally from the
posterior margin. Around midheight, the medial margin of this ridge becomes less distinct,
such that the transition between the medial and posterior surfaces is rounded. However,
the lateral margin of this ridge expands at approximately midheight, continuing distally for
approximately another quarter of the length of the bone, before it also disappears. On the
anterior surface, the scar for the articulation with the radius is only weakly developed, or

possibly also slightly eroded. The distal articular surface is transversely compressed and expanded posteriorly. As with the scapula and coracoid, given the lack of anatomical overlap with the *Amphicoelias altus* holotype, and the absence of synapomorphies of other taxa, we regard the AMNH FARB 5764a ulna as an indeterminate eusauropod.

The distal half of the femur (AMNH FARB 5764a [Fem. 2]) preserves the distal-most extension of the fourth trochanter (Fig. 9), which was not recognized by Osborn and Mook (1921). The location of this feature demonstrates that the femur is a right element, rather than a left as originally identified, and it is likely that the femur is proximodistally much shorter than indicated in the reconstruction in Osborn and Mook (1921: fig. 126). The preserved distal end of the fourth trochanter is restricted to the medial margin of the posterior surface of the shaft and has a convex medial surface. Although slender, the shaft below the fourth trochanter lacks the subcircular outline (Table 4) that characterises the holotypic femur of *Amphicoelias altus*. The anterior parts of the distal condyles are damaged, likely during excavation. Whereas parts of the distal articular surface are preserved, the posterior extension of the condyles is damaged as well. The distal articular condyles are moderately expanded transversely, and there is no indication of a well-developed epicondyle (although this might be affected by damage). The tibial condyle projects slightly further distally than the fibular condyle, which is the reverse of the condition in the holotypic femur of *Amphicoelias altus*. As such, this distal femur clearly differs from that of *Amphicoelias altus* and should be regarded as an indeterminate eusauropod.

4.2. Additional AMNH FARB material catalogued as *Amphicoelias altus*

Two pubes and a humerus from the Morrison Formation of the Garden Park area have been tentatively identified as *Amphicoelias altus* in the AMNH collections. These identifications were presumably based on locality and/or gross morphology.

A left (?) pubis (AMNH FARB 5761 [Pb. 6]; Fig. 10) has no associated information concerning locality, other than being from Cope's Garden Park collection. It is associated with a note in the collections, which seems to identify this bone as AMNH FARB 30012, but this number was erroneously attributed to this pubis and should be ignored (see 'Curatorial history'). The pubis is in two pieces, and the medial side is not preserved, including the articular surface for its counterpart. An anonymous hand-written note in the collections indicates that its referral to *Amphicoelias altus* was based on its similarity in slenderness to the holotypic pubis. However, as in the holotypic pubis of *Amphicoelias altus*, there is little that can be observed concerning informative anatomy, and we regard this specimen as belonging to an indeterminate sauropod.

The second pubis (AMNH FARB 5760 [Pb. 3]) preserves only the proximal half, and there is also no further locality information associated with this specimen. No significant morphological information can be gleaned that could enable the identification of this bone to any sauropod taxon. Combined with the lack of provenance information, there is no basis for its referral to *Amphicoelias* and we consider it to represent an indeterminate eusauropod.

A right humerus (AMNH FARB 5761 [H. 1]; Fig. 11; see Table 4 for measurements) was apparently found at the same site as the *Amphicoelias altus* holotypic material (McIntosh, 1998). As with the Pb. 6 pubis, this humerus has also been associated with a new specimen number (AMNH FARB 30011), which should be ignored (see 'Curatorial history'). The

humerus is fairly complete, lacking only the proximomedial and proximolateral corners. It is
relatively stout (Robustness Index of 0.3), and is symmetrically expanded proximally,
indicating that it is probably referable to Diplodocoidea (Tschopp et al., 2015). The
deltopectoral crest has a distinct distal end that extends to approximately 42% of the total
proximodistal length of the humerus. Its lateral surface is anteroposteriorly concave and
posteriorly accompanied by a distinct, striated ridge that extends for a little more than the
proximal half of the deltopectoral crest. The tubercle for the attachment of the M.
coracobrachialis is situated in the centre of the anterior concavity of the proximal end, as in
most neosauropods (Mannion et al., 2019), and differs from the medially displaced tubercle
that characterises the diplodocine *Galeamopus pabsti* (Tschopp and Mateus, 2017). At
midshaft, the humerus has an elliptical cross-section, with the mediolateral diameter 1.7
14 times greater than its anteroposterior dimension. The distal articular surface is associated
with a relatively distinct intercondylar groove on the posterior surface of the shaft. Although
damaged, the medial and lateral ridges on the anterior surface of the distal end would have
been located close to the midline of the articular surface. The ratio of the length of this
humerus to that of the type femur of *Amphicoelias* is 0.64 (McIntosh, 1998), which is
consistent with the ratios in most diplodocoids, but is much lower than other eusauropods
(McIntosh, 1990; Wilson, 2002; Upchurch et al., 2004a; Tschopp et al., 2015; Mannion et al.,
2019). However, McIntosh (1998) considered it unlikely that this robust humerus could have
belonged to *Amphicoelias*, with its slender femur. Presumably based on its symmetrically
expanded proximal portion and the relative robusticity, McIntosh (1998) referred this
humerus to *Apatosaurus* instead. Here, we refer this humerus to Diplodocoidea, and note
the possibility that it could belong to *Amphicoelias*.

**4.3. ‘*Amphicoelias latus*’ – AMNH FARB 5765**

In the same publication in which he named *Amphicoelias altus*, Cope (1877a) erected a
second species – ‘*Amphicoelias latus*’ – on the basis of four caudal vertebrae and a femur
(AMNH FARB 5765) from a nearby locality (Cope Quarry XV). The femur of this specimen is
much more robust than that of *Amphicoelias altus*, and also has an anteroposteriorly
compressed, elliptical midshaft cross-section. Osborn and Mook (1921) noted this difference
and regarded ‘*Amphicoelias latus*’ as a junior synonym of *Camarasaurus supremus*. This
referral has been followed by subsequent authors (e.g. McIntosh, 1990, 1998; Upchurch et
al., 2004a), and was supported through phylogenetic analysis (Tschopp et al., 2015).
Woodruff and Foster (2014) incorrectly stated that previous authors (i.e. Osborn and Mook,
1921; McIntosh, 1998) had synonymized ‘*Amphicoelias latus*’ with *Amphicoelias altus*, a
taxonomic assignment they ‘agreed’ with. However, there is no basis for such a referral and
we agree with other workers that ‘*Amphicoelias latus*’ is a junior synonym of *Camarasaurus*
*supremus*.

**4.4. *Maraapunisaurus* (‘*Amphicoelias*’) *fragillimus***

Cope (1878) named a third species of *Amphicoelias* from a nearby locality (Cope Quarry
III) the following year. Based only on a middle–posterior dorsal neural arch (AMNH FARB
5777), Cope (1878) erected *Amphicoelias fragillimus*. This specimen was unfortunately lost
(or destroyed) but, despite this, has been the focus of several studies because of its
potentially gigantic size (Carpenter, 2006, 2018; Woodruff and Foster, 2014). Most authors

have synonymised it with *Amphicoelias altus* (e.g. Osborn and Mook, 1921; McIntosh, 1990;
Upchurch et al., 2004a; Woodruff and Foster, 2014), or listed it as a nomen dubium (e.g.
Tschopp et al., 2015 [note that these authors did not provide a diagnosis for *Amphicoelias*
*fragillimus*, as incorrectly claimed by Woodruff, 2019]). However, Carpenter (2018) provided
a novel reinterpretation in which he considered *Amphicoelias fragillimus* as a
rebbachisaurid, erecting the new genus *Maraapunisaurus*. All that remains of
*Maraapunisaurus* ('*Amphicoelias*') *fragillimus* is the drawing of the neural arch in posterior
view, as presented by Cope (1878). Based on this, we agree with Carpenter (2018) that it
differs in a number of anatomical features from *Amphicoelias altus* (e.g. the morphology of
the SPOLs, and the presence of a distinct postspinal lamina), and tentatively concur that
*Maraapunisaurus* might represent a rebbachisaurid.

**4.5. MOR 592**

MOR 592 is a skeleton from the Morrison Formation of Montana that consists of a
braincase, partial dentary, 12 presacral and seven caudal vertebrae, a pelvis, and femur. In a
conference abstract, Wilson and Smith (1996) suggested that MOR 592 might be referable
to *Amphicoelias*, based on the slenderness of the femur, a subcircular femoral cross-section
at midshaft, and reduced pleurocentral openings in the posterior dorsal centra. Those
authors noted that the phylogenetic position of the material was unstable, and the
assignment to *Amphicoelias* was regarded as tentative (J. A. Wilson Mantilla pers. comm. in
Whitlock, 2011a). Whitlock (2011a) revisited this material and considered it to potentially
represent a dicraeosaurid, based on the sharp supraoccipital crest and a symphyseal
tuberosity on the dentary, a diagnosis followed by some later work (e.g. Wedel and Taylor,
2013). However, based on the postcrania, Woodruff and Fowler (2012) suggested that MOR
592 was instead a juvenile morphotype of a diplodocine. Woodruff and Foster (2014) and
Woodruff et al. (2017, 2018) later went further and considered MOR 592 to be a juvenile
specimen of *Diplodocus*.

Regardless of whether MOR 592 is considered a dicraeosaurid or a juvenile *Diplodocus*, it
is generally agreed that it does not belong to *Amphicoelias*. However, the femur of MOR 592
is apparently characterised by a subcircular cross-section at midshaft (Wilson and Smith,
1996), and its distal end appears to be slightly bevelled, with the fibular condyle extending
further distally than its tibial counterpart (Woodruff et al., 2017: fig. 15), both of which are
herein regarded as potential autapomorphies of *Amphicoelias*. Photographs of the femur of
MOR 592, provided by C. Woodruff, reveal that the ratio of the mediolateral to
anteroposterior diameters of its midshaft is >1.3. As such, the femur of MOR 592 lacks the
circular cross-section that characterises that of *Amphicoelias*. The distal bevelling in MOR
592 is less pronounced than in the type of *Amphicoelias*. Furthermore, it is possible that
both the shape of the midshaft and the distal end morphology have been affected by
crushing in MOR 592, with mediolateral compression apparent from the heavily cracked
anterior surface (Woodruff et al., 2017: fig. 15). For now, we exclude MOR 592 from
*Amphicoelias*, but this specimen is clearly in need of detailed study to determine its
taxonomic affinities.

**5. Discussion**

**5.1. Is *Amphicoelias altus* synonymous with *Diplodocus*?**

Our detailed redescription, combined with recent phylogenetic analyses (Rauhut et al., 2005; Whitlock, 2011a; Mannion et al., 2012, 2019; Tschopp et al., 2015; Tschopp and Mateus, 2017), supports *Amphicoelias altus* as a valid taxon, diagnosed by three autapomorphies. We cannot currently determine whether *Amphicoelias* was a 'basal' diplodocoid (e.g. Mannion et al., 2019) or an apatosaurine (e.g. Tschopp and Mateus, 2017), but it seems unlikely that it was a member of Diplodocinae. As such, we reject recent proposals that *Amphicoelias altus* might be synonymous with *Diplodocus* (Foster, 2003; Woodruff and Foster, 2014; Woodruff, 2019), which also means that we retain the latter as a valid genus.

Although we restrict the known record of *Amphicoelias* to a single individual, it is possible that it was more common than we currently recognise. Given that much of the collection efforts in the Morrison Formation began at the dawn of dinosaur paleontology, many elements were excavated and labelled as one of the handful of well-known sauropod taxa at the time, particularly *Diplodocus* and *Camarasaurus*. Recent years have witnessed the erection of multiple new taxa from material previously referred to existing species, including *Galeamopus* (Tschopp and Mateus, 2017), *Kaatedocus* (Tschopp and Mateus, 2013), and *Smitanosaurus* (Whitlock and Wilson Mantilla, in press), and re-evaluation of historic material has shown that some specimens previously referred to *Diplodocus* and *Apatosaurus* might represent *Galeamopus* instead (Tschopp et al., 2019). It is therefore possible that material pertaining to *Amphicoelias* awaits a collections visitor in a cabinet labelled *Diplodocus*.

5.2. Sauropod diversity in the Morrison Formation

Recent taxonomic studies support the validity of 24 sauropod species assigned to 14 genera in the Upper Jurassic Morrison Formation (Upchurch et al., 2004a,b; Tschopp et al., 2015, 2017; Carpenter, 2018; Whitlock and Wilson Mantilla, in press; Table 1). In a series of papers, Woodruff and colleagues have argued that this diversity is overestimated (see also Prothero, 2019), and that many species considered valid in recent studies are synonymous, primarily interpreted as semaphoronts or 'ontogimorphs' (i.e. growth series) of a smaller number of valid taxa (Woodruff and Fowler, 2012; Woodruff and Foster, 2014; Woodruff et al. 2017, 2018; Woodruff, 2019). These putative synonyms include *Amphicoelias altus*, *Barosaurus lentus*, *Haplocanthosaurus priscus*, *Kaatedocus siberi*, and *Suuwassea emilieae*. In addition, Woodruff and colleagues considered it unlikely that this high diversity of megaherbivores could have all coexisted in one geographic region. Below we evaluate these claims that sauropod diversity in the Morrison Formation is inflated.

5.2.1. Are some Morrison sauropod species ontogimorphs?

There is a global dearth of sauropod remains for which we have a growth series that can be unambiguously referred to a single species, making it difficult to test hypotheses pertaining to taxonomic inflation resulting from ontogeny. The best-known test case of ontogenetic variation in a sauropod is the dwarf macronarian *Europasaurus holgeri* from the Late Jurassic of Germany, which is represented by more than a dozen individuals from one locality, with remains representing juvenile through to adult stages of growth (Sander et al. 2006). Carballido and Sander (2014) demonstrated that most anatomical features, including

autapomorphies, were present by the late immature stage of growth, and that these
provide the same phylogenetic signal as adult individuals. Although only based on one
species, and a potentially unusual one because of its dwarf status, *Europasaurus* suggests
that we might expect that sauropod anatomy does not change dramatically once individuals
reach subadult growth stages, and that autapomorphies will neither develop nor be lost
beyond that stage. Similar inferences can be made from recent descriptions of juvenile
material referred to the monospecific genus *Barosaurus* (Melstrom et al., 2016; Hanik et al.,
2017), where clearly defined autapomorphies of the taxon appear early in osteological
development.

The most robust test of this hypothesis in the Morrison Formation comes from Ikejiri et
al. (2005). Two adult individuals of *Camarasaurus lentus* at different osteological stages
were found in one quarry and compared with two subadult specimens from elsewhere in
the formation. Although Ikejiri et al. (2005) noted several anatomical differences between
the two adult individuals, these were typical of the features associated with advanced age
or repetitive stress (e.g. heterotopic ossification; Meyers et al., 2019). All four semaphoronts
in the study were clearly referable to the same species (and distinguishable from other
species) based on autapomorphies, as well as the overwhelming majority of anatomical
features. The limited evidence from *Camarasaurus* and *Europasaurus* suggests that
sauropods did not radically change their anatomy once they reached mature stages of
growth, undermining claims that species such as *Barosaurus lentus* and *Amphicoelias altus*,
known from a  remains, represent growth stages in *Diplodocus* ontogeny.

One consequence of this is that we might therefore expect that subadult sauropod
individuals should be unaffected by stemward slippage when included in phylogenetic
analyses, given that they should possess a character suite that is extremely close to that of
adult members of their species (e.g. Carballido and Sander, 2014). None of the type
specimens of Morrison Formation sauropod species regarded as potential ontogimorphs are
interpreted as juveniles (Wedel and Taylor, 2013; Hedrick et al., 2014), and it is ultimately
unlikely that many of them are truly subadult in a modern-day biological sense (e.g. Hone et
al., 2016). As such, we find little evidence to support the claim that species such as
*Haplocanthosaurus priscus* are immature individuals of other species when they are clearly
known from adult specimens (Wedel and Taylor, 2013). However, even allowing for the
possibility that some taxa are based on subadult material (e.g. *Kaatedocus*, *Suuwassea*), this
still necessitates major increases in the number of evolutionary steps for phylogenetic
analyses to recover them clustering with their proposed senior synonyms (Wedel and
Taylor, 2013). Furthermore, even the inclusion of juvenile specimens does not necessarily
result in their stemward slippage. In their diplodocid-focused phylogenetic analysis, Tschopp
et al. (2015) recovered the juvenile type specimen of *Brontosaurus* ('*Elosaurus*') *parvus* as
most closely related to the adult apatosaurine specimen found in association (Peterson and
Gilmore, 1902; Gilmore, 1936), despite the presence of several plesiomorphic features that
resulted in character scores variation between the two individuals (see also Upchurch et al.,
2004b). As such, there is no evidence that the phylogenetic positions of taxa such as
*Amphicoelias*, *Barosaurus*, *Haplocanthosaurus*, *Kaatedocus*, or *Suuwassea*, are incorrect to
the point that they have been misidentified as distinct from *Diplodocus* or other Morrison
taxa. It is important to note that whether apatosaurine species belong to *Apatosaurus* or
*Brontosaurus* is ultimately irrelevant to discussions of diversity (cf. Prothero, 2019), given
the arbitrary nature of genera.

The above does not preclude the possibility that some phylogenetically informative
features can have an ontogenetic signal too. For example, diplodocine individuals might
possess a postparietal foramen that is lost during ontogeny (Woodruff et al., 2017) *and* its
presence is also a synapomorphy of some dicraeosaurids (Whitlock, 2011a). Similarly,
ontogenetically old individuals of *Camarasaurus* might incorporate a sixth sacral vertebra
(e.g. Tidwell et al., 2005) *and* a 6-vertebra sacrum also characterises Somphospondyli
(Wilson and Sereno, 1998). However, although these ontogenetic features might introduce
some degree of 'directed noise' (Tschopp and Upchurch, 2019), they do not outweigh the
overwhelming phylogenetic signal when operational taxonomic units are not early juveniles,
at least in sauropods. Returning to *Europasaurus*, this taxon is universally recovered as an
early diverging macronarian (e.g. Sander et al., 2006; D'Emic, 2012; Carballido and Sander,
2014; Mannion et al., 2019). However, likely via pedomorphic retention, *Europasaurus*
possesses several features (including a postparietal foramen) that are inconsistent with a
placement in Macronaria or even Neosauropoda (Carballido and Sander, 2014; Marpmann
et al., 2015), but these do not result in stemward slippage (Mannion et al., 2017). We do not
disagree that ontogeny is an important issue and one that needs to be considered, but there
is currently no evidence to support the claim that Morrison sauropod diversity is
overestimated as a result of it. As also concluded by Wedel and Taylor (2013), we should be
wary of placing too much emphasis on 'critical' phylogenetic characters (*sensu* Woodruff
and Fowler, 2012).

24 25 26 27 28 29 **5.2.2. Was apparent Morrison Formation sauropod diversity too high to be ecologically** 30 **viable?**

The notion of 24 co-occurring megaherbivore species is, at face value, potentially difficult
to accept, with Woodruff (2019: pp. 93, 105) considering it "unlikely" and "ecologically
taxing". Although the Morrison Formation covers an area exceeding 1.2 million km² (Dodson
et al., 1980), it is possible that this would have been unable to support enough viable
individuals of each species (though see below). However, the idea that all of these species
co-occurred with one another is misleading. Firstly, none of these 24 species are found
throughout the spatial extent of the Morrison Formation. As noted above, many species are
limited to a small number (<5) of occurrences (e.g. *Brachiosaurus altithorax*,
*Haplocanthosaurus priscus*) or are currently known from a single locality (e.g.
*Haplocanthosaurus delfsi*, *Suuwassea emilieae*). However, even species known from
abundant remains show evidence for some degree of geographical restriction (e.g. Foster,
2003; Whitlock et al., 2018; Tschopp et al., 2019). Furthermore, numerous fossiliferous
Morrison localities have been sampled extensively and yet none contain more than five
sympatric sauropod species (Foster, 2003). These distributional data alone suggest that the
24 Morrison sauropod species did not all co-occur.

Secondly, the Morrison Formation was deposited over a period of at least seven million
48 years (Turner and Peterson, 1999; Trujillo and Kowallis, 2015; Maidment and Muxworthy,
2019). None of the 24 sauropod species are recovered throughout the formation's full
temporal extent and many species were clearly not contemporaneous with others (e.g.
Turner and Peterson, 1999; Foster, 2003; Tschopp et al., 2015), although a substantial
proportion of localities with radiometric dates are approximately coeval (Whitlock et al.,
2018). Despite framing his study with the 'problem' of having 24 contemporaneous
megaherbivore species to support his claim that Morrison sauropod diversity is

overestimated, Woodruff (2019: p. 99–105) was clearly aware that this is incorrect. For
example, Woodruff (2019: fig. 5) showed the Morrison Formation divided into six ‘biozones’,
in which the most diverse of these contains 12 distinct taxa (and note that three of these
are specifically indeterminate members of contemporaneous genera). It is not clear if this
number of species was also considered ecologically problematic by Woodruff (2019), but no
evidence was provided in that study (or others) to support the view that high sauropod
diversity was unviable.

[revised manuscript text omitted]

32 33 ACKNOWLEDGEMENTS

We thank Mark Norell for enabling us to study and 3D scan *Amphicoelias altus* at the AMNH,
as well as Carl Mehling, Verne Lee, Bruce Javors, and the late Jack Conrad for their help
accessing the material. We are also grateful to Mike Eklund and Bruce Javors for
contributing photographs of AMNH specimens, and to Carolyn Merrill for producing the 3D
scans. Cary Woodruff also kindly provided us with photographs of the femur of MOR 592.
We also acknowledge the Willi Hennig Society, which has sponsored the development and
free distribution of TNT. PDM.’s research was supported by a Royal Society University
Research Fellowship (UF160216). ET’s contribution was supported by an AMNH Division of
Paleontology Postdoctoral Fellowship.

48 REFERENCES

**Apesteguía S, Gallina PA, Haluza A. 2010.** Not just a pretty face: anatomical peculiarities in
the postcranium of rebbachisaurids (Sauropoda: Diplodocoidea). *Historical Biology* **22**:
165–174.
**Bakker RT. 1971.** Ecology of the brontosaurus. *Nature* **229**: 172–174.
**Barrett PM. 2014.** Paleobiology of herbivorous dinosaurs. *Annual Review of Earth and*
Planetary Sciences **42**: 207–30.
**Bonaparte JF. 1986.** The early radiation and phylogenetic relationships of the Jurassic
sauropod dinosaurs, based on vertebral anatomy. In: Padian K, ed. *The Beginning of the*
*Age of Dinosaurs*. Cambridge: Cambridge University Press, 247–258.

**Brochu CA, Sumrall CD. 2020.** Modern cryptic species and crocodylian diversity in the fossil
record. *Zoological Journal of the Linnean Society* (doi: 10.1093/zoolinnean/zlaa039).
- **Button DJ, Barrett PM, Rayfield EJ. 2017.** Craniodental functional evolution in
sauropodomorph dinosaurs. *Paleobiology* **43**: 435–462.
- **Button DJ, Zanno LE. 2020.** Repeated Evolution of Divergent Modes of Herbivory in Non-
avian Dinosaurs. *Current Biology* **30**: 158–168.
- **Calvo JO, Salgado L. 1995.** *Rebbachisaurus tessonei* sp. nov. a new Sauropoda from the
Albian-Cenomanian of Argentina; new evidence on the origin of the Diplodocidae. *Gaia* **11**:
13–33.
- **Carballido JL, Salgado L, Pol D, Canudo JI, Garrido A. 2012.** A new basal rebbachisaurid
(Sauropoda, Diplodocoidea) from the Early Cretaceous of the Neuquén Basin; evolution
and biogeography of the group. *Historical Biology* **24**: 631–654.
- **Carballido JL, Sander PM. 2014.** Postcranial axial skeleton of *Europasaurus holgeri*
(Dinosauria, Sauropoda) from the Upper Jurassic of Germany—implications for sauropod
ontogeny and phylogenetic relationships of basal Macronaria. *Journal of Systematic
Palaeontology* **12**: 335–387.
- **Carpenter K. 1998.** Vertebrate biostratigraphy of the Morrison Formation near Cañon City,
Colorado. *Modern Geology* **23**: 407–426.
- **Carpenter K. 2006.** Biggest of the big—a critical re-evaluation of the megasauropod
*Amphicoelias fragillimus*. *New Mexico Museum of Natural History and Science Bulletin* **36**:
131–137.
- **Carpenter K. 2018.** *Maraapunisaurus fragillimus*, n.g. (formerly *Amphicoelias fragillimus*), a
basal rebbachisaurid from the Morrison Formation (Upper Jurassic) of Colorado. *Geology
of the Intermountain West* **5**: 227–244.
- **Chiarenza AA, Mannion PD, Lunt DJ, Farnsworth A, Jones LA, Kelland S-J, Allison PA. 2019.**
Ecological niche modelling does not support climatically-driven dinosaur diversity decline
before the Cretaceous/Paleogene mass extinction. *Nature Communications* **10**: 1091.
- **Christiansen P. 2000.** Feeding mechanisms of the sauropod dinosaurs *Brachiosaurus*,
*Camarasaurus*, *Diplodocus*, and *Dicraeosaurus*. *Historical Biology* **14**: 137–152.
- **Cope ED. 1877a.** On *Amphicoelias*, a genus of Saurians from the Dakota epoch of Colorado.
*Proceedings of the American Philosophical Society* **17**: 242–246.
- **Cope ED. 1877b.** On a gigantic saurian from the Dakota epoch of Colorado. *Palaeontological
Bulletin* **25**: 5–10.
- **Cope ED. 1878.** A new species of *Amphicoelias*. *American Naturalist* **12**: 563–565.
- **D’Emic MD. 2012.** The early evolution of titanosauriform sauropod dinosaurs. *Zoological
Journal of the Linnean Society* **166**: 624–671.
- **D’Emic MD, Whitlock JA, Smith KM, Fisher DC, Wilson JA. 2013.** Evolution of high tooth
replacement rates in sauropod dinosaurs. *PLoS ONE* **8**: e69235.
- **Dodson P, Behrensmeyer AK, Bakker RT, Mcintosh JS. 1980.** Taphonomy and paleoecology
of the dinosaur beds of the Jurassic Morrison Formation. *Paleobiology* **6**: 208–232.
- **Eklund MJ, Aase AK, Bell CJ. 2018.** Progressive Photonics: Methods and applications of
sequential imaging using visible and non-visible spectra to enhance data-yield and
facilitate forensic interpretation of fossils. *Journal of Paleontological Techniques* **20**: 1–36.
- **Farlow JO, Coroian ID, Foster JR. 2010.** Giants on the landscape: modelling the abundance
of megaherbivorous dinosaurs of the Morrison Formation (Late Jurassic, western USA).
*Historical Biology* **22**: 403–429.

- Filla BJ, Redman PD. 1994.** *Apatosaurus yahnahpin*: a preliminary description of a new species of diplodocid dinosaur from the Late Jurassic Morrison Formation of Southern Wyoming, the first sauropod dinosaur found with a complete set of “belly ribs”. In: Nelson GE, ed. *The Dinosaurs of Wyoming. Wyoming Geological Association 44th annual field conference guidebook*. Casper: Wyoming Geological Association, 159–178.
- Fiorillo AR. 1998.** Dental microwear patterns of the sauropod dinosaurs *Camarasaurus* and *Diplodocus*—evidence for resource partitioning in the Late Jurassic of North America. *Historical Biology* **13**: 1–16.
- Foster JR. 2003.** Paleoecological analysis of the vertebrate fauna of the Morrison Formation (Upper Jurassic), Rocky Mountain region, U.S.A. *New Mexico Museum of Natural History and Science Bulletin* **23**: 1–95.
- Foster JR. 2007.** *Jurassic west—the dinosaurs of the Morrison Formation and their world*. Bloomington: Indiana University Press, 389 p.
- Gallina PA, Apesteguía S, Haluza A, Canale JI. 2014.** A diplodocid sauropod survivor from the Early Cretaceous of South America. *PLoS ONE* **9**: e97128.
- Gee CT. 2011.** Dietary options for the sauropod dinosaurs from an integrated botanical and paleobotanical perspective. In: Klein N, Remes K, Gee CT, Sander PM, eds. *Biology of the Sauropod Dinosaurs: Understanding the Life of Giants*. Bloomington, IN: Indiana University Press, 34–56.
- Gillette DG. 1991.** *Seismosaurus halli*, gen. et sp. nov., a new sauropod dinosaur from the Morrison Formation (Upper Jurassic/Lower Cretaceous) of New Mexico, USA. *Journal of Vertebrate Paleontology* **11**: 417–433.
- Gilmore CW. 1925.** A nearly complete articulated skeleton of *Camarasaurus*, a saurischian dinosaur from the Dinosaur National Monument, Utah. *Memoirs of the Carnegie Museum* **10**: 347–384.
- Gilmore CW. 1936.** Osteology of *Apatosaurus*: with special reference to specimens in the Carnegie Museum. *Memoirs of the Carnegie Museum* **11**: 175–300.
- Hanik GM, Lamanna MC, Whitlock JA. 2017.** A juvenile specimen of *Barosaurus* Marsh, 1890 (Sauropoda: Diplodocidae) from the Upper Jurassic Morrison Formation of Dinosaur National Monument, Utah, USA. *Annals of Carnegie Museum* **84**: 253–263.
- Harris JD, Dodson P. 2004.** A new diplodocoid sauropod dinosaur from the Upper Jurassic Morrison Formation of Montana, USA. *Acta Palaeontologica Polonica* **49**: 197–210.
- Hatcher JB. 1901.** *Diplodocus* Marsh, its osteology, taxonomy, and probable habits, with a restoration of the skeleton. *Memoirs of the Carnegie Museum* **1**: 1–64.
- Hatcher JB. 1903a.** A new sauropod dinosaur from the Jurassic of Colorado. *Proceedings of the Biological Society of Washington* **16**: 1–2.
- Hatcher JB. 1903b.** Osteology of *Haplocanthosaurus*, with description of a new species, and remarks on the probable habits of the Sauropoda and the age and origin of the *Atlantosaurus* beds. *Memoirs of the Carnegie Museum* **2**: 1–72.
- Hedrick BP, Tumarkin-Deratzian AR, Dodson P. 2014.** Bone microstructure and relative age of the holotype specimen of the diplodocoid sauropod dinosaur *Suuwassea emilieae*. *Acta Palaeontologica Polonica* **59**: 2295–304.
- Holland WJ. 1915.** A new species of *Apatosaurus*. *Annals of the Carnegie Museum* **10**: 143–145.
- Holland WJ. 1924.** The skull of *Diplodocus*. *Memoirs of the Carnegie Museum* **9**: 379–403.
- Hone DWE, Farke AA, Wedel MJ. 2016.** Ontogeny and the fossil record: what, if anything, is an adult dinosaur? *Biology Letters* **12**: 20150947.

- Hotton CL, Baghai-Riding NL. 2010.** Palynological evidence for conifer dominance within a heterogenous landscape in the Late Jurassic Morrison Formation, U.S.A. In: Gee CT, ed. *Plants in Mesozoic Time*. Bloomington, IN: Indiana University Press, 295–328.
- Huene F von. 1908.** Die Dinosaurier der Europäischen Triasformation mit Berücksichtigung der Ausseuropäischen Vorkommnisse. *Geologische und Palaeontologische Abhandlungen 1908 (supplement 1)*: 1–419.
- Ikejiri T, Tidwell V, Trexler DL. 2005.** New adult specimens of *Camarasaurus lentus* highlight ontogenetic variation within the species. In: Tidwell V, Carpenter K, eds. *Thunder Lizards: The Sauropodomorph Dinosaurs*. Bloomington, IN: Indiana University Press, 154–179.
- Jensen JA. 1985.** Three new sauropod dinosaurs from the Upper Jurassic of Colorado. *The Great Basin Naturalist* **45**: 697–709.
- Jensen JA. 1988.** A fourth new sauropod dinosaur from the Upper Jurassic of the Colorado Plateau and sauropod bipedalism. *The Great Basin Naturalist* **48**: 121–145.
- Linares Soriano MA, Carrascosa Moliner MB. 2016.** Consolidation of bone material: chromatic evolution of resins after UV accelerated aging. *Journal of Paleontological Techniques* **15**: 46–67.
- Lovelace DM, Hartman SA, Wahl WR. 2008.** Morphology of a specimen of *Supersaurus* (Dinosauria, Sauropoda) from the Morrison Formation of Wyoming, and a re-evaluation of diplodocid phylogeny. *Arquivos do Museu Nacional, Rio de Janeiro* **65**: 527–544.
- Maidment SCR, Muxworthy A. 2019.** A chronostratigraphic framework for the Upper Jurassic Morrison Formation, western U.S.A. *Journal of Sedimentary Research* **89**: 1017–1038.
- Mannion PD, Allain R, Moine O. 2017.** The earliest known titanosauriform sauropod dinosaur and the evolution of Brachiosauridae. *PeerJ* **5**: e3217.
- Mannion PD, Upchurch P, Barnes RN, Mateus O. 2013.** Osteology of the Late Jurassic Portuguese sauropod dinosaur *Lusotitan atalaiensis* (Macronaria) and the evolutionary history of basal titanosauriforms. *Zoological Journal of the Linnean Society* **168**: 98–206.
- Mannion PD, Upchurch P, Mateus O, Barnes RN, Jones MEH. 2012.** New information on the anatomy and systematic position of *Dinheirosaurus lourinhanensis* (Sauropoda: Diplodocoidea) from the Late Jurassic of Portugal, with a review of European diplodocoids. *Journal of Systematic Palaeontology* **10**: 521–551.
- Mannion PD, Upchurch P, Schwarz D, Wings O. 2019.** Taxonomic affinities of the putative titanosaurs from the Late Jurassic Tendaguru Formation of Tanzania: phylogenetic and biogeographic implications for eusauropod dinosaur evolution. *Zoological Journal of the Linnean Society* **185**: 784–909.
- Marpmann JS, Carballido JL, Sander PM, Knötschke N. 2015.** Cranial anatomy of the Late Jurassic dwarf sauropod *Europasaurus holgeri* (Dinosauria, Camarasauromorpha): ontogenetic changes and size dimorphism. *Journal of Systematic Palaeontology* **13**: 221–263.
- Marsh OC. 1877.** Notice of new dinosaurian reptiles from the Jurassic Formation. *American Journal of Science (Series 3)* **14**: 515–516.
- Marsh OC. 1878.** Principal characters of American Jurassic dinosaurs. Part I. *American Journal of Science (Series 3)* **16**: 411–416.
- Marsh OC. 1879.** Notice of new Jurassic dinosaurs. *American Journal of Science (Series 3)* **18**: 501–505.

**Marsh OC. 1884.** Principal characters of American Jurassic dinosaurs. Part VII. On the
Diplodocidae, a new family of the Sauropoda. *American Journal of Science (Series 3)* **27**:
161–167.
- **Marsh OC. 1889.** Comparison of the principal forms of the Dinosauria of Europe and
America. *American Journal of Science (Series 3)* **37**: 323–330.
- **Marsh OC. 1890.** Description of new dinosaurian reptiles. *American Journal of Science*
*(Series 3)* **39**: 81–86.
- **Marsh OC. 1896.** The Dinosaurs of North America. *United States Geological Survey* **55**: 133–
244.
- **McIntosh JS. 1990.** Sauropoda. In: Weishampel DB, Dodson P, Ósmolska H, eds. *The*
*Dinosauria, First edition*. Berkeley: University California Press, 345–401.
- **McIntosh JS. 1998.** New information about the Cope collection of sauropods from Garden
Park, Colorado. *Modern Geology* **23**: 481–506.
- **McIntosh JS, Coombs WP, Russell DA. 1992.** A new diplodocid sauropod (Dinosauria) from
Wyoming, U.S.A. *Journal of Vertebrate Paleontology* **12**: 158–167.
- **McIntosh JS, Williams ME. 1988.** A new species of sauropod dinosaur, *Haplocanthosaurus*
*delfsi* sp. nov., from the Upper Jurassic Morrison Fm. of Colorado. *Kirtlandia* **43**: 3–26.
- **Melstrom KM, D’Emic MD, Chure DJ, Wilson JA. 2015.** A juvenile sauropod dinosaur from
the Late Jurassic of Utah, USA, with evidence of an avian style air-sac system. *Journal of*
*Vertebrate Paleontology* **36**: e1111898.
- **Meyers C, Lisiecki J, Miller S, Levin A, Fayad L, Ding C, Sono T, McCarthy E, Levi B, James**
**AW. 2019.** Heterotopic ossification: a comprehensive review. *JBMR Plus* **3**: e10172.
- **Mook CC. 1917.** Criteria for the determination of species in the Sauropoda, with description
of a new species of *Apatosaurus*. *Bulletin of the American Museum of Natural History* **37**:
355–358.
- **Nopcsa F. 1928.** The genera of reptiles. *Palaeobiologica* **1**: 163–188.
- **Osborn HF, Mook CC. 1921.** *Camarasaurus, Amphicoelias*, and other sauropods of Cope.
*Memoirs of the American Museum of Natural History New Series* **3**: 247–387.
- **Otero A. 2010.** The appendicular skeleton of *Neuquensaurus*, a Late Cretaceous saltasaurine
sauropod from Patagonia, Argentina. *Acta Palaeontologica Polonica* **55**: 399–426.
- **Peterson OA, Gilmore CW. 1902.** *Elosaurus parvus*; a new genus and species of the
Sauropoda. *Annals of the Carnegie Museum* **1**: 490–499.
- **Plotnick RE, Smith FE, Lyons SK. 2016.** The fossil record of the sixth extinction. *Ecology*
*Letters* **19**: 546–553.
- **Prothero DR. 2019.** *The Story of the Dinosaurs in 25 Discoveries*. New York: Columbia
University Press, 488 pp.
- **Riggs ES. 1903.** *Brachiosaurus altithorax*, the largest known dinosaur. *American Journal of*
*Science (Series 4)* **15**: 299–306.
- **Salgado L, Coria RA, Calvo JO. 1997.** Evolution of titanosaurid sauropods. I: phylogenetic
analysis based on the postcranial evidence. *Ameghiniana* **34**: 3–32.
- **Sander PM, Mateus O, Laven T, Knötschke N. 2006.** Bone histology indicates insular
dwarfism in a new Late Jurassic sauropod dinosaur. *Nature* **441**: 739–741.
- **Stevens KA, Parrish JM. 2005.** Neck posture, dentition, and feeding strategies in Jurassic
sauropod dinosaurs. In: Tidwell V, Carpenter K, eds. *Thunder Lizards: The Sauropodomorph*
*Dinosaurs*. Bloomington, IN: Indiana University Press, 212–232.

**Tidwell V, Stadtman K, Shaw A. 2005.** Age-related characteristics found in a partial pelvis of
*Camarasaurus*. In: Tidwell V, Carpenter K, eds. *Thunder Lizards: The Sauropodomorph*
*Dinosaurs*. Bloomington, IN: Indiana University Press, 180–186.
- **Trujillo KC, Kowallis BJ. 2015.** Recalibrated legacy $^{40}\text{Ar}/^{39}\text{Ar}$ ages for the Upper Jurassic
Morrison Formation, Western Interior, U.S.A. *Geology of the Intermountain West* **2**: 1–8.
- **Tschopp E, Brinkman D, Henderson J, Turner M, Mateus O. 2018.** Considerations on the
replacement of a type species in the case of the sauropod dinosaur *Diplodocus* Marsh,
1878. *Geology of the Intermountain West* **5**: 245–262.
- **Tschopp E, Maidment SCR, Lamanna MC, Norell MA. 2019.** Reassessment of a Historical
Collection of Sauropod Dinosaurs from the Northern Morrison Formation of Wyoming,
with Implications for Sauropod Biogeography. *Bulletin of the American Museum of Natural*
*History* **437**: 1–79.
- **Tschopp E, Mateus O. 2013.** The skull and neck of a new flagellicaudatan sauropod from the
Morrison Formation and its implication for the evolution and ontogeny of diplodocid
dinosaurs. *Journal of Systematic Palaeontology* **11**: 853–888.
- **Tschopp E, Mateus O. 2017.** Osteology of *Galeamopus pabsti* sp. nov. (Sauropoda:
Diplodocidae), with implications for neurocentral closure timing, and the cervico-dorsal
transition in diplodocids. *PeerJ* **5**: e3179.
- **Tschopp E, Mateus O, Benson RBJ. 2015.** A specimen-level phylogenetic analysis and
taxonomic revision of Diplodocidae (Dinosauria, Sauropoda). *PeerJ* **3**: e857.
- **Tschopp E, Mehling C, Norell MA. In Press.** Reconstructing the specimens and history of the
Howe Quarry (Upper Jurassic Morrison Formation; Wyoming, USA). *American Museum*
*Novitates*.
- **Tschopp E, Upchurch P. 2019.** The challenges and potential utility of phenotypic specimen-
level phylogeny based on maximum parsimony. *Earth and Environmental Science*
*Transactions of The Royal Society of Edinburgh* **109**: 301–323.
- **Turner CE, Peterson F. 1999.** Biostratigraphy of dinosaurs in the Upper Jurassic Morrison
Formation of the Western Interior, USA. *Utah Geological Survey Miscellaneous*
*Publications* **99**: 77–114.
- **Tütken T. 2011.** The diet of sauropod dinosaurs: implications of carbon isotope analysis on
teeth, bones, and plants. In: Klein N, Remes K, Gee CT, Sander PM, eds. *Biology of the*
*Sauropod Dinosaurs: Understanding the Life of Giants*. Bloomington, IN: Indiana University
Press, 57–79.
- **Upchurch P. 1998.** The phylogenetic relationships of sauropod dinosaurs. *Zoological Journal*
*of the Linnean Society* **124**: 43–103.
- **Upchurch P, Barrett PM. 2000.** The evolution of sauropod feeding mechanisms. In: Sues H-
D, ed. *Evolution of Herbivory in Terrestrial Vertebrates—Perspectives from the Fossil*
*Record*. Cambridge: Cambridge University Press, 79–122.
- **Upchurch P, Barrett PM, Dodson P. 2004a.** Sauropoda. In: Weishampel DB, Dodson P,
Osmólska H, eds. *The Dinosauria, Second edition*. Berkeley: University of California Press,
259–324.
- **Upchurch P, Tomida Y, Barrett PM 2004b.** A new specimen of *Apatosaurus ajax*
(Sauropoda: Diplodocidae) from the Morrison Formation (Upper Jurassic) of Wyoming,
USA. *National Science Museum Monographs* **26**: 1–118.
- **Wedel MJ, Taylor MP. 2013.** Neural spine bifurcation in sauropod dinosaurs of the Morrison
Formation: ontogenetic and phylogenetic implications. *PalArch's Journal of Vertebrate*
*Palaeontology* **10**: 1–34.

**Whitlock JA. 2011a.** A phylogenetic analysis of Diplodocoidea (Saurischia: Sauropoda).
*Zoological Journal of the Linnean Society* **161**: 872–915.
- **Whitlock JA. 2011b.** Inferences of diplodocoid (Sauropoda: Dinosauria) feeding behavior
from snout shape and microwear analyses. *PLoS ONE* **6**: e18304.
- **Whitlock JA, Trujillo KC, Hanik GM. 2018.** Assemblage-level structure in Morrison
Formation dinosaurs, Western Interior, USA. *Geology of the Intermountain West* **5**: 9–22.
- **Whitlock JA, Wilson Mantilla JA. In press.** The Late Jurassic sauropod dinosaur
“*Morosaurus*” *agilis* Marsh, 1889 reexamined and reinterpreted as a dicraeosaurid. *Journal*
*of Vertebrate Paleontology*.
- **Wiersma K, Sander PM. 2016.** The dentition of a well-preserved specimen of *Camarasaurus*
sp.: implications for function, tooth replacement, soft part reconstruction, and food intake.
*Paläontologische Zeitschrift* **91**: 145–161.
- **Wilson JA. 1999.** A nomenclature for vertebral laminae in sauropods and other saurischian
dinosaurs. *Journal of Vertebrate Paleontology* **19**: 639–653.
- **Wilson JA. 2002.** Sauropod dinosaur phylogeny: critique and cladistic analysis. *Zoological*
*Journal of the Linnean Society* **136**: 217–276.
- **Wilson JA, Allain R. 2015.** Osteology of *Rebbachisaurus garasbae* Lavocat, 1954, a
diplodocoid (Dinosauria, Sauropoda) from the early Late Cretaceous-aged Kem Kem beds
of southeastern Morocco. *Journal of Vertebrate Paleontology* **35**: e1000701.
- **Wilson JA, Sereno PC. 1998.** Early evolution and higher-level phylogeny of sauropod
dinosaurs. *Society of Vertebrate Paleontology Memoir* **5**: 1–68.
- **Wilson JA, Smith MB. 1996.** New remains of *Amphicoelias* Cope (Dinosauria: Sauropoda)
from the Upper Jurassic of Montana and diplodocoid phylogeny. *Journal of Vertebrate*
*Paleontology* **16**: 73A.
- **Wilson JA, D’Emic MD, Ikejiri T, Moacdieh EM, Whitlock JA. 2011.** A nomenclature for
vertebral fossae in sauropods and other saurischian dinosaurs. *PLoS ONE* **6**: e17114.
- **Woodruff DC. 2019.** What Factors Influence our Reconstructions of Morrison Formation
Sauropod Diversity? *Geology of the Intermountain West* **6**: 93–112.
- **Woodruff DC, Carr TD, Storrs GW, Waskow K, Scannella JB, Norden KK, Wilson JP. 2018.**
The smallest diplodocid skull reveals cranial ontogeny and growth-related dietary changes
in the largest dinosaurs. *Scientific Reports* **8**: 14341.
- **Woodruff DC, Foster JR. 2014.** The fragile legacy of *Amphicoelias fragillimus* (Dinosauria:
Sauropoda; Morrison Formation – latest Jurassic). *Volumina Jurassica* **12**: 211–220.
- **Woodruff DC, Fowler DW. 2012.** Ontogenetic influence on neural spine bifurcation in
Diplodocoidea (Dinosauria: Sauropoda): a critical phylogenetic character. *Journal of*
*Morphology* **273**: 754–764.
- **Woodruff DC, Fowler DW, Horner JR. 2017.** A new multi-faceted framework for deciphering
diplodocid ontogeny. *Palaeontologia Electronica* **20**: 43A.

TABLES

**Table 1.** Currently recognized sauropod genera and species in the Upper Jurassic Morrison
Formation of North America. *Dyslocosaurus polyonychius* potentially represents an
additional valid sauropod species, but its stratigraphic provenance remains uncertain
(McIntosh et al., 1992; Tschopp et al., 2015). ‘*Apatosaurus*’ *minimus* might also represent a
distinct sauropod species (Mook, 1917), although its affinities require further evaluation
(Upchurch et al., 2004a; Tschopp et al., 2015). Note that the validity of *Diplodocus longus*

(highlighted with an asterisk) is disputed and it is unlikely that this species can be diagnosed (Tschopp et al. 2018 [and references therein]).

Taxon	Authors
Amphicoelias altus	Cope, 1877a
Apatosaurus ajax (type)	Marsh, 1877
Apatosaurus louisae	Holland, 1915
Barosaurus lentus	Marsh, 1890
Brachiosaurus altithorax	Riggs, 1903
Brontosaurus excelsus (type)	Marsh, 1879
Brontosaurus parvus	Peterson and Gilmore, 1902
Brontosaurus yahnahpin	Filla and Redman, 1994
Camarasaurus grandis	Marsh, 1877
Camarasaurus lentus	Marsh, 1889
Camarasaurus lewisi	Jensen, 1988
Camarasaurus supremus	Cope, 1877b
Diplodocus carnegii	Hatcher, 1901
Diplodocus longus (type)*	Marsh, 1878
Diplodocus hallorum	Gillette, 1991
Galeamopus hayi (type)	Holland, 1924; Tschopp et al., 2015
Galeamopus pabsti	Tschopp and Mateus, 2017
Haplocanthosaurus delfsi	McIntosh and Williams, 1988
Haplocanthosaurus priscus (type)	Hatcher, 1903
Kaatedocus siberi	Tschopp and Mateus, 2013
Maraapunisaurus fragillimus	Cope, 1878; Carpenter, 2018
Smitanosaurus agilis	Marsh, 1889; Whitlock and Wilson Mantilla, in press
Supersaurus vivianae	Jensen, 1985
Suuwassea emilieae	Harris and Dodson, 2004

Table 2. Measurements of holotypic dorsal vertebrae of *Amphicoelias altus* (AMNH FARB 5764). DvA = middle–posterior dorsal vertebra; DvB = posterior dorsal vertebra. Measurements in millimetres.

Dimension	DvA	DvB
Anteroposterior length of centrum	~220	242
Dorsoventral height of anterior end of centrum	–	251
Dorsoventral height of posterior end of centrum	~274	266
Mediolateral width of posterior end of centrum	~231	258
Dorsoventral height of neural arch	~239	212
Dorsoventral height of neural spine	–	620
Anteroposterior length of neural spine (near base)	–	158
Mediolateral width of neural spine (near base)	–	72
Maximum mediolateral width of neural spine (near apex)	–	148
Distance from midline to lateral tip of diapophysis	–	303
Dorsoventral height at lateral tip of diapophysis	–	111

Table 3. Measurements of holotypic femur of *Amphicoelias altus* (AMNH FARB 5764). Measurements in millimetres.

Dimension	Measurement
Proximodistal length	~1700
Distance from proximal end to distal tip of fourth trochanter	~900
Maximum diameter of shaft at distal end of proximal half	233
Diameter of shaft at distal end of proximal half, perpendicular to maximum diameter	222
Midshaft mediolateral width	219
Midshaft anteroposterior length	238
Mediolateral width of shaft at two-thirds of femur length (from proximal end)	228
Anteroposterior length of shaft at two-thirds of femur length (from proximal end)	211
Minimum shaft circumference	701
Distal end anteroposterior length	~354
Mediolateral width of tibial condyle	166
Mediolateral width of fibular condyle	~187

Table 4. Measurements of specimens previously referred to *Amphicoelias altus*: AMNH FARB 5764 (tooth), 5764a (scapula, coracoid, ulna, femur) and 5761 (humerus). Measurements in millimetres.

Element	Dimension	Measurement
Tooth	Base of crown maximum mesiodistal width	25
	Base of crown maximum labiolingual width	18
Scapula	Anteroposterior length as preserved	1450
	Dorsoventral height of acromion	940
	Minimum dorsoventral height of scapular blade	270
Coracoid	Maximum dorsoventral height	710
	Maximum anteroposterior length	450
Humerus	Proximodistal length	1140
	Maximum mediolateral width of proximal end	~510
	Mediolateral width at midshaft	215
	Anteroposterior diameter at midshaft	125
	Mediolateral width at distal end	370
Ulna	Proximodistal length	1035
	Mediolateral width of proximal end	265
Femur	Proximodistal length as preserved	840
	Mediolateral width at midshaft	190
	Anteroposterior diameter at midshaft	160
	Mediolateral width of distal end	360

FIGURE CAPTIONS

**Figure 1.** *Amphicoelias altus* holotype (AMNH FARB 5764) middle–posterior dorsal vertebra
in anterior (A, E, I), right lateral (B, F, J), posterior (C, G, K), left lateral (D, H), and ventral (L,
anterior towards top) views. Grey tone indicates anatomy reconstructed or obscured by
matrix; hatched areas indicate broken surfaces. Abbreviations: D, diapophysis; CPOL,
centropostzygapophyseal lamina; CPRL, centroprezygapophyseal lamina; P, parapophysis;
PCDL, posterior centrodiapophyseal lamina; PCPL, posterior centroparapophyseal lamina;
PRDL, prezygodiapophyseal lamina; PRZ, prezygapophysis; SPDL, spinodiapophyseal lamina;
SPRL, spinoprezygapophyseal lamina. Scalebar = 100 mm. Parts I, J and K modified from
Osborn and Mook (1921).

**Figure 2.** *Amphicoelias altus* holotype (AMNH FARB 5764) posterior dorsal vertebra in
anterior (A, E), right lateral (B, F), posterior (C, G), left lateral (D) and ventral (H, anterior
toward top) views. Blue polygon in C obscures hand used to brace the fragile vertebra for
photography. Abbreviations: CPOL, centropostzygapophyseal lamina; CPRL+ACPL, conjoined
centroprezygapophyseal lamina and anterior centroparapophyseal lamina; D, diapophysis;
HS, hyposphene; ISPOL, lateral spinopostzygapophyseal lamina; mSPOL, medial
spinopostzygapophyseal lamina; P, parapophysis; PCDL, posterior centrodiapophyseal
lamina; PCPDL, posterior centroparapophyseal lamina; PODL, postzygodiapophyseal lamina;
PRSL, prespinal lamina; SPDL, spinodiapophyseal lamina; SPRL, spinoprezygapophyseal
lamina. Scalebar = 100 mm. Parts E–G modified from Osborn and Mook (1921).

**Figure 3.** *Amphicoelias altus* holotype (AMNH FARB 5764) right pubis in lateral view.
Scalebar = 100 mm.

**Figure 4.** *Amphicoelias altus* holotype (AMNH FARB 5764) right femur in anterior (A), lateral
(B), posterior (C), and medial (D) views. Cross-sectional shape from break at mid-shaft
shown in (E). Scalebar = 100 mm. Parts A–D modified from Osborn and Mook (1921).

**Figure 5.** Conservation history of the holotypic dorsal vertebrae (AMNH FARB 5764) of
*Amphicoelias altus*. The middle–posterior (A–D) and posterior dorsal (E–H) vertebrae were
photographed under UV light (UV-A in parts A, D; combined UV-A, B, and C wavelengths in
parts E, H), and under polarized light to increase contrast (parts B, C, F, G). The vertebrae
are shown in right lateral (A, B, E, F) and anterior (C, D, G, H) views. Note the different types
of filler material, indicating several generations of interventions, and the uniform blueish
colour, which indicates consolidation with shellac. Shellac coating and attached fuzz remains
nearly invisible under normal and polarized light. The individual photographs were not
modified with any kind of editing software. Photographs provided by M. Eklund (a complete
set of Progressive Photonics photographs is available in the electronic supplementary
material).

**Figure 6.** AMNH FARB 5764, tooth, in labial (A), lingual (B), and occlusal (C) views. Scalebar =
50 mm.

**Figure 7.** AMNH FARB 5764a [Cor. 3], left coracoid (A, C), and AMNH FARB 5764a [Sc. 7], left
scapula (B, D), in lateral view. Scalebar = 100 mm. Parts C and D modified from Osborn and
Mook (1921).

**Figure 8.** AMNH FARB 5764a [Ul. 1], left ulna, in lateral (A), anterior (B), medial (C), posterior
(D), proximal (E), and distal (F) views. Scalebar = 100 mm. Parts C, E and F modified from
Osborn and Mook (1921).

**Figure 9.** AMNH FARB 5764a [Fem. 2], distal femur, in anterior (A) and posterior (B) views.
Scalebar = 100 mm.

**Figure 10.** AMNH FARB 5761 [Pb. 6], left (?) pubis, in presumed lateral (A) and medial (B)
views. Scalebar = 100 mm.

**Figure 11.** AMNH FARB 5761 [H.1], right humerus, in anterior (A) and posterior (B) views.
Scalebar = 100 mm.

Figure 1. *Amphicoelias altus* holotype (AMNH FARB 5764) middle-posterior dorsal vertebra in anterior (A, E, I), right lateral (B, F, J), posterior (C, G, K), left lateral (D, H), and ventral (L, anterior towards top) views. Grey tone indicates anatomy reconstructed or obscured by matrix; hatched areas indicate broken surfaces.

Abbreviations: D, diapophysis; CPOL, centropostzygapophyseal lamina; CPRL, centrozygapophyseal lamina; P, parapophysis; PCDL, posterior centrodiapophyseal lamina; PCPL, posterior centroparapophyseal lamina; PRDL, prezygodiapophyseal lamina; PRZ, prezygapophysis; SPDL, spinodiapophyseal lamina; SPRL, spinoprezygapophyseal lamina. Scalebar = 100 mm. Parts I, J and K modified from Osborn and Mook (1921).

149x214mm (300 x 300 DPI)

Figure 2. *Amphicoelias altus* holotype (AMNH FARB 5764) posterior dorsal vertebra in anterior (A, E), right lateral (B, F), posterior (C, G), left lateral (D) and ventral (H, anterior toward top) views. Blue polygon in C obscures hand used to brace the fragile vertebra for photography. Abbreviations: CPOL, centropostzygapophyseal lamina; CPRL+ACPL, conjoined centroprezygapophyseal lamina and anterior centroparapophyseal lamina; D, diapophysis; HS, hyposphene; ISPOL, lateral spinopostzygapophyseal lamina; mSPOL, medial spinopostzygapophyseal lamina; P, parapophysis; PCDL, posterior centrodiaepophyseal lamina; PCPDL, posterior centroparapophyseal lamina; PODL, postzygodiapophyseal lamina; PRSL, prespinal lamina; SPDL, spinodiapophyseal lamina; SPRL, spinoprezygapophyseal lamina. Scalebar = 100 mm. Parts E–G modified from Osborn and Mook (1921).

182x225mm (300 x 300 DPI)

Figure 3. *Amphicoelias altus* holotype (AMNH FARB 5764) right pubis in lateral view. Scalebar = 100 mm.

68x145mm (300 x 300 DPI)

Figure 4. *Amphicoelias altus* holotype (AMNH FARB 5764) right femur in anterior (A), lateral (B), posterior (C), and medial (D) views. Cross-sectional shape from break at mid-shaft shown in (E). Scalebar = 100 mm. Parts A–D modified from Osborn and Mook (1921).

145x153mm (300 x 300 DPI)

Figure 5. Conservation history of the holotypic dorsal vertebrae (AMNH FARB 5764) of *Amphicoelias altus*. The middle-posterior (A–D) and posterior dorsal (E–H) vertebrae were photographed under UV light (UV-A in parts A, D; combined UV-A, B, and C wavelengths in parts E, H), and under polarized light to increase contrast (parts B, C, F, G). The vertebrae are shown in right lateral (A, B, E, F) and anterior (C, D, G, H) views. Note the different types of filler material, indicating several generations of interventions, and the uniform blueish colour, which indicates consolidation with shellac. Shellac coating and attached fuzz remains nearly invisible under normal and polarized light. The individual photographs were not modified with any kind of editing software. Photographs provided by M. Eklund (a complete set of Progressive Photonics photographs is available in the electronic supplementary material).

Figure 6. AMNH FARB 5764, tooth, in labial (A), lingual (B), and occlusal (C) views. Scalebar = 50 mm.

72x132mm (300 x 300 DPI)

Figure 7. AMNH FARB 5764a [Cor. 3], left coracoid (A, C), and AMNH FARB 5764a [Sc. 7], left scapula (B, D), in lateral view. Scalebar = 100 mm. Parts C and D modified from Osborn and Mook (1921).

160x139mm (300 x 300 DPI)

Figure 8. AMNH FARB 5764a [Ul. 1], left ulna, in lateral (A), anterior (B), medial (C), posterior (D), proximal (E), and distal (F) views. Scalebar = 100 mm. Parts C, E and F modified from Osborn and Mook (1921).

117x94mm (300 x 300 DPI)

Figure 9. AMNH FARB 5764a [Fem. 2], distal femur, in anterior (A) and posterior (B) views. Scalebar = 100 mm.

68x72mm (300 x 300 DPI)

Figure 10. AMNH FARB 5761 [Pb. 6], left (?) pubis, in presumed lateral (A) and medial (B) views. Scalebar = 100 mm.

74x92mm (300 x 300 DPI)

Figure 11. AMNH FARB 5761 [H.1], right humerus, in anterior (A) and posterior (B) views. Scalebar = 100 mm.

74x83mm (300 x 300 DPI)

Appendix B

Review: Mannion et al. "Anatomy and systematics of the diplodocoid Amphicoelias altus supports high sauropod dinosaur diversity in the Upper Jurassic Morrison Formation, USA"

This paper redescribes important sauropod dinosaur material collected from the Morrison Formation of western North America in the late 19th Century. The authors reexamination of this material is followed by a reexamination of the validity of the species of *Amphicoelias* and a discussion of sauropod species diversity in the Morrison Formation and its ecological implications.

Description: The description is for the most part good, but there are two ways that it could be improved considerably, particularly for the two dorsal vertebrae. First, there needs to be more clarification about what is preserved and missing and what is reconstructed. The description includes details on aspects of the morphology of the prezygapophyses that are illustrated as missing/reconstructed in the interpretive diagram. Second, there is a mismatch between what is described in text and what one can see from the photographs and interpretive line drawings done by the authors. There are claims about laminae, for example, that are difficult to verify from the images and labeling provided. There are also some minor quibbles I had with the laminar terminology (e.g., use of "ISPOL" instead of "lat SPOL"), which are detailed in comments on the PDF.

Figures: The figures are not up to par. As I mentioned above, it is difficult to identify key structures on the interpretive drawings. These latter images are not well done, with labeling set in too close to the object, leader lines almost running into labels, etc. I have left a series of comments in both figures of the vertebrae to help the authors. At least one of them (John Whitlock) will have heard this previously.

Systematics: The title promises systematics, but there is no formal analysis of the interrelationships of *Amphicoelias* in this paper. This is a serious omission that diminishes the strength of arguments about the phylogenetic affinities of the taxon, including the claim that it is not synonymous with *Diplodocus* (see Section 5.1.). It is difficult to imagine that with three authors with experience in this group that there is no analysis.

Discussion: Following the more systematic part of the discussion is a consideration of the implications of the validity of *Amphicoelias* for Morrison Fm. diversity (section 5.2.). The authors discuss whether previous claims about certain Morrison Formation sauropod species being 'ontogimorphs.' I agree that the Morrison is not rife with 'ontogimorphs' posing as real taxa, but the form of argumentation made by the authors here is one from authority and therefore weak. If the authors wish to address this issue, I suggest they outline the predictions of the ontogimorph hypothesis and then counter them one by one. Ironically, one of the obvious counters is about the phylogenetic affinities of *Amphicoelias*, which they do not test in this paper. Without this, this section dissolves into an exercise in arm-waving that won't carry more weight than the original by Woodruff and co-authors. The ontogimorph section is followed by a lengthy section evaluating the claim that the sauropod diversity is too great for

the Morrison Formation. Again, the authors argue authoritatively, in this case by simply dismissing Woodruff (and to some extent Farlow). They never provide a robust defense (or proposal) that the Morrison ecosystem could in fact support the implied diversity.

Appendix C**ROYAL SOCIETY
OPEN SCIENCE****Anatomy and systematics of the diplodocoid *Amphicoelias altus* supports high sauropod dinosaur diversity in the Upper Jurassic Morrison Formation, USA**

Journal:	Royal Society Open Science
Manuscript ID	RSOS-200963
Article Type:	Research
Date Submitted by the Author:	03-Jun-2020
Complete List of Authors:	Mannion, Philip; University College London, Earth Sciences Tschopp, Emanuel; Universität Hamburg, Centrum für Naturkunde; American Museum of Natural History, Division of Paleontology Whitlock, John; Mount Aloysius College Department of Science and Mathematics; Carnegie Museum of Natural History, Section of Vertebrate Paleontology
Subject:	ecology < BIOLOGY, evolution < BIOLOGY, palaeontology < BIOLOGY
Keywords:	Dinosauria, Diplodocus , Late Jurassic, Morrison Formation, Ontogeny, Sauropoda
Subject Category:	Organismal and Evolutionary Biology

**Author-supplied statements**

Relevant information will appear here if provided.

***Ethics***

*Does your article include research that required ethical approval or permits?:*

This article does not present research with ethical considerations

*Statement (if applicable):*

CUST_IF_YES_ETHICS :No data available.

***Data***

*It is a condition of publication that data, code and materials supporting your paper are made publicly*
*available. Does your paper present new data?:*

Yes

*Statement (if applicable):*

The datasets supporting this article have been uploaded as part of the supplementary material.

***Conflict of interest***

I/We declare we have no competing interests

*Statement (if applicable):*

CUST_STATE_CONFLICT :No data available.

***Authors' contributions***

This paper has multiple authors and our individual contributions were as below

*Statement (if applicable):*

PM conceived of and coordinated the study. All authors participated in the design of the study,
contributed data, performed the analyses and interpretation of the data, and wrote the manuscript.
ET and JW drafted the figures. All authors gave final approval for publication and agree to be held
accountable for the work performed therein.

Anatomy and systematics of the diplodocoid *Amphicoelias altus* supports high sauropod dinosaur diversity in the Upper Jurassic Morrison Formation of the USA

Philip D. Mannion¹, Emanuel Tschopp² and John A. Whitlock^{4,5}

¹ Department of Earth Sciences, University College London, London, WC1E 6BT, UK

² Centrum für Naturkunde, Universität Hamburg, 20146 Hamburg, Germany

³ Division of Paleontology, American Museum of Natural History, Central Park West at 79th Street, New York 10024-5192, USA

⁴ Department of Science and Mathematics, Mount Aloysius College, Cresson, Pennsylvania 16630-1999, USA

⁵ Section of Vertebrate Paleontology, Carnegie Museum of Natural History, Pittsburgh, Pennsylvania 15213-4007, USA

Author for correspondence (Email: philipdmannion@gmail.com)

RRH: *AMPHICOELIAS ALTUS* & MORRISON SAUROPOD DIVERSITY

LRH: MANNION, TSCHOPP & WHITLOCK

ABSTRACT

Sauropod dinosaurs were an abundant and diverse component of the Upper Jurassic Morrison Formation of the USA, with 24 currently recognised species. However, some authors consider this high diversity to have been ecologically unviable and the validity of some species has been questioned, with suggestions that they represent growth series ('ontogimorphs') of other species. Under this scenario, high sauropod diversity in the Late Jurassic of North America is greatly overestimated. One putative ontogimorph is the enigmatic diplodocoid *Amphicoelias altus*, which has been suggested to be synonymous with *Diplodocus*. Given that *Amphicoelias* was named first, it has priority and thus *Diplodocus* would become a junior synonym. Here we provide a detailed re-description of *Amphicoelias altus* in which we restrict it to the holotype individual and support its validity, based on three autapomorphies. Our re-evaluation supports recent phylogenetic analyses that recover *Amphicoelias* as distantly related to *Diplodocus*, and thus the latter is also retained as a valid taxon. There is no evidence to support the view that any of the currently recognised Morrison sauropods are ontogimorphs. Available data indicate that sauropod anatomy did not dramatically alter once individuals approached maturity. Furthermore, subadult sauropod individuals are not prone to stemward slippage in phylogenetic analyses, casting doubt on the possibility that their taxonomic affinities are substantially misinterpreted. An anatomical feature can have both an ontogenetic and phylogenetic signature, but the former does not outweigh the latter when other characters overwhelmingly support the affinities of a taxon. Many sauropods were spatiotemporally and/or ecologically separated from one another. Combined with the biases that cloud our reading of the fossil record, we contend that the number of sauropod dinosaur species in the Morrison Formation is currently likely to be underestimated, not overestimated.

Keywords: Dinosauria; *Diplodocus*; Late Jurassic; Morrison Formation; Ontogeny; Sauropoda

1. Introduction

The Upper Jurassic Morrison Formation of the western USA has yielded a high diversity of sauropods, including some of the most iconic dinosaurs, such as *Brachiosaurus*, *Brontosaurus*, and *Diplodocus* (Bakker, 1971; McIntosh, 1990; Foster, 2007). *Apatosaurus* and *Camarasaurus* were also abundant components of this fauna, whereas a number of other sauropod taxa are known from far fewer remains (Foster, 2003). Some of these have only been recognised this century, comprising *Galeamopus* (Tschopp et al., 2015), *Kaatedocus* (Tschopp and Mateus, 2013), *Smitanosaurus* (Whitlock and Wilson Mantilla, in press), and *Suuwassea* (Harris and Dodson, 2004), whereas others have been known for much longer, consisting of *Amphicoelias* (Cope, 1877a), *Barosaurus* (Marsh, 1890), *Haplocanthosaurus* (Hatcher, 1903a), and *Supersaurus* (Jensen, 1985). Currently, 24 sauropod species assigned to 14 genera are recognised as valid in the Morrison Formation (e.g. Upchurch et al., 2004a,b; Lovelace et al., 2008; Tschopp et al., 2015, 2017; Carpenter, 2018; Whitlock and Wilson Mantilla, in press; Table 1), although some authors have suggested that many of these species are synonyms, and that they represent growth series of a smaller number of valid taxa (Woodruff and Fowler, 2012; Woodruff and Foster, 2014; Woodruff et al. 2017, 2018; Woodruff, 2019).

One of the earliest named and most enigmatic of Morrison sauropods is *Amphicoelias*. The type species, *Amphicoelias altus*, was erected by Cope in 1877 for two dorsal vertebrae, a partial pubis, and a femur. These elements were collected by A. Ripley from Cope Quarry XII in Garden Park, north of Cañon City, in Fremont County, Colorado, and were shipped to Cope by O. W. Lucas (McIntosh, 1998). Following Cope's death, his collection was acquired by the American Museum of Natural History (AMNH) in 1902. Two further species of *Amphicoelias* were named by Cope – *Amphicoelias latus* (Cope, 1877a) and *Amphicoelias fragillimus* (Cope, 1878) – but neither species is currently considered to belong to the genus (e.g. Osborn and Mook, 1921; Carpenter 2018). No further remains can currently be unambiguously referred to *Amphicoelias*, making it one of the rarest taxa in the Morrison Formation, despite being known for over 140 years.

Cope (1877a) considered *Amphicoelias* to be a close relative of *Camarasaurus*, although he classified them in the separate, monogeneric families, Amphicoeliidae and Camarasauridae, respectively. *Amphicoelias* was generally regarded as either a close relative or synonym of *Camarasaurus* for the following four decades (e.g. Marsh, 1896; Huene, 1908), before Osborn and Mook (1921) undertook a review of Cope's sauropod taxa, in which they argued that *Amphicoelias* was most closely related to the diplodocids *Barosaurus* and *Diplodocus*. Nopcsa (1928) also allied *Amphicoelias* with diplodocids and, from Romer (1956) onwards, a diplodocid or diplodocoid placement has been universally accepted (e.g. Wilson, 2002; Upchurch et al., 2004a; Rauhut et al., 2005; Whitlock, 2011a; Tschopp et al., 2015; Mannion et al., 2019). Rauhut et al. (2005) were the first to incorporate *Amphicoelias* into a phylogenetic analysis, recovering it as the most 'basal' diplodocoid, outside Diplodocimorpha. The diplodocoid-focused analysis of Whitlock (2011a) also placed *Amphicoelias* as a non-diplodocimorph diplodocoid, a position recovered in subsequent analyses of revised versions of this data matrix (e.g. Mannion et al., 2012; Gallina et al., 2014). This placement was also supported in analyses of a recent independent phylogenetic data matrix of eusauropods (Mannion et al., 2019). By contrast, analyses of a second independent data matrix, focussed on diplodocids, recovered *Amphicoelias* within Diplodocidae, either as a 'basal' member of this clade, or within

Apatosaurinae (Tschopp et al., 2015). The latter position was also supported by analyses of a
revised version of this data matrix (Tschopp and Mateus, 2017).

Although generally considered a valid genus (e.g. Upchurch et al., 2004a), there is not
universal agreement on this point. Osborn and Mook (1921) and other authors (e.g.
McIntosh, 1990) have commented upon similarities between *Amphicoelias* and other
Morrison diplodocids, especially *Diplodocus*, and a small number of studies have argued that
*Amphicoelias* is actually synonymous with *Diplodocus* (Foster, 2003; Woodruff and Foster,
2014; Woodruff, 2019). Given that *Amphicoelias* (Cope, 1877a) was named before
*Diplodocus* (Marsh, 1878), and therefore has priority, any such synonymisation would have
notable taxonomic ramifications.

Given the uncertainty of its taxonomic status and phylogenetic affinity, here we provide a
detailed re-description of the holotypic remains of *Amphicoelias altus*, reassess its validity,
and discuss the systematics of remains previously attributed to this genus. Finally, we
present a revised view of sauropod diversity in the Morrison Formation.

1.1. Institutional abbreviations

**AMNH**, American Museum of Natural History, New York, USA; **MOR**, Museum of the
Rockies, Bozeman, Montana, USA.

1.2. Anatomical abbreviations

**ACDL**, anterior centrodiapophyseal lamina; **ACPL**, anterior centroprezygapophyseal lamina;
**CPOF**, centropostzygapophyseal fossa; **CPRL**, centroprezygapophyseal lamina; **ISPOL**, lateral
spinopostzygapophyseal lamina; **mSPOL**, medial spinopostzygapophyseal lamina; **PACDF**,
parapophyseal centrodiapophyseal fossa; **PCDL**, posterior centrodiapophyseal lamina; **PCPL**,
posterior centroparapophyseal lamina; **PODL**, postzygodiapophyseal lamina; **POSDF**,
postzygapophyseal spinodiapophyseal fossa; **POSL**, postspinal lamina; **PRDL**,
prezygodiapophyseal lamina; **PRSDF**, prezygapophyseal spinodiapophyseal fossa; **PRSL**,
prespinal lamina; **SPDL**, spinodiapophyseal lamina; **SPOL**, spinopostzygapophyseal lamina;
**SPOL-F**, spinopostzygapophyseal lamina fossa; **SPRL**, spinoprezygapophyseal lamina; **TPRL**,
interprezygapophyseal lamina.

2. Systematic Paleontology

SAUROPODA Marsh, 1878
NEOSAUROPODA Bonaparte, 1986
DIPLODOCOIDEA Marsh, 1884
*AMPHICOELIAS* Cope, 1877a

**Type species:** *Amphicoelias altus* Cope, 1877a

**Holotype:** AMNH FARB 5764 – two middle–posterior dorsal vertebrae, a fragmentary pubis,
and a right femur (Figs 1–5).

**Locality and horizon:** Cope Quarry XII, Garden Park, 8 miles N/NE of Cañon City, Fremont
County, Colorado, USA (Cope, 1877a; Osborn and Mook, 1921; McIntosh, 1998); Morrison

Formation, C5 systems tract, 150.44–149.21 Ma, lower Tithonian, Upper Jurassic (Carpenter,
1998; Trujillo and Kowallis, 2015; Maidment and Muxworthy, 2019).

**Revised diagnosis:** *Amphicoelias* can be diagnosed by two autapomorphies (denoted by an
asterisk), as well as one local autapomorphy: (1) apex of posterior dorsal neural spine with
rounded, non-tapered lateral projections resulting from expansion of spinodiapophyseal
laminae*; (2) femoral shaft with subcircular cross section*; and (3) femur distally bevelled,
with the fibula condyle extending further distally than the tibial condyle.

**Additional information:** Numerous additional remains have been referred to *Amphicoelias*
(e.g. Cope, 1877a, 1878; Osborn and Mook, 1921; Wilson and Smith, 1996). These are
discussed and re-evaluated below, but we follow recent authors (e.g. Mannion et al., 2012,
2019; Tschopp et al., 2015) in restricting *Amphicoelias* to the type material.

**Curatorial history:** Curatorial history of the type and previously referred material is
complicated, mostly because of inadequate field notes from the excavations, but also due to
issues emanating from renumbering after the acquisition of Cope's Garden Park collection
by the AMNH. After acquisition, the material was catalogued with numbers ranging from
AMNH FARB 5760 to 5777. The type specimens of the three *Amphicoelias* species erected
by Cope were catalogued as AMNH FARB 5764 (*A. altus*), AMNH FARB 5765 (*A. latus*), and
AMNH FARB 5777 (*A. fragillimus*; this specimen is missing, and it is unclear if this specimen
was lost before, during, or after the acquisition by the AMNH). Additional material was later
referred to *Amphicoelias*, both in publications and internally in the museum's collections.

More recently, a second renumbering attempt seems to have been undertaken at AMNH,
presumably based on the detailed historical work of McIntosh (1998), who successfully
identified several partial skeletons among the AMNH material from Garden Park. It is not
known who conducted this renumbering, as this happened before current collection and
curatorial staff took office, and no notes exist that could be used to identify those involved.
To make matter even more complicated, these new numbers were solely noted on sheets of
pink paper associated with the bones in the collection space and were not transferred onto
the collection catalogue. Numbers used for the Garden Park material range from 30001 to
at least 30014, and possibly higher (ET, pers. obs. 2019). The renumbered specimens are
mostly from the original catalogue numbers AMNH FARB 5760 and 5761, which were shown
to include material from multiple individuals (Osborn and Mook, 1921; McIntosh, 1998).
These original numbers are the ones used in all scientific publications concerning the Cope
specimens from Garden Park (e.g. Osborn and Mook, 1921; McIntosh, 1998). The new
numbers, instead, do not correspond to their entries in the AMNH collection catalogue, in
which the numbers AMNH FARB 30001–30014 are listed as specimens of the turtles
*Brachyopsemys tingitana* (holotype, AMNH FARB 30001) and *Phosphatochelys* sp. (AMNH
FARB 30008), Testudines indet. (AMNH FARB 30002 and 30004–30007), an indeterminate
therapsid (AMNH FARB 30003), and the holotype (AMNH FARB 30009) and paratypes of the
frog *Xenopus arabiensis* (AMNH FARB 30010–30014; C. Mehling, pers. comm. 2019, 2020).
Among these erroneously recatalogued sauropod specimens are a humerus and pubis
originally cataloged as AMNH FARB 5761, which are also associated with anonymous
collection notes referring them to *Amphicoelias altus*, as is a second pubis numbered AMNH
FARB 5760, which was not renumbered. These elements correspond to the right humerus
H.1 (Osborn and Mook 1921: fig. 89), and the pubes P.3 (Osborn and Mook 1921: fig. 105)

and P.6 (Osborn and Mook 1921: fig. 106). Given that the renumbering has never been
transferred to the official AMNH collection catalogue, that these numbers are registered as
belonging to different species and specimens (including some type specimens), and that we
are not aware of any scientific publication referring to these numbers, we ignore the new
numbers between 30001 and 30014 associated with these bones in the collection space,
and use only the original catalogue numbers, plus the specific bone identifiers proposed by
Osborn and Mook (1921) when necessary to refer to single bones or discuss affiliation to
individual skeletons (e.g. H.1, P.3, P.6).

**Conservation history:** The two holotypic dorsal vertebrae of *Amphicoelias altus* are
incomplete, and parts of them are heavily restored. Much of the restoration and repair
likely stems from the original excavation in Colorado and preparation in Philadelphia, but
the subsequent move from Philadelphia to New York, and further study (including ours) has
resulted in additional damage. Mineralization of these fossil vertebrae seems to be very
extensive, meaning that the bones are heavy, and manipulation easily results in damage to
the delicate vertebral laminae and processes.

In an attempt to better identify reconstructed parts and repaired breaks, we used
Progressive Photonics (Eklund et al., 2018) to document the vertebrae in anterior and right
lateral views (electronic supplementary material). Progressive Photonics is a workflow to
document an object under various lighting conditions and wavelengths, including frontal
and oblique lighting, polarized light, and UV stimulation (Eklund et al., 2018). UV
stimulation, in particular, produces different reactions that can help to distinguish real fossil
bone from various materials used in repair and restoration. Whereas distinct materials for
repairs could be easily distinguished from each other, distinction between restored and real
fossil bone was more difficult. At least three different adhesives were used to repair breaks,
in some cases at the same fracture, indicating that the vertebra broke several times at the
same location (e.g. at the junction of the neural arch with the centrum in the posterior
dorsal vertebra [Fig. 5]). Restoration of the vertebrae was likely performed early in their
conservation history, because restored parts and original bone react in a very similar way to
UV stimulation, producing a blueish hue (Fig. 5). A similar blueish hue was identified as
shellac coating in other historic fossil material from Wyoming (Tschopp et al., in press),
which was commonly used as a consolidant in the field and in fossil preparation until at least
the 1930s (Linares Soriano and Carrascosa, Moliner 2016). This coating must have been
applied over the entire restored vertebrae, thereby obscuring the distinction between
original bone and the restored portions under UV stimulation. At the anterior-most portion
of the right surface of the centrum of the posterior dorsal vertebra, the original shellac
seems to have flaked off, revealing the actual, lighter colour of the bone, which does react
slightly differently to UV stimulation (Fig. 5).

**3. Description and comparisons of *Amphicoelias altus***

**3.1. Middle–posterior dorsal vertebra**

Nomenclature for vertebral laminae and fossae follows Wilson (1999) and Wilson et al.
(2011), respectively. An anteriorly and posteriorly incomplete centrum, most of the neural
arch, and the base of the neural spine of a middle–posterior dorsal vertebra is preserved
(Figs 1, 5; see Table 2 for measurements). Despite being reconstructed with a bifid neural

spine in Osborn and Mook (1921: fig. 119), not enough of the neural spine is preserved to be
able to determine its morphology.

The ventral surface of the centrum has been slightly crushed. There is a weak, rounded
ridge close to the midline, although this does not form a distinct keel; otherwise, the ventral
surface is fairly flat centrally, becoming gently transversely convex towards the lateral
margins. There are no ventrolateral ridges or fossae. The lateral pneumatic foramen is fairly
consistent on each side of the centrum, but the margins of both are mostly reconstructed.
Each foramen is situated on the dorsal half of the vertebra, occupying approximately half of
the centrum length, with a slight anterior bias. As reconstructed, each foramen has an
elliptical outline, with its long axis oriented anteroposteriorly, although it is also fairly tall
dorsoventrally; this reconstruction appears fairly accurate. Each foramen is deep, leaving a
thin midline septum and, internally, it ramifies strongly dorsally and, especially ventrally, as
well as a small distance anteriorly and posteriorly. In this regard, *Amphicoelias* is similar to
most neosauropods, with the notable exceptions of dicraeosaurids and some titanosaurs, in
which the lateral excavations are shallow (Upchurch, 1998). There are no vertical ridges
inside the foramen, contrasting with the condition in several diplodocids (Mannion et al.,
2012; Tschopp et al., 2015), but there is a sub-horizontal ridge at the anteroventral corner,
which is not visible in lateral view – this is present on both sides, although is more
developed on the right side.

The sharp lateral margin of the centroprezygapophyseal lamina (CPRL) also forms the
anterior centroparapophyseal lamina (ACPL). A posterior centroparapophyseal lamina
(PCPL) is also present, although this only becomes visible towards the parapophysis,
situated on the lateral surface of the prezygapophysis. The posterior centrodiaepophyseal
lamina (PCDL) is near-vertical, and forms the posterior margin to a deep parapophyseal
centrodiaepophyseal fossa (PACDF) on the lateral surface of the upper part of the neural
arch, bounded anteroventrally by the PCPL, and dorsally by the gently posterodorsally
directed prezygodiaepophyseal lamina (PRDL).

Although this region is heavily reconstructed, the anterior neural canal opening is clearly
set within a fossa, as in most eusauropods (Carballido et al., 2012). The prezygapophyseal
articular surfaces are gently convex mediolaterally, and there are well developed hypantral
surfaces. A weakly developed, horizontal interprezygapophyseal lamina (TPRL) is strongly U-
shaped in dorsal view. A TPRL separates the prezygapophyses in most eusauropods,
whereas their articular surfaces are confluent in rebbachisaurids (Apesteguía et al., 2010;
Wilson and Allain, 2015). The postzygapophyseal articular surfaces are gently concave,
suggesting that the morphology of the prezygapophyseal surfaces might be genuine.
Although there is no hyposphene, its absence is almost certainly a preservational artefact,
based on the ventral surface of the postzygapophyseal midline being clearly damaged. The
poor preservation in this region means that we also cannot determine whether a vertical
midline lamina extended between the posterior neural canal opening and the
postzygapophyseal complex.

The diapophysis projects laterally and is dorsally deflected, but it has been broken and
slightly deformed, and so this dorsal deflection might merely be artefactual. The
spinoprezygapophyseal laminae (SPRLs) extend from close to the medial margins of the
posterior ends of the prezygapophyses and merge close to the apex of the incomplete
neural spine, forming a narrow, anteriorly projecting prespinal lamina (PRSL). Ventral to
their junction, there is no evidence for a PRSL. The base of the subvertical
spinodiapophyseal lamina (SPDL) is preserved, and fully extends down to the diapophysis. A

prezygapophyseal spinodiapophyseal fossa (PRSDF) is formed anterior to the SPDL, bounded ventrally by the PRDL, and anterodorsally by the SPRL. The spinopostzygapophyseal lamina (SPOL) is bifurcated a short distance above the postzygapophyses, with a large spinopostzygapophyseal lamina fossa (SPOL-F) in between the medial (mSPOL) and lateral (lSPOL) branches. A bifid SPOL is common across an array of eusauropod lineages (Wilson, 2002; Mannion et al., 2013), and *Amphicoelias* differs from some rebbachisaurids in which the SPOL is divided throughout its entire length (Whitlock, 2011a; Mannion et al., 2019). The mSPOL is directed dorsomedially along its preserved basal portion, with no midline postspinal lamina (POSL) within the postspinal fossa. As such, the dorsal vertebrae of *Amphicoelias* differ from those of most diplodocimorphs, in which pre- and postspinal laminae are present throughout nearly the full length of the neural spine, and are often not formed solely by convergence of the SPRLs or SPOLs (Wilson, 1999; Whitlock, 2011a; Mannion et al., 2019). The dorsal neural spines of *Haplocanthosaurus priscus* (Hatcher, 1903b) have a similar laminae configuration to *Amphicoelias*. 
[revised manuscript text omitted]
 1.5 (Mannion et al., 2012, 2013). Although several diplodocoids have low ratios (e.g. *Tornieria* = 1.3; Mannion et al., 2012; Tschopp et al., 2015), these do not approach the subcircular morphology that we therefore regard as autapomorphic for the femur of *Amphicoelias*.

The femur has an anteroposteriorly thick shaft relative to that of the distal end (ratio > 0.6) (Whitlock, 2011a). This is comparable to some 'basal' macronarians (e.g. *Camarasaurus*), several rebbachisaurids, and the dicraeosaurid *Amargasaurus*, but most other sauropods (including diplodocoids) have much lower ratios (Whitlock, 2011a; Tschopp et al., 2015; Mannion et al., 2019). Distally, the femur is bevelled, with the fibular condyle extending further distally than the tibial condyle. This type of bevelling is generally restricted to derived titanosaurs (Wilson, 2002), and is therefore regarded as a local

[revised manuscript text omitted]

The scapula (AMNH FARB 5764a [Sc. 7]) is preserved in four pieces (Fig. 7). It lacks part of
the acromion, the dorsal edge of most of the distal blade, and part of the end of the blade.
The short acromial ridge is at an obtuse angle to the long axis of the distal blade. Anterior to
this ridge, the acromion expands dorsoventrally. The preserved dorsal margin of the
acromion therefore has a somewhat sinuous shape in lateral view, which is an unusual
condition in sauropods. The glenoid is strongly expanded mediolaterally, and slightly
medially bevelled. The latter feature is primarily restricted to somphospondylans, but has
also been noted in the scapula of *Apatosaurus* (Wilson, 2002). There is a rugosity on the
medial surface of the blade, close to the acromion. The proximal part of the scapular blade
is D-shaped in cross-section, with a flat medial surface and dorsoventrally convex lateral
surface, as is the case in most eusauropods (Wilson, 2002). Although the distal-most portion
of the blade is not preserved, a small piece indicates that the blade expanded dorsally at its
distal end, whereas no ventral expansion is indicated along that edge, as preserved.
Although the bevelled glenoid could indicate referral to *Apatosaurus*, we conservatively
regard the AMNH FARB 5764a scapula as belonging to an indeterminate eusauropod.

The coracoid (AMNH FARB 5764a [Cor. 3]) is nearly complete, lacking only a small portion
of its rounded anterodorsal margin (Fig. 7). Based on its size, it is possible that is from the
same individual as the scapula (Osborn and Mook, 1921). The coracoid is slightly
dorsoventrally taller than anteroposteriorly long and has a straight articular surface for the
scapula. The glenoid is strongly expanded mediolaterally, which is unusual for diplodocids
(Tschopp et al., 2019). The articular surface of the glenoid extends onto the lateral surface
of the coracoid, and is anteroventrally bound by a distinct, relatively short, U-shaped notch.
Anterior to the notch, at the ventral end of the anterior margin, there is some broken bone
surface indicating the possible presence of a glenoid lip. The coracoid foramen is situated
slightly dorsal to the glenoid, extending dorsomedially and a little posteriorly through the
coracoid. We consider the AMNH FARB 5764a coracoid as representing an indeterminate
eusauropod.

The ulna (AMNH FARB 5764a [Ul.1]) is preserved in one piece (Fig. 8). Its proximal and
especially distal articular surfaces are partly eroded, but still indicate their general, original
shape. The proximal articular surface of the ulna has the typical triradiate outline of
sauropods. Its anterior and anterolateral proximal rami are approximately equidimensional,
forming a slightly obtuse angle. The posterior ramus does not form a distinct projection
dorsally (there is no well-defined olecranon process) or posteriorly (the ulna is V-shaped,
rather than Y-shaped, in proximal view). A transversely broad ridge extends distally from the
posterior margin. Around midheight, the medial margin of this ridge becomes less distinct,
such that the transition between the medial and posterior surfaces is rounded. However,
the lateral margin of this ridge expands at approximately midheight, continuing distally for
approximately another quarter of the length of the bone, before it also disappears. On the
anterior surface, the scar for the articulation with the radius is only weakly developed, or

possibly also slightly eroded. The distal articular surface is transversely compressed and expanded posteriorly. As with the scapula and coracoid, given the lack of anatomical overlap with the *Amphicoelias altus* holotype, and the absence of synapomorphies of other taxa, we regard the AMNH FARB 5764a ulna as an indeterminate eusauropod.

The distal half of the femur (AMNH FARB 5764a [Fem. 2]) preserves the distal-most extension of the fourth trochanter (Fig. 9), which was not recognized by Osborn and Mook (1921). The location of this feature demonstrates that the femur is a right element, rather than a left as originally identified, and it is likely that the femur is proximodistally much shorter than indicated in the reconstruction in Osborn and Mook (1921: fig. 126). The preserved distal end of the fourth trochanter is restricted to the medial margin of the posterior surface of the shaft and has a convex medial surface. Although slender, the shaft below the fourth trochanter lacks the subcircular outline (Table 4) that characterises the holotypic femur of *Amphicoelias altus*. The anterior parts of the distal condyles are damaged, likely during excavation. Whereas parts of the distal articular surface are preserved, the posterior extension of the condyles is damaged as well. The distal articular condyles are moderately expanded transversely, and there is no indication of a well-developed epicondyle (although this might be affected by damage). The tibial condyle projects slightly further distally than the fibular condyle, which is the reverse of the condition in the holotypic femur of *Amphicoelias altus*. As such, this distal femur clearly differs from that of *Amphicoelias altus* and should be regarded as an indeterminate eusauropod.

4.2. Additional AMNH FARB material catalogued as *Amphicoelias altus*

Two pubes and a humerus from the Morrison Formation of the Garden Park area have been tentatively identified as *Amphicoelias altus* in the AMNH collections. These identifications were presumably based on locality and/or gross morphology.

A left (?) pubis (AMNH FARB 5761 [Pb. 6]; Fig. 10) has no associated information concerning locality, other than being from Cope's Garden Park collection. It is associated with a note in the collections, which seems to identify this bone as AMNH FARB 30012, but this number was erroneously attributed to this pubis and should be ignored (see 'Curatorial history'). The pubis is in two pieces, and the medial side is not preserved, including the articular surface for its counterpart. An anonymous hand-written note in the collections indicates that its referral to *Amphicoelias altus* was based on its similarity in slenderness to the holotypic pubis. However, as in the holotypic pubis of *Amphicoelias altus*, there is little that can be observed concerning informative anatomy, and we regard this specimen as belonging to an indeterminate eusauropod.

The second pubis (AMNH FARB 5760 [Pb. 3]) preserves only the proximal half, and there is also no further locality information associated with this specimen. No significant morphological information can be gleaned that could enable the identification of this bone to any sauropod taxon. Combined with the lack of provenance information, there is no basis for its referral to *Amphicoelias* and we consider it to represent an indeterminate eusauropod.

A right humerus (AMNH FARB 5761 [H. 1]; Fig. 11; see Table 4 for measurements) was **apparently** found at the same site as the *Amphicoelias altus* holotypic material (McIntosh, 1998). As with the Pb. 6 pubis, this humerus has also been associated with a new specimen number (AMNH FARB 30011), which should be ignored (see 'Curatorial history'). The

humerus is fairly complete, lacking only the proximomedial and proximolateral corners. It is
relatively stout (Robustness Index of 0.3), and is symmetrically expanded proximally,
indicating that it is probably referable to Diplodocoidea (Tschopp et al., 2015). The
deltopectoral crest has a distinct distal end that extends to approximately 42% of the total
proximodistal length of the humerus. Its lateral surface is anteroposteriorly concave and
posteriorly accompanied by a distinct, striated ridge that extends for a little more than the
proximal half of the deltopectoral crest. The tubercle for the attachment of the M.
coracobrachialis is situated in the centre of the anterior concavity of the proximal end, as in
most neosauropods (Mannion et al., 2019), and differs from the medially displaced tubercle
that characterises the diplodocine *Galeamopus pabsti* (Tschopp and Mateus, 2017). At
midshaft, the humerus has an elliptical cross section, with the mediolateral diameter 1.7
14 times greater than its anteroposterior dimension. The distal articular surface is associated
with a relatively distinct intercondylar groove on the posterior surface of the shaft. Although
damaged, the medial and lateral ridges on the anterior surface of the distal end would have
been located close to the midline of the articular surface. The ratio of the length of this
humerus to that of the type femur of *Amphicoelias* is 0.64 (McIntosh, 1998), which is
consistent with the ratios in most diplodocoids, but is much lower than other eusauropods
(McIntosh, 1990; Wilson, 2002; Upchurch et al., 2004a; Tschopp et al., 2015; Mannion et al.,
2019). However, McIntosh (1998) considered it unlikely that this robust humerus could have
belonged to *Amphicoelias*, with its slender femur. Presumably based on its symmetrically
expanded proximal portion and the relative robusticity, McIntosh (1998) referred this
humerus to *Apatosaurus* instead. Here, we refer this humerus to Diplodocoidea, and note
the possibility that it could belong to *Amphicoelias*.

4.3. '*Amphicoelias latus*' – AMNH FARB 5765

In the same publication in which he named *Amphicoelias altus*, Cope (1877a) erected a
second species – '*Amphicoelias latus*' – on the basis of four caudal vertebrae and a femur
(AMNH FARB 5765) from a nearby locality (Cope Quarry XV). The femur of this specimen is
much more robust than that of *Amphicoelias altus*, and also has an anteroposteriorly
compressed, elliptical midshaft cross section. Osborn and Mook (1921) noted this difference
and regarded '*Amphicoelias latus*' as a junior synonym of *Camarasaurus supremus*. This
referral has been followed by subsequent authors (e.g. McIntosh, 1990, 1998; Upchurch et
al., 2004a), and was supported through phylogenetic analysis (Tschopp et al., 2015).
Woodruff and Foster (2014) incorrectly stated that previous authors (i.e. Osborn and Mook,
1921; McIntosh, 1998) had synonymized '*Amphicoelias latus*' with *Amphicoelias altus*, a
taxonomic assignment they 'agreed' with. However, there is no basis for such a referral and
we agree with other workers that '*Amphicoelias latus*' is a junior synonym of *Camarasaurus*
*supremus*.

4.4. *Maraapunisaurus* ('*Amphicoelias*') *fragillimus*

Cope (1878) named a third species of *Amphicoelias* from a nearby locality (Cope Quarry
III) the following year. Based only on a middle–posterior dorsal neural arch (AMNH FARB
5777), Cope (1878) erected *Amphicoelias fragillimus*. This specimen was unfortunately lost
(or destroyed) but, despite this, has been the focus of several studies because of its
potentially gigantic size (Carpenter, 2006, 2018; Woodruff and Foster, 2014). Most authors

have synonymised it with *Amphicoelias altus* (e.g. Osborn and Mook, 1921; McIntosh, 1990;
Upchurch et al. 2004a; Woodruff and Foster, 2014), or listed it as a nomen dubium (e.g.
Tschopp et al., 2015 [note that these authors did not provide a diagnosis for *Amphicoelias*
*fragillimus*, as incorrectly claimed by Woodruff, 2019]). However, Carpenter (2018) provided
a novel reinterpretation in which he considered *Amphicoelias fragillimus* as a
rebbachisaurid, erecting the new genus *Maraapunisaurus*. All that remains of
*Maraapunisaurus* ('*Amphicoelias*') *fragillimus* is the drawing of the neural arch in posterior
view, as presented by Cope (1878). Based on this, we agree with Carpenter (2018) that it
differs in a number of anatomical features from *Amphicoelias altus* (e.g. the morphology of
the SPOLs, and the presence of a distinct postspinal lamina), and tentatively concur that
*Maraapunisaurus* might represent a rebbachisaurid.

15 16 17 18 **4.5. MOR 592**

MOR 592 is a skeleton from the Morrison Formation of Montana that consists of a
braincase, partial dentary, 12 presacral and seven caudal vertebrae, a pelvis, and femur. In a
conference abstract, Wilson and Smith (1996) suggested that MOR 592 might be referable
to *Amphicoelias*, based on the slenderness of the femur, a subcircular femoral cross-section
at midshaft, and reduced pleurocentral openings in the posterior dorsal centra. Those
authors noted that the phylogenetic position of the material was unstable, and the
assignment to *Amphicoelias* was regarded as tentative (J. A. Wilson Mantilla pers. comm. in
Whitlock, 2011a). Whitlock (2011a) revisited this material and considered it to potentially
represent a dicraeosaurid, based on the sharp supraoccipital crest and a symphyseal
tuberosity on the dentary, a diagnosis followed by some later work (e.g. Wedel and Taylor,
2013). However, based on the postcrania, Woodruff and Fowler (2012) suggested that MOR
592 was instead a juvenile morphotype of a diplodocine. Woodruff and Foster (2014) and
Woodruff et al. (2017, 2018) later went further and considered MOR 592 to be a juvenile
specimen of *Diplodocus*.

Regardless of whether MOR 592 is considered a dicraeosaurid or a juvenile *Diplodocus*, it
is generally agreed that it does not belong to *Amphicoelias*. However, the femur of MOR 592
is apparently characterised by a subcircular cross-section at midshaft (Wilson and Smith,
1996), and its distal end appears to be slightly bevelled, with the fibular condyle extending
further distally than its tibial counterpart (Woodruff et al., 2017: fig. 15), both of which are
herein regarded as potential autapomorphies of *Amphicoelias*. Photographs of the femur of
MOR 592, provided by C. Woodruff, reveal that the ratio of the mediolateral to
anteroposterior diameters of its midshaft is >1.3. As such, the femur of MOR 592 lacks the
circular cross-section that characterises that of *Amphicoelias*. The distal bevelling in MOR
592 is less pronounced than in the type of *Amphicoelias*. Furthermore, it is possible that
both the shape of the midshaft and the distal end morphology have been affected by
crushing in MOR 592, with mediolateral compression apparent from the heavily cracked
anterior surface (Woodruff et al., 2017: fig. 15). For now, we exclude MOR 592 from
*Amphicoelias*, but this specimen is clearly in need of detailed study to determine its
taxonomic affinities.

53 54 55 56 57 **5. Discussion**

58 59 60 **5.1. Is *Amphicoelias altus* synonymous with *Diplodocus*?**

Our detailed redescription, combined with recent phylogenetic analyses (Rauhut et al., 2005; Whitlock, 2011a; Mannion et al., 2012, 2019; Tschopp et al., 2015; Tschopp and Mateus, 2017), supports *Amphicoelias altus* as a valid taxon, diagnosed by three autapomorphies. We cannot currently determine whether *Amphicoelias* was a 'basal' diplodocoid (e.g. Mannion et al., 2019) or an apatosaurine (e.g. Tschopp and Mateus, 2017), but it seems unlikely that it was a member of Diplodocinae. As such, we reject recent proposals that *Amphicoelias altus* might be synonymous with *Diplodocus* (Foster, 2003; Woodruff and Foster, 2014; Woodruff, 2019), which also means that we retain the latter as a valid genus.

Although we restrict the known record of *Amphicoelias* to a single individual, it is possible that it was more common than we currently recognise. Given that much of the collection efforts in the Morrison Formation began at the dawn of dinosaur paleontology, many elements were excavated and labelled as one of the handful of well-known sauropod taxa at the time, particularly *Diplodocus* and *Camarasaurus*. Recent years have witnessed the erection of multiple new taxa from material previously referred to existing species, including *Galeamopus* (Tschopp and Mateus, 2017), *Kaatedocus* (Tschopp and Mateus, 2013), and *Smitanosaurus* (Whitlock and Wilson Mantilla, in press), and re-evaluation of historic material has shown that some specimens previously referred to *Diplodocus* and *Apatosaurus* might represent *Galeamopus* instead (Tschopp et al., 2019). It is therefore possible that material pertaining to *Amphicoelias* awaits a collections visitor in a cabinet labelled *Diplodocus*.

5.2. Sauropod diversity in the Morrison Formation

Recent taxonomic studies support the validity of 24 sauropod species assigned to 14 genera in the Upper Jurassic Morrison Formation (Upchurch et al., 2004a,b; Tschopp et al., 2015, 2017; Carpenter, 2018; Whitlock and Wilson Mantilla, in press; Table 1). In a series of papers, Woodruff and colleagues have argued that this diversity is overestimated (see also Prothero, 2019), and that many species considered valid in recent studies are synonymous, primarily interpreted as semaphoronts or 'ontogimorphs' (i.e. growth series) of a smaller number of valid taxa (Woodruff and Fowler, 2012; Woodruff and Foster, 2014; Woodruff et al. 2017, 2018; Woodruff, 2019). These putative synonyms include *Amphicoelias altus*, *Barosaurus lentus*, *Haplocanthosaurus priscus*, *Kaatedocus siberi*, and *Suuwassea emilieae*. In addition, Woodruff and colleagues considered it unlikely that this high diversity of megaherbivores could have all coexisted in one geographic region. Below we evaluate these claims that sauropod diversity in the Morrison Formation is inflated.

5.2.1. Are some Morrison sauropod species ontogimorphs?

There is a global dearth of sauropod remains for which we have a growth series that can be unambiguously referred to a single species, making it difficult to test hypotheses pertaining to taxonomic inflation resulting from ontogeny. The best-known test case of ontogenetic variation in a sauropod is the dwarf macronarian *Europasaurus holgeri* from the Late Jurassic of Germany, which is represented by more than a dozen individuals from one locality, with remains representing juvenile through to adult stages of growth (Sander et al. 2006). Carballido and Sander (2014) demonstrated that most anatomical features, including

autapomorphies, were present by the late immature stage of growth, and that these
provide the same phylogenetic signal as adult individuals. Although only based on one
species, and a potentially unusual one because of its dwarf status, *Europasaurus* suggests
that we might expect that sauropod anatomy does not change dramatically once individuals
reach subadult growth stages, and that autapomorphies will neither develop nor be lost
beyond that stage. Similar inferences can be made from recent descriptions of juvenile
material referred to the monospecific genus *Barosaurus* (Melstrom et al., 2016; Hanik et al.,
2017), where clearly defined autapomorphies of the taxon appear early in osteological
development.

The most robust test of this hypothesis in the Morrison Formation comes from Ikejiri et
al. (2005). Two adult individuals of *Camarasaurus lentus* at different osteological stages
were found in one quarry and compared with two subadult specimens from elsewhere in
the formation. Although Ikejiri et al. (2005) noted several anatomical differences between
the two adult individuals, these were typical of the features associated with advanced age
or repetitive stress (e.g. heterotopic ossification; Meyers et al., 2019). All four semaphoronts
in the study were clearly referable to the same species (and distinguishable from other
species) based on autapomorphies, as well as the overwhelming majority of anatomical
features. The limited evidence from *Camarasaurus* and *Europasaurus* suggests that
sauropods did not radically change their anatomy once they reached mature stages of
growth, undermining claims that species such as *Barosaurus lentus* and *Amphicoelias altus*,
known from adult remains, represent growth stages in *Diplodocus* ontogeny.

One consequence of this is that we might therefore expect that subadult sauropod
individuals should be unaffected by stemward slippage when included in phylogenetic
analyses, given that they should possess a character suite that is extremely close to that of
adult members of their species (e.g. Carballido and Sander, 2014). None of the type
specimens of Morrison Formation sauropod species regarded as potential ontogimorphs are
interpreted as juveniles (Wedel and Taylor, 2013; Hedrick et al., 2014), and it is ultimately
unlikely that many of them are truly subadult in a modern-day biological sense (e.g. Hone et
al., 2016). As such, we find little evidence to support the claim that species such as
*Haplocanthosaurus priscus* are immature individuals of other species when they are clearly
known from adult specimens (Wedel and Taylor, 2013). However, even allowing for the
possibility that some taxa are based on subadult material (e.g. *Kaatedocus*, *Suuwassea*), this
still necessitates major increases in the number of evolutionary steps for phylogenetic
analyses to recover them clustering with their proposed senior synonyms (Wedel and
Taylor, 2013). Furthermore, even the inclusion of juvenile specimens does not necessarily
result in their stemward slippage. In their diplodocid-focused phylogenetic analysis, Tschopp
et al. (2015) recovered the juvenile type specimen of *Brontosaurus* ('*Elosaurus*') *parvus* as
most closely related to the adult apatosaurine specimen found in association (Peterson and
Gilmore, 1902; Gilmore, 1936), despite the presence of several plesiomorphic features that
resulted in character scores variation between the two individuals (see also Upchurch et al.,
2004b). As such, there is no evidence that the phylogenetic positions of taxa such as
*Amphicoelias*, *Barosaurus*, *Haplocanthosaurus*, *Kaatedocus*, or *Suuwassea*, are incorrect to
the point that they have been misidentified as distinct from *Diplodocus* or other Morrison
taxa. It is important to note that whether apatosaurine species belong to *Apatosaurus* or
*Brontosaurus* is ultimately irrelevant to discussions of diversity (cf. Prothero, 2019), given
the arbitrary nature of genera.

The above does not preclude the possibility that some phylogenetically informative
features can have an ontogenetic signal too. For example, diplodocine individuals might
possess a postparietal foramen that is lost during ontogeny (Woodruff et al., 2017) *and* its
presence is also a synapomorphy of some dicraeosaurids (Whitlock, 2011a). Similarly,
ontogenetically old individuals of *Camarasaurus* might incorporate a sixth sacral vertebra
(e.g. Tidwell et al., 2005) *and* a 6-vertebra sacrum also characterises Somphospondyli
(Wilson and Sereno, 1998). However, although these ontogenetic features might introduce
some degree of 'directed noise' (Tschopp and Upchurch, 2019), they do not outweigh the
overwhelming phylogenetic signal when operational taxonomic units are not early juveniles,
at least in sauropods. Returning to *Europasaurus*, this taxon is universally recovered as an
early diverging macronarian (e.g. Sander et al., 2006; D'Emic, 2012; Carballido and Sander,
2014; Mannion et al., 2019). However, likely via pedomorphic retention, *Europasaurus*
possesses several features (including a postparietal foramen) that are inconsistent with a
placement in Macronaria or even Neosauropoda (Carballido and Sander, 2014; Marpmann
et al., 2015), but these do not result in stemward slippage (Mannion et al., 2017). We do not
disagree that ontogeny is an important issue and one that needs to be considered, but there
is currently no evidence to support the claim that Morrison sauropod diversity is
overestimated as a result of it. As also concluded by Wedel and Taylor (2013), we should be
wary of placing too much emphasis on 'critical' phylogenetic characters (*sensu* Woodruff
and Fowler, 2012).

25 26 27 28 29 **5.2.2. Was apparent Morrison Formation sauropod diversity too high to be ecologically** 30 **viable?**

The notion of 24 co-occurring megaherbivore species is, at face value, potentially difficult
to accept, with Woodruff (2019: pp. 93, 105) considering it "unlikely" and "ecologically
taxing". Although the Morrison Formation covers an area exceeding 1.2 million km² (Dodson
et al., 1980), it is possible that this would have been unable to support enough viable
individuals of each species (though see below). However, the idea that all of these species
co-occurred with one another is misleading. Firstly, none of these 24 species are found
throughout the spatial extent of the Morrison Formation. As noted above, many species are
limited to a small number (<5) of occurrences (e.g. *Brachiosaurus altithorax*,
*Haplocanthosaurus priscus*) or are currently known from a single locality (e.g.
*Haplocanthosaurus delfsi*, *Suuwassea emilieae*). However, even species known from
abundant remains show evidence for some degree of geographical restriction (e.g. Foster,
2003; Whitlock et al., 2018; Tschopp et al., 2019). Furthermore, numerous fossiliferous
Morrison localities have been sampled extensively and yet none contain more than five
sympatric sauropod species (Foster, 2003). These distributional data alone suggest that the
24 Morrison sauropod species did not all co-occur.

Secondly, the Morrison Formation was deposited over a period of at least seven million
52 years (Turner and Peterson, 1999; Trujillo and Kowallis, 2015; Maidment and Muxworthy,
2019). None of the 24 sauropod species are recovered throughout the formation's full
temporal extent and many species were clearly not contemporaneous with others (e.g.
Turner and Peterson, 1999; Foster, 2003; Tschopp et al., 2015), although a substantial
proportion of localities with radiometric dates are approximately coeval (Whitlock et al.,
2018). Despite framing his study with the 'problem' of having 24 contemporaneous
megaherbivore species to support his claim that Morrison sauropod diversity is

overestimated, Woodruff (2019: p. 99–105) was clearly aware that this is incorrect. For
example, Woodruff (2019: fig. 5) showed the Morrison Formation divided into six ‘biozones’,
in which the most diverse of these contains 12 distinct taxa (and note that three of these
are specifically indeterminate members of contemporaneous genera). It is not clear if this
number of species was also considered ecologically problematic by Woodruff (2019), but no
evidence was provided in that study (or others) to support the view that high sauropod
diversity was unviable.

[revised manuscript text omitted]

32 33 ACKNOWLEDGEMENTS

We thank Mark Norell for enabling us to study and 3D scan *Amphicoelias altus* at the AMNH,
as well as Carl Mehling, Verne Lee, Bruce Javors, and the late Jack Conrad for their help
accessing the material. We are also grateful to Mike Eklund and Bruce Javors for
contributing photographs of AMNH specimens, and to Carolyn Merrill for producing the 3D
scans. Cary Woodruff also kindly provided us with photographs of the femur of MOR 592.
We also acknowledge the Willi Hennig Society, which has sponsored the development and
free distribution of TNT. PDM.’s research was supported by a Royal Society University
Research Fellowship (UF160216). ET’s contribution was supported by an AMNH Division of
Paleontology Postdoctoral Fellowship.

48 REFERENCES

**Apesteguía S, Gallina PA, Haluza A. 2010.** Not just a pretty face: anatomical peculiarities in
the postcranium of rebbachisaurids (Sauropoda: Diplodocoidea). *Historical Biology* **22**:
165–174.
**Bakker RT. 1971.** Ecology of the brontosaurus. *Nature* **229**: 172–174.
**Barrett PM. 2014.** Paleobiology of herbivorous dinosaurs. *Annual Review of Earth and*
*Planetary Sciences* **42**: 207–30.
**Bonaparte JF. 1986.** The early radiation and phylogenetic relationships of the Jurassic
sauropod dinosaurs, based on vertebral anatomy. In: Padian K, ed. *The Beginning of the*
Age of Dinosaurs. Cambridge: Cambridge University Press, 247–258.

**Brochu CA, Sumrall CD. 2020.** Modern cryptic species and crocodylian diversity in the fossil
record. *Zoological Journal of the Linnean Society* (doi: 10.1093/zoolinnean/zlaa039).
- **Button DJ, Barrett PM, Rayfield EJ. 2017.** Craniodental functional evolution in
sauropodomorph dinosaurs. *Paleobiology* **43**: 435–462.
- **Button DJ, Zanno LE. 2020.** Repeated Evolution of Divergent Modes of Herbivory in Non-
avian Dinosaurs. *Current Biology* **30**: 158–168.
- **Calvo JO, Salgado L. 1995.** *Rebbachisaurus tessonei* sp. nov. a new Sauropoda from the
Albian-Cenomanian of Argentina; new evidence on the origin of the Diplodocidae. *Gaia* **11**:
13–33.
- **Carballido JL, Salgado L, Pol D, Canudo JI, Garrido A. 2012.** A new basal rebbachisaurid
(Sauropoda, Diplodocoidea) from the Early Cretaceous of the Neuquén Basin; evolution
and biogeography of the group. *Historical Biology* **24**: 631–654.
- **Carballido JL, Sander PM. 2014.** Postcranial axial skeleton of *Europasaurus holgeri*
(Dinosauria, Sauropoda) from the Upper Jurassic of Germany—implications for sauropod
ontogeny and phylogenetic relationships of basal Macronaria. *Journal of Systematic
Palaeontology* **12**: 335–387.
- **Carpenter K. 1998.** Vertebrate biostratigraphy of the Morrison Formation near Cañon City,
Colorado. *Modern Geology* **23**: 407–426.
- **Carpenter K. 2006.** Biggest of the big—a critical re-evaluation of the megasauropod
*Amphicoelias fragillimus*. *New Mexico Museum of Natural History and Science Bulletin* **36**:
131–137.
- **Carpenter K. 2018.** *Maraapunisaurus fragillimus*, n.g. (formerly *Amphicoelias fragillimus*), a
basal rebbachisaurid from the Morrison Formation (Upper Jurassic) of Colorado. *Geology
of the Intermountain West* **5**: 227–244.
- **Chiarenza AA, Mannion PD, Lunt DJ, Farnsworth A, Jones LA, Kelland S-J, Allison PA. 2019.**
Ecological niche modelling does not support climatically-driven dinosaur diversity decline
before the Cretaceous/Paleogene mass extinction. *Nature Communications* **10**: 1091.
- **Christiansen P. 2000.** Feeding mechanisms of the sauropod dinosaurs *Brachiosaurus*,
*Camarasaurus*, *Diplodocus*, and *Dicraeosaurus*. *Historical Biology* **14**: 137–152.
- **Cope ED. 1877a.** On *Amphicoelias*, a genus of Saurians from the Dakota epoch of Colorado.
*Proceedings of the American Philosophical Society* **17**: 242–246.
- **Cope ED. 1877b.** On a gigantic saurian from the Dakota epoch of Colorado. *Palaeontological
Bulletin* **25**: 5–10.
- **Cope ED. 1878.** A new species of *Amphicoelias*. *American Naturalist* **12**: 563–565.
- **D’Emic MD. 2012.** The early evolution of titanosauriform sauropod dinosaurs. *Zoological
Journal of the Linnean Society* **166**: 624–671.
- **D’Emic MD, Whitlock JA, Smith KM, Fisher DC, Wilson JA. 2013.** Evolution of high tooth
replacement rates in sauropod dinosaurs. *PLoS ONE* **8**: e69235.
- **Dodson P, Behrensmeyer AK, Bakker RT, Mcintosh JS. 1980.** Taphonomy and paleoecology
of the dinosaur beds of the Jurassic Morrison Formation. *Paleobiology* **6**: 208–232.
- **Eklund MJ, Aase AK, Bell CJ. 2018.** Progressive Photonics: Methods and applications of
sequential imaging using visible and non-visible spectra to enhance data-yield and
facilitate forensic interpretation of fossils. *Journal of Paleontological Techniques* **20**: 1–36.
- **Farlow JO, Coroian ID, Foster JR. 2010.** Giants on the landscape: modelling the abundance
of megaherbivorous dinosaurs of the Morrison Formation (Late Jurassic, western USA).
*Historical Biology* **22**: 403–429.

- Filla BJ, Redman PD. 1994.** *Apatosaurus yahnahpin*: a preliminary description of a new species of diplodocid dinosaur from the Late Jurassic Morrison Formation of Southern Wyoming, the first sauropod dinosaur found with a complete set of “belly ribs”. In: Nelson GE, ed. *The Dinosaurs of Wyoming. Wyoming Geological Association 44th annual field conference guidebook*. Casper: Wyoming Geological Association, 159–178.
- Fiorillo AR. 1998.** Dental microwear patterns of the sauropod dinosaurs *Camarasaurus* and *Diplodocus*—evidence for resource partitioning in the Late Jurassic of North America. *Historical Biology* **13**: 1–16.
- Foster JR. 2003.** Paleoecological analysis of the vertebrate fauna of the Morrison Formation (Upper Jurassic), Rocky Mountain region, U.S.A. *New Mexico Museum of Natural History and Science Bulletin* **23**: 1–95.
- Foster JR. 2007.** *Jurassic west—the dinosaurs of the Morrison Formation and their world*. Bloomington: Indiana University Press, 389 p.
- Gallina PA, Apesteguía S, Haluza A, Canale JI. 2014.** A diplodocid sauropod survivor from the Early Cretaceous of South America. *PLoS ONE* **9**: e97128.
- Gee CT. 2011.** Dietary options for the sauropod dinosaurs from an integrated botanical and paleobotanical perspective. In: Klein N, Remes K, Gee CT, Sander PM, eds. *Biology of the Sauropod Dinosaurs: Understanding the Life of Giants*. Bloomington, IN: Indiana University Press, 34–56.
- Gillette DG. 1991.** *Seismosaurus halli*, gen. et sp. nov., a new sauropod dinosaur from the Morrison Formation (Upper Jurassic/Lower Cretaceous) of New Mexico, USA. *Journal of Vertebrate Paleontology* **11**: 417–433.
- Gilmore CW. 1925.** A nearly complete articulated skeleton of *Camarasaurus*, a saurischian dinosaur from the Dinosaur National Monument, Utah. *Memoirs of the Carnegie Museum* **10**: 347–384.
- Gilmore CW. 1936.** Osteology of *Apatosaurus*: with special reference to specimens in the Carnegie Museum. *Memoirs of the Carnegie Museum* **11**: 175–300.
- Hanik GM, Lamanna MC, Whitlock JA. 2017.** A juvenile specimen of *Barosaurus* Marsh, 1890 (Sauropoda: Diplodocidae) from the Upper Jurassic Morrison Formation of Dinosaur National Monument, Utah, USA. *Annals of Carnegie Museum* **84**: 253–263.
- Harris JD, Dodson P. 2004.** A new diplodocoid sauropod dinosaur from the Upper Jurassic Morrison Formation of Montana, USA. *Acta Palaeontologica Polonica* **49**: 197–210.
- Hatcher JB. 1901.** *Diplodocus* Marsh, its osteology, taxonomy, and probable habits, with a restoration of the skeleton. *Memoirs of the Carnegie Museum* **1**: 1–64.
- Hatcher JB. 1903a.** A new sauropod dinosaur from the Jurassic of Colorado. *Proceedings of the Biological Society of Washington* **16**: 1–2.
- Hatcher JB. 1903b.** Osteology of *Haplocanthosaurus*, with description of a new species, and remarks on the probable habits of the Sauropoda and the age and origin of the *Atlantosaurus* beds. *Memoirs of the Carnegie Museum* **2**: 1–72.
- Hedrick BP, Tumarkin-Deratzian AR, Dodson P. 2014.** Bone microstructure and relative age of the holotype specimen of the diplodocoid sauropod dinosaur *Suuwassea emilieae*. *Acta Palaeontologica Polonica* **59**: 2295–304.
- Holland WJ. 1915.** A new species of *Apatosaurus*. *Annals of the Carnegie Museum* **10**: 143–145.
- Holland WJ. 1924.** The skull of *Diplodocus*. *Memoirs of the Carnegie Museum* **9**: 379–403.
- Hone DWE, Farke AA, Wedel MJ. 2016.** Ontogeny and the fossil record: what, if anything, is an adult dinosaur? *Biology Letters* **12**: 20150947.

- Hotton CL, Baghai-Riding NL. 2010.** Palynological evidence for conifer dominance within a heterogenous landscape in the Late Jurassic Morrison Formation, U.S.A. In: Gee CT, ed. *Plants in Mesozoic Time*. Bloomington, IN: Indiana University Press, 295–328.
- Huene F von. 1908.** Die Dinosaurier der Europäischen Triasformation mit Berücksichtigung der Ausseuropäischen Vorkommnisse. *Geologische und Palaeontologische Abhandlungen 1908 (supplement 1)*: 1–419.
- Ikejiri T, Tidwell V, Trexler DL. 2005.** New adult specimens of *Camarasaurus lentus* highlight ontogenetic variation within the species. In: Tidwell V, Carpenter K, eds. *Thunder Lizards: The Sauropodomorph Dinosaurs*. Bloomington, IN: Indiana University Press, 154–179.
- Jensen JA. 1985.** Three new sauropod dinosaurs from the Upper Jurassic of Colorado. *The Great Basin Naturalist* **45**: 697–709.
- Jensen JA. 1988.** A fourth new sauropod dinosaur from the Upper Jurassic of the Colorado Plateau and sauropod bipedalism. *The Great Basin Naturalist* **48**: 121–145.
- Linares Soriano MA, Carrascosa Moliner MB. 2016.** Consolidation of bone material: chromatic evolution of resins after UV accelerated aging. *Journal of Paleontological Techniques* **15**: 46–67.
- Lovelace DM, Hartman SA, Wahl WR. 2008.** Morphology of a specimen of *Supersaurus* (Dinosauria, Sauropoda) from the Morrison Formation of Wyoming, and a re-evaluation of diplodocid phylogeny. *Arquivos do Museu Nacional, Rio de Janeiro* **65**: 527–544.
- Maidment SCR, Muxworthy A. 2019.** A chronostratigraphic framework for the Upper Jurassic Morrison Formation, western U.S.A. *Journal of Sedimentary Research* **89**: 1017–1038.
- Mannion PD, Allain R, Moine O. 2017.** The earliest known titanosauriform sauropod dinosaur and the evolution of Brachiosauridae. *PeerJ* **5**: e3217.
- Mannion PD, Upchurch P, Barnes RN, Mateus O. 2013.** Osteology of the Late Jurassic Portuguese sauropod dinosaur *Lusotitan atalaiensis* (Macronaria) and the evolutionary history of basal titanosauriforms. *Zoological Journal of the Linnean Society* **168**: 98–206.
- Mannion PD, Upchurch P, Mateus O, Barnes RN, Jones MEH. 2012.** New information on the anatomy and systematic position of *Dinheirosaurus lourinhanensis* (Sauropoda: Diplodocoidea) from the Late Jurassic of Portugal, with a review of European diplodocoids. *Journal of Systematic Palaeontology* **10**: 521–551.
- Mannion PD, Upchurch P, Schwarz D, Wings O. 2019.** Taxonomic affinities of the putative titanosaurs from the Late Jurassic Tendaguru Formation of Tanzania: phylogenetic and biogeographic implications for eusauropod dinosaur evolution. *Zoological Journal of the Linnean Society* **185**: 784–909.
- Marpmann JS, Carballido JL, Sander PM, Knötschke N. 2015.** Cranial anatomy of the Late Jurassic dwarf sauropod *Europasaurus holgeri* (Dinosauria, Camarasauromorpha): ontogenetic changes and size dimorphism. *Journal of Systematic Palaeontology* **13**: 221–263.
- Marsh OC. 1877.** Notice of new dinosaurian reptiles from the Jurassic Formation. *American Journal of Science (Series 3)* **14**: 515–516.
- Marsh OC. 1878.** Principal characters of American Jurassic dinosaurs. Part I. *American Journal of Science (Series 3)* **16**: 411–416.
- Marsh OC. 1879.** Notice of new Jurassic dinosaurs. *American Journal of Science (Series 3)* **18**: 501–505.

- Marsh OC. 1884.** Principal characters of American Jurassic dinosaurs. Part VII. On the Diplodocidae, a new family of the Sauropoda. *American Journal of Science (Series 3)* **27**: 161–167.
- Marsh OC. 1889.** Comparison of the principal forms of the Dinosauria of Europe and America. *American Journal of Science (Series 3)* **37**: 323–330.
- Marsh OC. 1890.** Description of new dinosaurian reptiles. *American Journal of Science (Series 3)* **39**: 81–86.
- Marsh OC. 1896.** The Dinosaurs of North America. *United States Geological Survey* **55**: 133–244.
- McIntosh JS. 1990.** Sauropoda. In: Weishampel DB, Dodson P, Ósmolska H, eds. *The Dinosauria, First edition*. Berkeley: University California Press, 345–401.
- McIntosh JS. 1998.** New information about the Cope collection of sauropods from Garden Park, Colorado. *Modern Geology* **23**: 481–506.
- McIntosh JS, Coombs WP, Russell DA. 1992.** A new diplodocid sauropod (Dinosauria) from Wyoming, U.S.A. *Journal of Vertebrate Paleontology* **12**: 158–167.
- McIntosh JS, Williams ME. 1988.** A new species of sauropod dinosaur, *Haplocanthosaurus delfsi* sp. nov., from the Upper Jurassic Morrison Fm. of Colorado. *Kirtlandia* **43**: 3–26.
- Melstrom KM, D’Emic MD, Chure DJ, Wilson JA. 2015.** A juvenile sauropod dinosaur from the Late Jurassic of Utah, USA, with evidence of an avian style air-sac system. *Journal of Vertebrate Paleontology* **36**: e1111898.
- Meyers C, Lisiecki J, Miller S, Levin A, Fayad L, Ding C, Sono T, McCarthy E, Levi B, James AW. 2019.** Heterotopic ossification: a comprehensive review. *JBMR Plus* **3**: e10172.
- Mook CC. 1917.** Criteria for the determination of species in the Sauropoda, with description of a new species of *Apatosaurus*. *Bulletin of the American Museum of Natural History* **37**: 355–358.
- Nopcsa F. 1928.** The genera of reptiles. *Palaeobiologica* **1**: 163–188.
- Osborn HF, Mook CC. 1921.** *Camarasaurus*, *Amphicoelias*, and other sauropods of Cope. *Memoirs of the American Museum of Natural History New Series* **3**: 247–387.
- Otero A. 2010.** The appendicular skeleton of *Neuquensaurus*, a Late Cretaceous saltasaurine sauropod from Patagonia, Argentina. *Acta Palaeontologica Polonica* **55**: 399–426.
- Peterson OA, Gilmore CW. 1902.** *Elosaurus parvus*; a new genus and species of the Sauropoda. *Annals of the Carnegie Museum* **1**: 490–499.
- Plotnick RE, Smith FE, Lyons SK. 2016.** The fossil record of the sixth extinction. *Ecology Letters* **19**: 546–553.
- Prothero DR. 2019.** *The Story of the Dinosaurs in 25 Discoveries*. New York: Columbia University Press, 488 pp.
- Riggs ES. 1903.** *Brachiosaurus altithorax*, the largest known dinosaur. *American Journal of Science (Series 4)* **15**: 299–306.
- Salgado L, Coria RA, Calvo JO. 1997.** Evolution of titanosaurid sauropods. I: phylogenetic analysis based on the postcranial evidence. *Ameghiniana* **34**: 3–32.
- Sander PM, Mateus O, Laven T, Knötschke N. 2006.** Bone histology indicates insular dwarfism in a new Late Jurassic sauropod dinosaur. *Nature* **441**: 739–741.
- Stevens KA, Parrish JM. 2005.** Neck posture, dentition, and feeding strategies in Jurassic sauropod dinosaurs. In: Tidwell V, Carpenter K, eds. *Thunder Lizards: The Sauropodomorph Dinosaurs*. Bloomington, IN: Indiana University Press, 212–232.

**Tidwell V, Stadtman K, Shaw A. 2005.** Age-related characteristics found in a partial pelvis of
*Camarasaurus*. In: Tidwell V, Carpenter K, eds. *Thunder Lizards: The Sauropodomorph*
*Dinosaurs*. Bloomington, IN: Indiana University Press, 180–186.
- **Trujillo KC, Kowallis BJ. 2015.** Recalibrated legacy $^{40}\text{Ar}/^{39}\text{Ar}$ ages for the Upper Jurassic
Morrison Formation, Western Interior, U.S.A. *Geology of the Intermountain West* **2**: 1–8.
- **Tschopp E, Brinkman D, Henderson J, Turner M, Mateus O. 2018.** Considerations on the
replacement of a type species in the case of the sauropod dinosaur *Diplodocus* Marsh,
1878. *Geology of the Intermountain West* **5**: 245–262.
- **Tschopp E, Maidment SCR, Lamanna MC, Norell MA. 2019.** Reassessment of a Historical
Collection of Sauropod Dinosaurs from the Northern Morrison Formation of Wyoming,
with Implications for Sauropod Biogeography. *Bulletin of the American Museum of Natural*
*History* **437**: 1–79.
- **Tschopp E, Mateus O. 2013.** The skull and neck of a new flagellicaudatan sauropod from the
Morrison Formation and its implication for the evolution and ontogeny of diplodocid
dinosaurs. *Journal of Systematic Palaeontology* **11**: 853–888.
- **Tschopp E, Mateus O. 2017.** Osteology of *Galeamopus pabsti* sp. nov. (Sauropoda:
Diplodocidae), with implications for neurocentral closure timing, and the cervico-dorsal
transition in diplodocids. *PeerJ* **5**: e3179.
- **Tschopp E, Mateus O, Benson RBJ. 2015.** A specimen-level phylogenetic analysis and
taxonomic revision of Diplodocidae (Dinosauria, Sauropoda). *PeerJ* **3**: e857.
- **Tschopp E, Mehling C, Norell MA. In Press.** Reconstructing the specimens and history of the
Howe Quarry (Upper Jurassic Morrison Formation; Wyoming, USA). *American Museum*
*Novitates*.
- **Tschopp E, Upchurch P. 2019.** The challenges and potential utility of phenotypic specimen-
level phylogeny based on maximum parsimony. *Earth and Environmental Science*
*Transactions of The Royal Society of Edinburgh* **109**: 301–323.
- **Turner CE, Peterson F. 1999.** Biostratigraphy of dinosaurs in the Upper Jurassic Morrison
Formation of the Western Interior, USA. *Utah Geological Survey Miscellaneous*
*Publications* **99**: 77–114.
- **Tütken T. 2011.** The diet of sauropod dinosaurs: implications of carbon isotope analysis on
teeth, bones, and plants. In: Klein N, Remes K, Gee CT, Sander PM, eds. *Biology of the*
*Sauropod Dinosaurs: Understanding the Life of Giants*. Bloomington, IN: Indiana University
Press, 57–79.
- **Upchurch P. 1998.** The phylogenetic relationships of sauropod dinosaurs. *Zoological Journal*
*of the Linnean Society* **124**: 43–103.
- **Upchurch P, Barrett PM. 2000.** The evolution of sauropod feeding mechanisms. In: Sues H-
D, ed. *Evolution of Herbivory in Terrestrial Vertebrates—Perspectives from the Fossil*
*Record*. Cambridge: Cambridge University Press, 79–122.
- **Upchurch P, Barrett PM, Dodson P. 2004a.** Sauropoda. In: Weishampel DB, Dodson P,
Osmólska H, eds. *The Dinosauria, Second edition*. Berkeley: University of California Press,
259–324.
- **Upchurch P, Tomida Y, Barrett PM 2004b.** A new specimen of *Apatosaurus ajax*
(Sauropoda: Diplodocidae) from the Morrison Formation (Upper Jurassic) of Wyoming,
USA. *National Science Museum Monographs* **26**: 1–118.
- **Wedel MJ, Taylor MP. 2013.** Neural spine bifurcation in sauropod dinosaurs of the Morrison
Formation: ontogenetic and phylogenetic implications. *PalArch's Journal of Vertebrate*
*Palaeontology* **10**: 1–34.

**Whitlock JA. 2011a.** A phylogenetic analysis of Diplodocoidea (Saurischia: Sauropoda).
*Zoological Journal of the Linnean Society* **161**: 872–915.
- **Whitlock JA. 2011b.** Inferences of diplodocoid (Sauropoda: Dinosauria) feeding behavior
from snout shape and microwear analyses. *PLoS ONE* **6**: e18304.
- **Whitlock JA, Trujillo KC, Hanik GM. 2018.** Assemblage-level structure in Morrison
Formation dinosaurs, Western Interior, USA. *Geology of the Intermountain West* **5**: 9–22.
- **Whitlock JA, Wilson Mantilla JA. In press.** The Late Jurassic sauropod dinosaur
“*Morosaurus*” *agilis* Marsh, 1889 reexamined and reinterpreted as a dicraeosaurid. *Journal*
*of Vertebrate Paleontology*.
- **Wiersma K, Sander PM. 2016.** The dentition of a well-preserved specimen of *Camarasaurus*
sp.: implications for function, tooth replacement, soft part reconstruction, and food intake.
*Paläontologische Zeitschrift* **91**: 145–161.
- **Wilson JA. 1999.** A nomenclature for vertebral laminae in sauropods and other saurischian
dinosaurs. *Journal of Vertebrate Paleontology* **19**: 639–653.
- **Wilson JA. 2002.** Sauropod dinosaur phylogeny: critique and cladistic analysis. *Zoological*
*Journal of the Linnean Society* **136**: 217–276.
- **Wilson JA, Allain R. 2015.** Osteology of *Rebbachisaurus garasbae* Lavocat, 1954, a
diplodocoid (Dinosauria, Sauropoda) from the early Late Cretaceous-aged Kem Kem beds
of southeastern Morocco. *Journal of Vertebrate Paleontology* **35**: e1000701.
- **Wilson JA, Sereno PC. 1998.** Early evolution and higher-level phylogeny of sauropod
dinosaurs. *Society of Vertebrate Paleontology Memoir* **5**: 1–68.
- **Wilson JA, Smith MB. 1996.** New remains of *Amphicoelias* Cope (Dinosauria: Sauropoda)
from the Upper Jurassic of Montana and diplodocoid phylogeny. *Journal of Vertebrate*
*Paleontology* **16**: 73A.
- **Wilson JA, D’Emic MD, Ikejiri T, Moacdieh EM, Whitlock JA. 2011.** A nomenclature for
vertebral fossae in sauropods and other saurischian dinosaurs. *PLoS ONE* **6**: e17114.
- **Woodruff DC. 2019.** What Factors Influence our Reconstructions of Morrison Formation
Sauropod Diversity? *Geology of the Intermountain West* **6**: 93–112.
- **Woodruff DC, Carr TD, Storrs GW, Waskow K, Scannella JB, Norden KK, Wilson JP. 2018.**
The smallest diplodocid skull reveals cranial ontogeny and growth-related dietary changes
in the largest dinosaurs. *Scientific Reports* **8**: 14341.
- **Woodruff DC, Foster JR. 2014.** The fragile legacy of *Amphicoelias fragillimus* (Dinosauria:
Sauropoda; Morrison Formation – latest Jurassic). *Volumina Jurassica* **12**: 211–220.
- **Woodruff DC, Fowler DW. 2012.** Ontogenetic influence on neural spine bifurcation in
Diplodocoidea (Dinosauria: Sauropoda): a critical phylogenetic character. *Journal of*
*Morphology* **273**: 754–764.
- **Woodruff DC, Fowler DW, Horner JR. 2017.** A new multi-faceted framework for deciphering
diplodocid ontogeny. *Palaeontologia Electronica* **20**: 43A.

TABLES

**Table 1.** Currently recognized sauropod genera and species in the Upper Jurassic Morrison
Formation of North America. *Dyslocosaurus polyonychius* potentially represents an
additional valid sauropod species, but its stratigraphic provenance remains uncertain
(McIntosh et al., 1992; Tschopp et al., 2015). ‘*Apatosaurus*’ *minimus* might also represent a
distinct sauropod species (Mook, 1917), although its affinities require further evaluation
(Upchurch et al., 2004a; Tschopp et al., 2015). Note that the validity of *Diplodocus longus*

(highlighted with an asterisk) is disputed and it is unlikely that this species can be diagnosed (Tschopp et al. 2018 [and references therein]).

Taxon	Authors
Amphicoelias altus	Cope, 1877a
Apatosaurus ajax (type)	Marsh, 1877
Apatosaurus louisae	Holland, 1915
Barosaurus lentus	Marsh, 1890
Brachiosaurus altithorax	Riggs, 1903
Brontosaurus excelsus (type)	Marsh, 1879
Brontosaurus parvus	Peterson and Gilmore, 1902
Brontosaurus yahnahpin	Filla and Redman, 1994
Camarasaurus grandis	Marsh, 1877
Camarasaurus lentus	Marsh, 1889
Camarasaurus lewisi	Jensen, 1988
Camarasaurus supremus	Cope, 1877b
Diplodocus carnegii	Hatcher, 1901
Diplodocus longus (type)*	Marsh, 1878
Diplodocus hallorum	Gillette, 1991
Galeamopus hayi (type)	Holland, 1924; Tschopp et al., 2015
Galeamopus pabsti	Tschopp and Mateus, 2017
Haplocanthosaurus delfsi	McIntosh and Williams, 1988
Haplocanthosaurus priscus (type)	Hatcher, 1903
Kaatedocus siberi	Tschopp and Mateus, 2013
Maraapunisaurus fragillimus	Cope, 1878; Carpenter, 2018
Smitanosaurus agilis	Marsh, 1889; Whitlock and Wilson Mantilla, in press
Supersaurus vivianae	Jensen, 1985
Suuwassea emilieae	Harris and Dodson, 2004

Table 2. Measurements of holotypic dorsal vertebrae of *Amphicoelias altus* (AMNH FARB 5764). DvA = middle–posterior dorsal vertebra; DvB = posterior dorsal vertebra. Measurements in millimetres.

Dimension	DvA	DvB
Anteroposterior length of centrum	~220	242
Dorsoventral height of anterior end of centrum	–	251
Dorsoventral height of posterior end of centrum	~274	266
Mediolateral width of posterior end of centrum	~231	258
Dorsoventral height of neural arch	~239	212
Dorsoventral height of neural spine	–	620
Anteroposterior length of neural spine (near base)	–	158
Mediolateral width of neural spine (near base)	–	72
Maximum mediolateral width of neural spine (near apex)	–	148
Distance from midline to lateral tip of diapophysis	–	303
Dorsoventral height at lateral tip of diapophysis	–	111

Table 3. Measurements of holotypic femur of *Amphicoelias altus* (AMNH FARB 5764). Measurements in millimetres.

Dimension	Measurement
Proximodistal length	~1700
Distance from proximal end to distal tip of fourth trochanter	~900
Maximum diameter of shaft at distal end of proximal half	233
Diameter of shaft at distal end of proximal half, perpendicular to maximum diameter	222
Midshaft mediolateral width	219
Midshaft anteroposterior length	238
Mediolateral width of shaft at two-thirds of femur length (from proximal end)	228
Anteroposterior length of shaft at two-thirds of femur length (from proximal end)	211
Minimum shaft circumference	701
Distal end anteroposterior length	~354
Mediolateral width of tibial condyle	166
Mediolateral width of fibular condyle	~187

Table 4. Measurements of specimens previously referred to *Amphicoelias altus*: AMNH FARB 5764 (tooth), 5764a (scapula, coracoid, ulna, femur) and 5761 (humerus). Measurements in millimetres.

Element	Dimension	Measurement
Tooth	Base of crown maximum mesiodistal width	25
	Base of crown maximum labiolingual width	18
Scapula	Anteroposterior length as preserved	1450
	Dorsoventral height of acromion	940
	Minimum dorsoventral height of scapular blade	270
Coracoid	Maximum dorsoventral height	710
	Maximum anteroposterior length	450
Humerus	Proximodistal length	1140
	Maximum mediolateral width of proximal end	~510
	Mediolateral width at midshaft	215
	Anteroposterior diameter at midshaft	125
	Mediolateral width at distal end	370
Ulna	Proximodistal length	1035
	Mediolateral width of proximal end	265
Femur	Proximodistal length as preserved	840
	Mediolateral width at midshaft	190
	Anteroposterior diameter at midshaft	160
	Mediolateral width of distal end	360

FIGURE CAPTIONS

**Figure 1.** *Amphicoelias altus* holotype (AMNH FARB 5764) middle–posterior dorsal vertebra
in anterior (A, E, I), right lateral (B, F, J), posterior (C, G, K), left lateral (D, H), and ventral (L,
anterior towards top) views. Grey tone indicates anatomy reconstructed or obscured by
matrix; hatched areas indicate broken surfaces. Abbreviations: D, diapophysis; CPOL,
centropostzygapophyseal lamina; CPRL, centroprezygapophyseal lamina; P, parapophysis;
PCDL, posterior centrodiapophyseal lamina; PCPL, posterior centroparapophyseal lamina;
PRDL, prezygodiapophyseal lamina; PRZ, prezygapophysis; SPDL, spinodiapophyseal lamina;
SPRL, spinoprezygapophyseal lamina. Scalebar = 100 mm. Parts I, J and K modified from
Osborn and Mook (1921).

**Figure 2.** *Amphicoelias altus* holotype (AMNH FARB 5764) posterior dorsal vertebra in
anterior (A, E), right lateral (B, F), posterior (C, G), left lateral (D) and ventral (H, anterior
toward top) views. Blue polygon in C obscures hand used to brace the fragile vertebra for
photography. Abbreviations: CPOL, centropostzygapophyseal lamina; CPRL+ACPL, conjoined
centroprezygapophyseal lamina and anterior centroparapophyseal lamina; D, diapophysis;
HS, hyposphene; ISPOL, lateral spinopostzygapophyseal lamina; mSPOL, medial
spinopostzygapophyseal lamina; P, parapophysis; PCDL, posterior centrodiapophyseal
lamina; PCPDL, posterior centroparapophyseal lamina; PODL, postzygodiapophyseal lamina;
PRSL, prespinal lamina; SPDL, spinodiapophyseal lamina; SPRL, spinoprezygapophyseal
lamina. Scalebar = 100 mm. Parts E–G modified from Osborn and Mook (1921).

**Figure 3.** *Amphicoelias altus* holotype (AMNH FARB 5764) right pubis in lateral view.
Scalebar = 100 mm.

**Figure 4.** *Amphicoelias altus* holotype (AMNH FARB 5764) right femur in anterior (A), lateral
(B), posterior (C), and medial (D) views. Cross-sectional shape from break at mid-shaft
shown in (E). Scalebar = 100 mm. Parts A–D modified from Osborn and Mook (1921).

**Figure 5.** Conservation history of the holotypic dorsal vertebrae (AMNH FARB 5764) of
*Amphicoelias altus*. The middle–posterior (A–D) and posterior dorsal (E–H) vertebrae were
photographed under UV light (UV-A in parts A, D; combined UV-A, B, and C wavelengths in
parts E, H), and under polarized light to increase contrast (parts B, C, F, G). The vertebrae
are shown in right lateral (A, B, E, F) and anterior (C, D, G, H) views. Note the different types
of filler material, indicating several generations of interventions, and the uniform blueish
colour, which indicates consolidation with shellac. Shellac coating and attached fuzz remains
nearly invisible under normal and polarized light. The individual photographs were not
modified with any kind of editing software. Photographs provided by M. Eklund (a complete
set of Progressive Photonics photographs is available in the electronic supplementary
material).

**Figure 6.** AMNH FARB 5764, tooth, in labial (A), lingual (B), and occlusal (C) views. Scalebar =
50 mm.

**Figure 7.** AMNH FARB 5764a [Cor. 3], left coracoid (A, C), and AMNH FARB 5764a [Sc. 7], left
scapula (B, D), in lateral view. Scalebar = 100 mm. Parts C and D modified from Osborn and
Mook (1921).

**Figure 8.** AMNH FARB 5764a [Ul. 1], left ulna, in lateral (A), anterior (B), medial (C), posterior
(D), proximal (E), and distal (F) views. Scalebar = 100 mm. Parts C, E and F modified from
Osborn and Mook (1921).

**Figure 9.** AMNH FARB 5764a [Fem. 2], distal femur, in anterior (A) and posterior (B) views.
Scalebar = 100 mm.

**Figure 10.** AMNH FARB 5761 [Pb. 6], left (?) pubis, in presumed lateral (A) and medial (B)
views. Scalebar = 100 mm.

**Figure 11.** AMNH FARB 5761 [H.1], right humerus, in anterior (A) and posterior (B) views.
Scalebar = 100 mm.

Figure 1. *Amphicoelias altus* holotype (AMNH FARB 5764) middle-posterior dorsal vertebra in anterior (A, E, I), right lateral (B, F, J), posterior (C, G, K), left lateral (D, H), and ventral (L, anterior towards top) views. Grey tone indicates anatomy reconstructed or obscured by matrix; hatched areas indicate broken surfaces. Abbreviations: D, diapophysis; CPOL, centropostzygapophyseal lamina; CPRL, centroprezygapophyseal lamina; P, parapophysis; PCDL, posterior centrodiapophyseal lamina; PCPL, posterior centroparapophyseal lamina; PRDL, prezygodiapophyseal lamina; PRZ, prezygapophysis; SPDL, spinodiapophyseal lamina; SPRL, spinoprezygapophyseal lamina. **Scalebar = 100 mm.** Parts **I, J and K** modified from Osborn and Mook (1921).

149x214mm (300 x 300 DPI)

Figure 2. *Amphicoelias altus* holotype (AMNH FARB 5764) posterior dorsal vertebra in anterior (A, E), right lateral (B, F), posterior (C, G), left lateral (D) and ventral (H, anterior toward top) views. Blue polygon in C obscures hand used to brace the fragile vertebra for photography. Abbreviations: CPOL, centropostzygapophyseal lamina; CPRL+ACPL, conjoined centroprezygapophyseal lamina and anterior centroparapophyseal lamina; D, diapophysis; HS, hyposphene; **ISPOL**, lateral spinopostzygapophyseal lamina; **mSPOL**, medial spinopostzygapophyseal lamina; P, parapophysis; PCDL, posterior centrodiaepophyseal lamina; PCPD, posterior centroparapophyseal lamina; PODL, postzygodiapophyseal lamina; PRSL, prespinal lamina; SPD, spinodiapophyseal lamina; SPRL, spinoprezygapophyseal lamina. **Scalebar = 100 mm.** Parts E–G modified from Osborn and Mook (1921).

182x225mm (300 x 300 DPI)

Figure 3. *Amphicoelias altus* holotype (AMNH FARB 5764) right pubis in lateral view. Scalebar = 100 mm.

68x145mm (300 x 300 DPI)

Figure 4. *Amphicoelias altus* holotype (AMNH FARB 5764) right femur in anterior (A), lateral (B), posterior (C), and medial (D) views. Cross-sectional shape from break at mid-shaft shown in (E). Scalebar = 100 mm. Parts A–D modified from Osborn and Mook (1921).

145x153mm (300 x 300 DPI)

Figure 5. Conservation history of the holotypic dorsal vertebrae (AMNH FARB 5764) of *Amphicoelias altus*. The middle–posterior (A–D) and posterior dorsal (E–H) vertebrae were photographed under UV light (UV-A in parts A, D; combined UV-A, B, and C wavelengths in parts E, H), and under polarized light to increase contrast (parts B, C, F, G). The vertebrae are shown in right lateral (A, B, E, F) and anterior (C, D, G, H) views. Note the different types of filler material, indicating several generations of interventions, and the uniform blueish colour, which indicates consolidation with shellac. Shellac coating and attached fuzz remains nearly invisible under normal and polarized light. The individual photographs were not modified with any kind of editing software. Photographs provided by M. Eklund (a complete set of Progressive Photonics photographs is available in the electronic supplementary material).

Figure 6. AMNH FARB 5764, tooth, in labial (A), lingual (B), and occlusal (C) views. Scalebar = 50 mm.

72x132mm (300 x 300 DPI)

Figure 7. AMNH FARB 5764a [Cor. 3], left coracoid (A, C), and AMNH FARB 5764a [Sc. 7], left scapula (B, D), in lateral view. Scalebar = 100 mm. Parts C and D modified from Osborn and Mook (1921).

160x139mm (300 x 300 DPI)

Figure 8. AMNH FARB 5764a [Ul. 1], left ulna, in lateral (A), anterior (B), medial (C), posterior (D), proximal (E), and distal (F) views. Scalebar = 100 mm. Parts C, E and F modified from Osborn and Mook (1921).

117x94mm (300 x 300 DPI)

Figure 9. AMNH FARB 5764a [Fem. 2], distal femur, in anterior (A) and posterior (B) views. Scalebar = 100 mm.

68x72mm (300 x 300 DPI)

Figure 10. AMNH FARB 5761 [Pb. 6], left (?) pubis, in presumed lateral (A) and medial (B) views. Scalebar = 100 mm.

74x92mm (300 x 300 DPI)

Figure 11. AMNH FARB 5761 [H.1], right humerus, in anterior (A) and posterior (B) views. Scalebar = 100 mm.

74x83mm (300 x 300 DPI)

Appendix D

Response to reviewers

Editor comments to author:

Thanks for your submission. I am going to redirect the AE's recommendation from "major revision" to "reject/resub" mainly because the 3-week timeframe for a revision may be too short. Both reviewers have useful comments but Reviewer 2 brings up some perceived deficiencies that should require attention, notably improvements in the descriptions and illustrations, and the lack of a phylogenetic analysis (which should come naturally with a revised diagnosis). The perception that you are dismissive of some arguments of other authors should also be addressed. Please attend to these comments in your revision, and we look forward to a resubmission. Best wishes.

We have responded to all of the reviewers' comments below and strived to improve and rectify issues pertaining to both the text and illustrations. In terms of a phylogenetic analysis, the reasons for its exclusion is that the lead author published a new phylogeny of the group just last year (Mannion et al. 2019 in *ZJLS*) that included the anatomical information presented herein in that data matrix, but we realise this was not as clearly explained in the MS as it could have been. We disagree with Reviewer 2 that our review of the literature is dismissive of other authors (with the exception of the accidental instance pointed out by Reviewer 1) or that it is an argument from authority; however, we have now used this recently published phylogeny (and two others) to test alternative placements of *Amphicoelias*, which we believe also strengthens our Discussion. We have also followed Reviewer 2's suggestion of reframing the discussion in terms of hypotheses and expected outcomes.

Reviewer: 1

Amphicoelias is one of the least-studied sauropods from the Late Jurassic Morrison Formation and the authors provide a detailed, thoughtful account that provides new anatomical information and establishes its taxonomic status more firmly. They use previous suggestions that *Amphicoelias* was an adult individual of another Morrison sauropod taxon to initiate a useful discussion on sauropod diversity and palaeoecology, and the influence of ontogeny on taxonomic decision making. My comments are all relatively minor and most are provided on the annotated .pdf (attached).

Please see responses to those comments from the annotated PDF (copy and pasted after the reviewer's five summary comments).

They can be summarized in general as follows:

- 1. Although the authors do a thorough job in diagnosing *Amphicoelias*, a few more comparisons with other Morrison diplodocoids would reinforce their conclusions regarding the distinctiveness of *Amphicoelias* relative to the other taxa in the 'fauna', rather than relying on coarser clade-level comparisons.**

We have now added in additional comparisons with Morrison Formation sauropod genera and species throughout the Description following this suggestion. Related to this, we have also added in a comparative figure (Fig. 6) following the reviewer's subsequent comment.

2. The authors might consider some improvements to the figures (labeling of a few more key features mentioned in the text; comparative images to support the identification of autapomorphies).

Following Reviewer 2's more substantial comments regarding our figures, these have been overhauled in general, including greater annotation. As mentioned in the previous response, we have also added in far more comparisons with contemporaneous/closely related taxa to support our identification of autapomorphies, in addition to including a comparative figure (Fig. 6).

3. Providing further character evidence (or noting the lack of character evidence) to support the identifications of specimens previously referred to *Amphicoelias* that they now regard as 'Eusauropoda indet.'

We have added in further character evidence in this section. Reviewer 2 requested that we delete the descriptive part of this section, but we feel that these remains should be dealt with: if not here, probably no-one will ever revise these remains and they are pertinent to the history of *Amphicoelias* and thus warrant inclusion in this MS.

4. Addition of one or two further references.

These have been added in the relevant places.

5. Some minor typos/phrasing issues.

These have been corrected in the relevant places.

PDF comments (other than typos):

[Note that text that is quoted from the MS to provide context is in italics to differentiate from reviewer's comments]

Maybe cite Foster (2003, Bull New Mex Mus Nat Hist Sci) and his Indiana University Press books, which also provide recent inventories.

We have added Foster (2003, 2007) to this introduction on valid species in the Morrison Formation. Note that both were already cited elsewhere.

It might be useful to the reader if you say what the current taxonomic ID and provenance of these other species is so that they can be contrasted with that of *A. altus*.

We have revised this section to highlight where these species are from ("nearby Morrison Formation localities") and their taxonomic IDs ("*Amphicoelias latus* is universally regarded as

a synonym of *Camarasaurus supremus* (e.g. Osborn and Mook, 1921; McIntosh, 1990; Tschopp et al., 2015) and *Amphicoelias fragillimus* was recently referred to the newly erected rebbachisaurid genus, *Maraapunisaurus*, by Carpenter (2018)”)

Would be good to label the TPRL on Fig. 1.

This is now labelled (now Fig. 4).

Many of these features (e.g. PRSL, PRSDF, mSPOL) are not labelled on Fig. 1 but it might be useful to do so, which would also help the reader distinguish more easily between features that are original (labelled) vs reconstructed (unlabelled)

These are now labelled (now Fig. 5).

A comparative figure showing the difference between the neural spine summit of *A. altus* and a selection of other Morrison diplodocids might be useful.

This has been added and is now Fig. 6.

“OTU” - This abbreviation not defined

We have now defined this on its first usage here, i.e. operational taxonomic unit.

“Anterior to this ridge, the acromion expands dorsoventrally”: I don't recall seeing a sauropod scapula with this combination of features, so this could be taxonomically distinctive - potentially another apomorphy of the taxon if the material were considered to be part of the type. If not, it might indicate another possible otherwise unknown Morrison taxon.

We agree – this does seem unusual. The only taxon we've been able to find which has a broadly similar morphology is *Camarasaurus lentus* – we have now added in this comparison.

“We conservatively regard the AMNH FARB 5764a scapula as belonging to an indeterminate eusauropod”: might be worthwhile listing a few features that exclude it from major clades such as Diplodocoidea or Titanosauriformes (or noting that all of its preserved characters are widespread in sauropods) to back up the ID as Eusauropoda indet. At the moment, this is an assertion, it's not character-based.

We have now provided more comparisons to support our conclusions regarding its affinities, although we still cannot go further than Neosauropoda indet. given that it has features linking it with *Camarasaurus* (see above) and apatosaurines.

“We consider the AMNH FARB 5764a coracoid as representing an indeterminate eusauropod”: same comment as above. Would be good to cite characters that narrow down the ID (or that, conversely, show that the ID cannot be narrowed further). Again, this is currently an assertion.

We have followed the same approach outlined for the scapula, but here we are able to better narrow down the likely affinities of the coracoid, which we now identify as aff. *Camarasaurus*. We also note that if the scapula is from the same individual as the coracoid, which is possible, this would help to reconcile the identification of the former element.

“and the absence of synapomorphies of other taxa”: such as? Would be useful to say that the absence of these features rules out other IDs.

We have clarified this as suggested by the reviewer.

“As such, this distal femur clearly differs from that of *Amphicoelias altus* and should be regarded as an indeterminate eusauropod”: again, what characters support this ID or reject placement in other clades?

As with preceding elements, we have now revised this, suggesting that the combination of features indicates an identification as a diplodocine (and we note that it could be *Diplodocus*).

When you say 'tentatively identified' do you mean registered with this name? If so, you could emphasize that there is no documentation providing the rationale for these IDs. At the moment, the phrasing makes it unclear if this is the author's tentative ID or someone else's and if someone else's what the basis for the ID was.

We have revised these two sentences to better clarify that the ID is not ours: “Two pubes and a humerus from the Morrison Formation of the Garden Park area in the AMNH collections have been tentatively identified as *Amphicoelias altus* in the accession records. No basis for these identifications is documented but they were presumably based on locality and/or gross morphology.”

Again, a little more info to reject alternative IDs useful, even if only statements like - "it possesses no features currently considered to be synapomorphic of any sauropod taxa and it's preserved anatomy is consistent with that of numerous other eusauropods. As a result, we regard it as an indeterminate eusauropod". Something like this for each indet. ID would show you've considered all of the other possibilities.

We have followed this suggestion for the pubis.

How do you know that *A. altus* is an adult? You've not mentioned the ontogenetic status of the material anywhere else in the paper and it is key to this part of the discussion. It would be useful to comment on any ontogenetically informative features in the Description to establish its age before you reach the Discussion.

Good point – I think because it's a pretty big animal we completely forgot the need to say anything about this! We have now added a short paragraph about the probable ontogenetic status of the material as a new section at the end of our Description. Unfortunately, the AMNH would not allow us to carry out a histological study on the material, but other evidence all supports the view that this is an adult individual, as now presented.

“Despite framing his study with the ‘problem’ of having 24 contemporaneous megaherbivore species to support his claim that Morrison sauropod diversity is overestimated, Woodruff (2019: p. 99–105) was clearly aware that this is incorrect”: this wording might be construed as an ad hominem attack - would be better to rephrase very slightly and criticize the idea rather than the person.

We thank the reviewer for pointing this out and this was not our intention. We have revised this part of the sentence to now read: “Although framed around the ‘problem’ of having 24 contemporaneous megaherbivore species, Woodruff (2019: fig. 5) showed the Morrison Formation...”.

Also Button et al. (2014) in Proc B

We have added this reference to our examples of work on sauropod palaeoecology.

Reviewer: 2

Description: The description is for the most part good, but there are two ways that it could be improved considerably, particularly for the two dorsal vertebrae. First, there needs to be more clarification about what is preserved and missing and what is reconstructed. The description includes details on aspects of the morphology of the prezygapophyses that are illustrated as missing/reconstructed in the interpretive diagram. Second, there is a mismatch between what is described in text and what one can see from the photographs and interpretive line drawings done by the authors. There are claims about laminae, for example, that are difficult to verify from the images and labeling provided. There are also some minor quibbles I had with the laminar terminology (e.g., use of “ISPOL” instead of “lat SPOL”), which are detailed in comments on the PDF.

We have revised our Description to more carefully clarify what is/isn't preserved and to make it a better match with the figures. In addition, we have simplified our figures, now just showing the unedited photographs of the vertebrae (with the original drawings from Osborn and Mook 1921 reproduced as separate, unedited figures), with our text outlining what is/isn't preserved. We have also added the following sentence to the beginning of the Description: “Although substantial portions of the two vertebrae have undergone reconstruction, it is still possible to discern genuine morphology, even in places where the bone is entirely coated with consolidant and other materials”. We have retained ISPOL, which has been widely used in previous publications and was also defined in our anatomical abbreviations section.

Figures: The figures are not up to par. As I mentioned above, it is difficult to identify key structures on the interpretive drawings. These latter images are not well done, with labeling set in too close to the object, leader lines almost running into labels, etc. I have left a series of comments in both figures of the vertebrae to help the authors. At least one of them (John Whitlock) will have heard this previously.

We have heavily revised our figures following these suggestions.

Systematics: The title promises systematics, but there is no formal analysis of the interrelationships of *Amphicoelias* in this paper. This is a serious omission that diminishes the strength of arguments about the phylogenetic affinities of the taxon, including the claim that it is not synonymous with *Diplodocus* (see Section 5.1.). It is difficult to imagine that with three authors with experience in this group that there is no analysis.

We explained its original omission in our response to the Editor (i.e. that the lead author published a new phylogeny of the group just last year that included the anatomical information presented herein in that data matrix), but we have now incorporated constrained analyses based on three phylogenetic data matrices to test the position of *Amphicoelias* – see also more detailed response below.

Discussion: Following the more systematic part of the discussion is a consideration of the implications of the validity of *Amphicoelias* for Morrison Fm. diversity (section 5.2.). The authors discuss whether previous claims about certain Morrison Formation sauropod species being 'ontogimorphs.' I agree that the Morrison is not rife with 'ontogimorphs' posing as real taxa, but the form of argumentation made by the authors here is one from authority and therefore weak. If the authors wish to address this issue, I suggest they outline the predictions of the ontogimorph hypothesis and then counter them one by one. Ironically, one of the obvious counters is about the phylogenetic affinities of *Amphicoelias*, which they do not test in this paper. Without this, this section dissolves into an exercise in arm-waving that won't carry more weight than the original by Woodruff and co-authors. The ontogimorph section is followed by a lengthy section evaluating the claim that the sauropod diversity is too great for the Morrison Formation. Again, the authors argue authoritatively, in this case by simply dismissing Woodruff (and to some extent Farlow). They never provide a robust defense (or proposal) that the Morrison ecosystem could in fact support the implied diversity.

We disagree with the reviewer that this discussion section is either “one from authority” or “arm-waving”. The discussion section contains a review of disparate lines of evidence that we synthesise to formulate our arguments. As noted earlier, we had not included a new phylogenetic analysis simply because the lead author had published one recently that included the anatomical information presented in detail herein. We referred to this result (and that of others) at the start of this Discussion. We also did outline some of the expectations of the ontogimorph hypothesis and counter them (e.g. pertaining to stemward slippage). Furthermore, we didn't dismiss the work of Farlow – we used it to support our argument – but we have reworded it slightly in our revised MS in case our original use of “attempted to” unintentionally came across as dismissive. However, we have taken the reviewer's overall criticisms on board, incorporating constraint tests on three phylogenetic analyses and reframing parts of the Discussion as suggested.

PDF comments (other than typos):

What evidence supports the hypothesis that these elements pertain to a single individual?

They were collected from one locality, with no element duplication or size/preservation discrepancy, and they are all from the same region of the skeleton. This information has been incorporated into the Systematic section, including a statement that we consider them to pertain to one individual following this and based on previous authors.

Middle–posterior dorsal vertebra: please provide detail on how serial position was determined.

This has now been added at the start of the Description (and is based on comparisons with sauropods preserving complete dorsal sequences).

“An anteriorly and posteriorly incomplete centrum, most of the neural arch, and the base of the neural spine of a middle–posterior dorsal vertebra is preserved”: This is a strange way to write a sentence ... that is, to list a series of 3 items with adjectival additions like "anteriorly incomplete") and then, after it all to say "is preserved". It seems more straightforward to start with what is is (a middle-posterior dorsal vertebra) and state which parts are preserved and which are missing. Also note that the current phrasing makes it sound like the neural spine is not part of the neural arch, which it is.

This sentence has been revised, following the suggestion of the reviewer.

“The lateral pneumatic foramen is fairly consistent on each side of the centrum”: from the images in Figure 1, there does not appear to be a consistent size or position the one side is completely reconstructed and shown as missing in the interpretive diagram

We’ve removed the statement about consistency.

“Each foramen is deep, leaving a thin midline septum and, internally, it ramifies strongly dorsally and, especially ventrally, as well as a small distance anteriorly and posteriorly”: Is this based on CT data or visual inspection of the specimen?

This is purely based on visual inspection – we’ve tweaked the text and removed “internally”, which we suspect was the accidentally misleading part of the sentence.

“The sharp lateral margin of the centroprezygapophyseal lamina (CPRL) also forms the anterior centroparapophyseal lamina (ACPL)”: this is not apparent from Figure 1 ... not labeled as such either

Following the reviewer’s similar comment on the other dorsal vertebra, we have revised this to describe these laminae as merging. This has also now been labelled on the figure (now Fig. 5).

This is more apparent in Osborn & Mook's figures than in the photos or your interpretive drawing. So, do you interpret this [the PCPL] to join the pcpl?

Yes – now added in.

PACDF – not labelled

This has now been labelled (now Fig. 4).

“Although this region is heavily reconstructed...”: according to your illustration, it is **COMPLETELY** reconstructed. It isn't clear to (and won't be clear to the reader) how you are interpreting morphology that is either not preserved or reconstructed by plaster (or whatever). Same goes for your comments about the shape/size/position of the pleurocoels.

We agree with the reviewer that our attempt to capture the reconstruction in our figures didn't work and was confusing. As noted above, we have now gone with unedited figures and left the identification of what is/isn't preserved to our description. In instances such as this feature, the morphology is clear despite the reconstruction.

“The prezygapophyseal articular surfaces are gently convex mediolaterally”: Do you mean dorsally convex? Again, this does not show up in your interpretive drawing, so it isn't clear how you're making this assessment.

No – we do mean mediolaterally: we're talking about the articular surface, which is the dorsal surface. It's subtle, which is difficult to see from the images.

“The postzygapophyseal articular surfaces are gently concave, suggesting that the morphology of the prezygapophyseal surfaces might be genuine”: seems odd to insert this uncertainty after describing it

Now deleted.

“The diapophysis projects laterally and is dorsally deflected, but it has been broken and slightly deformed, and so this dorsal deflection might merely be artefactual”: seems difficult to say given what's preserved. Could you do more to resolve this and other ambiguities? It is less than helpful to describe something that may or may not be real.

We've reworded this sentence such that we acknowledge that we can't be certain, but that the reconstructed morphology is consistent with that of the other, better preserved, dorsal vertebra.

“The spinoprezygapophyseal laminae (SPRLs)...merge close to the apex of the incomplete neural spine”: it isn't clear here whether they actually merge on the part of the nsp that is preserved, or whether this is an interpretation ... that they merge close to the apex of the nsp, which is absent. There is no PRSL labeled in figure 1.

We have simplified this just to say that they merge dorsally and added the PRSL to the figure (now Fig. 5).

SPDL - this should be labeled in lateral view on the interpretive diagram

This has now been labelled in lateral view (now Fig. 4).

Posterior dorsal vertebra: please provide detail on how serial position was determined.

This has now been added at the start of the Description (see previous response for the other dorsal vertebra).

Please state what's missing/reconstructed

We have added that it is missing the left diapophysis.

“The centrum is slightly dorsoventrally taller than wide (note that the anterior surface is incomplete along its left side, and the posterior surface along its right side)”: I find this slightly ambiguous as stated. It isn't clear that you're stating the preserved proportions or the proportions that we'd expect if the centrum were complete (e.g., by doubling a complete measurement from the center of the centrum face to one side [width] or to top or bottom [height]).

This has been reworded to first describe the preservation and then to state the proportions, clarifying that this is with the width extrapolated.

“The ventral surface of the centrum is gently convex transversely”: but not dorsoventrally?

The reviewer must mean anteroposteriorly, rather than dorsoventrally, and this has now been clarified in the Description.

“The neural arch extends...”: seems clearer to refer to its pedicle, since parts of the arch extend beyond the centrum anteriorly and posteriorly.

We have added 'pedicles' after 'neural arch'.

“Each CPRL also forms the ACPL”: This formulation doesn't make sense. It seems more likely to say they are merged at a certain point rather than to say one "forms" the other (especially since there are parts of both that are not merged). The CRPL "comprises" a ridge?

We have revised this to use 'merge' and replaced the offending term ('comprises') with 'forms', such that this now reads: “...each CPRL merges with the ACPL and forms a sharp ridge”.

“There is no evidence for an anterior centrodiaepophyseal lamina (ACDL) on either side, and this appears to be a genuine absence”: this is often/usually the case when the para is elevated on the n. arch.

An ACDL remains present in the middle–posterior dorsal vertebrae of most titanosauriforms though, so there is some phylogenetic signal in this (now added).

“...at an angle of approximately 45° to the horizontal”: this raises the question of how you are orienting the vertebra in space. That is, how are you establishing orientational planes?

We're orientating it in the standard way (i.e. the way that everyone does), with the floor of the neural canal held horizontally, which is also how it's figured.

"If our interpretation is correct...": I don't follow. You're not sure of your identification of the side? Why leave this ambiguous in a redescription?? What are the points in favor of the opposite interpretation?

The preceding two sentences stated: *"The missing portions include the acetabular margin, the ambiens process, the obturator foramen, and the complete articular surface for the ischium, which makes it difficult to determine whether this is a left or right pubis. However, we interpret the pubis as being from the right side, consistent with the holotypic femur, and in contrast to the interpretation of Osborn and Mook"*. As such, we have already clearly stated that we're unsure of the identification. We think ambiguity is better than incorrectly saying that we're sure that this is a right pubis. We have changed the word 'If' to 'Assuming that' to reflect that we have greater confidence in our ID than originally expressed, but we think it is better to leave open the possibility that it's not possible to be certain because of preservation.

"The right femur is...largely complete": what is missing?

The next line states: *"However, there is material missing throughout, with lots of plaster reconstruction in places"*. We've clarified what's missing further in this sentence.

"The femur has also undergone transverse compression": This seems a very odd type of distortion. What is the evidence for this?

We mean compression in the transverse (mediolateral) axis of the femur, whereas we suspect that the reviewer (understandably) confused this with actual transverse compression, which we agree would be unusual. We have now simply said that it might have been affected by crushing to remove any ambiguity.

"The posterior surface of the proximal end, towards the lateral margin, forms a concavity, giving the initial impression of a trochanteric shelf, such as that seen in several rebbachisaurids (Whitlock, 2011a) and many somphospondylans (Mannion et al., 2013). However, this concavity seems to be merely the product of crushing": Again you seem to describe a structure, make comparisons, and then explain it away as distortion. Why not lead with ascribing the feature to distortion?

In this particular case we feel it useful to clarify what this feature is and how it might be (mistakenly) interpreted, and then explain that it's almost certainly an artefact. It's also easier to explain what we're referring to by describing it in the first place.

"contrasting with the reduced processes that characterise the femora of some rebbachisaurids (Mannion et al., 2012)": most sauropod femora have reduced fourth trochanters

Although we fully agree with the reviewer, we are here referring to the near absence of this structure in some rebbachisaurids. We realise this was not clear and have now revised it accordingly: “contrasting with the femora of some rebbachisaurids, in which this process is barely discernible”.

“Most sauropod femora have transversely expanded, elliptical midshaft cross sections (Wilson, 2002), with ratios of the mediolateral to anteroposterior diameters exceeding 1.5 (Mannion et al., 2012, 2013)”: Not to be too picky here, but this is a feature that Wilson and Carrano (1999) discussed and provided data for (see their tables 1 and 2).

We have added this suggested reference as the first paper to really discuss this feature and provide data. Wilson and Carrano (1999) provided a table with a mean value for titanosaurs and a mean value for other sauropods, which showed that the latter had an average of 1.5 (no other data was reported), whereas Mannion et al. (2013) presented the raw values for 25 eusauropod species (note that the 2012 citation was an error and has now been removed). The original sentence in the MS makes the point that nearly all eusauropods have values above 1.5 (now modified to 1.4 in the MS), which is not something you can easily extract from the table in Wilson and Carrano (1999). All three references (Wilson and Carrano, 1999; Wilson, 2002; Mannion et al. 2013) are also now just placed together, at the end of the sentence.

“...that we therefore regard as autapomorphic for the femur of *Amphicoelias*”: Wilson and Smith (1996) did as well — this was part of the rationale for assigning MOR 592 to this taxon.

We have added this reference in here. We had cited this abstract in this regard elsewhere, so it was not our intention to claim that this was a novel finding.

“Distally, the femur is bevelled, with the fibular condyle extending further distally than the tibial condyle. This type of bevelling is generally restricted to derived titanosaurs (Wilson, 2002), and is therefore regarded as a local autapomorphy of *Amphicoelias*”: What? Above you state: “With the exception of the anterior portion of the distal condyles, the distal tip of the femur is entirely reconstructed.”

The anterior portion of the distal condyles provides enough information to determine the nature of the bevelling – the distal surface of the anterior part of the femur is always essentially level with that of the remainder of this surface. We have now revised the start of the sentence so that it reads: “Based on the preserved anterior portion, the femur is bevelled at its distal end, with...”

“...and it is not clear that any of them came from the type locality”: I would appreciate more detail here. Is there no evidence that the bones come from there, or is there evidence that the bones come from somewhere else?

We have clarified this to say that there’s no evidence that the bones are from the type locality.

“However, for the sake of completeness, we re-describe and figure all of these previously referred elements here, re-evaluating their taxonomic affinities”: I would argue the

opposite. If these elements don't pertain to *Amphicoelias* and they aren't from the same quarry, then it doesn't make sense to describe them together with remains that are *Amphi.* from the same quarry. I think the discussion of why they don't belong is useful, but the description isn't.

Following our response to Reviewer 1, who requested more information on these remains, we prefer to retain the description of these previously referred remains, many of which are historically tied up with *Amphicoelias altus*, one way or another. We have removed “for the sake of completeness” though, which is not the primary reason for their inclusion. We also cannot justify our identifications of these specimens without some aspect of description, which would be an argument from authority.

“A right humerus was apparently found at the same site as the *Amphicoelias altus* holotypic material”: is there reason to discount whatever conclusions McIntosh (1998) came to?

There isn't and we've now removed “apparently”: the specimen was collected 5 years after the type material though, and this detail has now been added.

“Here, we refer this humerus to *Diplodocoidea*, and note the possibility that it could belong to *Amphicoelias*”: Based on what?? Why leave this ambiguous? McIntosh's reasoning makes sense. Are there examples of sauropods with slender femur proportions and robust humeral proportions? If so, point that out. but if not, and the entirety of the group is built the other way (presumably based on biomechanical principles) then it doesn't seem reasonable to hold out the possibility that this is *Amphi.*

We have flipped this around so that we start by leaving open the possibility that it could ultimately belong to *Amphicoelias*, given that it comes from the same locality, but now finish with saying that we follow McIntosh in excluding it from that taxon (for the reasons mentioned by the reviewer) and refer it to *Diplodocoidea*.

“Woodruff and Foster (2014) incorrectly stated that previous authors (i.e. Osborn and Mook, 1921; McIntosh, 1998) had synonymized ‘*Amphicoelias latus*’ with *Amphicoelias altus*, a taxonomic assignment they ‘agreed’ with”: If it is a quotation (double quotes) please include a page number.

It isn't quite a quotation as they used the word “agrees”, but the inverted commas were an attempt to capture this. As such, we've revised the text to present the quotation instead (using double quotes) and provided a page reference.

“...and tentatively concur that *Maraapunisaurus* might represent a rebbachisaurid”: Really? Ok. That would be a significant statement to make — what is the justification for this attribution? And where are all the other rebbachisaurid bones that we presumably should have in the Morrison Fm?

We're not sure that “tentatively concur” that it “might represent a rebbachisaurid” can really be considered a significant statement to make! Given that the affinities of *Maraapunisaurus* are largely tangential to our paper, and that we do not think that this is the right place to re-

evaluate this, we have added “following the arguments presented by Carpenter (2018)” as our justification for this attribution. We also don’t consider the last comment of the reviewer an angle we can use given that our paper argues that *Amphicoelias altus* is a valid species known from only one specimen. We don’t make this point in the MS because it is too tangential, but rebbachisaurids had to have been somewhere during the Late Jurassic based on ghost lineages and thus their absence from sampled faunas suggests that either they were living in areas we are yet to sample, or they were rare components of their ecosystems.

“Photographs of the femur of MOR 592, provided by C. Woodruff, reveal that the ratio of the mediolateral to anteroposterior diameters of its midshaft is >1.3”: why not provide the actual ratio here?

This was intended to reflect that this was an estimate from the photographs and so we can only estimate a ratio that is around or slightly above 1.3 – this has been changed to ‘approximately 1.3’.

“Is *Amphicoelias altus* synonymous with *Diplodocus*?”: This is a phylogenetic question, but there is no phylogeny in this paper, despite being written by a team of three researchers who have done previous analyses of Diplodocoidea! In that context, the claims about where *Amphicoelias* fits in strike me as argument from authority rather than analyses.

As mentioned in our response to the Editor, the reasons for its exclusion is that the lead author published a new phylogeny of the group just last year that included the anatomical information presented herein, but we realise this was not as clearly explained in the MS as it could have been. We do not understand why we cannot utilise information from previous work to develop an argument. The original submission contained the following sentences at the start of this section: “*Our detailed redescription, combined with recent phylogenetic analyses (Rauhut et al., 2005; Whitlock, 2011a; Mannion et al., 2012, 2019; Tschopp et al., 2015; Tschopp and Mateus, 2017), supports Amphicoelias altus as a valid taxon, diagnosed by three autapomorphies. We cannot currently determine whether Amphicoelias was a ‘basal’ diplodocoid (e.g. Mannion et al., 2019) or an apatosaurine (e.g. Tschopp and Mateus, 2017), but it seems unlikely that it was a member of Diplodocinae*”. We are unclear why this argument is from authority, but if we include a new analysis in our current paper it becomes OK. Regardless, we have now used this recently published phylogeny (and others) to test alternative placements of *Amphicoelias*.

I don't disagree with the conclusion that the Morrison is not rife with 'ontogimorphs' posing as real taxa. . . that makes sense to me — but the form of argumentation here is one from authority and therefore weak. The authors should set out predictions of the hypothesis and then counter them one by one. Without this, this section dissolves into an exercise in arm-waving that won't carry more weight than the original.

Although we disagree with the reviewer here in terms of authority (see earlier response), we have followed their advice in reframing aspects of the discussion.

“Stemward slippage”: explain what you mean by this in the context of 'ontogimorphs'. Actually resolved at a more basal position or unresolved and dropping there in a consensus tree?

We meant actually resolved in a more basal position – this has now been clarified in the text at its first usage.

“None of the type specimens of Morrison Formation sauropod species regarded as potential ontogimorphs are interpreted as juveniles”: Smitanosaurus is a subadult. Perhaps that raises the issue of explaining what you mean by these terms. How are you dividing up ontogenetic series ...is "juvenile" equivalent to "subadult"?

We had tried to explain this in the next sentence, but realise it is better coming beforehand. As such, we have added the following sentence early on in this Discussion section: “Here and below, we follow Hone et al. (2016) in broad definitions of ‘juveniles’ versus ‘subadults’, with specimens in the former age class showing little or no skeletal fusion”.

“Furthermore, even the inclusion of juvenile specimens does not necessarily result in their stemward slippage”: more reason for explicit definition and clarification of this term above.

With the revisions to the text following the previous two comments, this sentence should now be fine.

Please state what the evidence would be that would indicate the phylogenetic positions of these taxa were really wrong.

Good point! This has been revised to say that it seems unlikely that these positions could be so grossly incorrect.

“Was apparent Morrison Formation sauropod diversity too high to be ecologically viable?”: Again, I don't necessarily disagree with what is stated here... but this is more a dismissal of Woodruff (and to some extent Farlow) rather than a robust defense (or proposal) that the Morrison ecosystem could support the implied diversity.

Given that we cite Farlow to support our arguments, it's unclear how our discussion is a dismissal of that paper (though see earlier response), but we have taken on board the reviewer's previous comments about reframing this part of the Discussion.